# The *Arabidopsis* SAC9 enzyme is enriched in a cortical population of early endosomes and restricts PI(4,5)P$_2$ at the plasma membrane

**Alexis Lebecq**[1†], **Mehdi Doumane**[1†], **Aurelie Fangain**[1], **Vincent Bayle**[1], **Jia Xuan Leong**[2], **Frédérique Rozier**[1], **Maria del Marques-Bueno**[1‡], **Laia Armengot**[1§], **Romain Boisseau**[3], **Mathilde Laetitia Simon**[1], **Mirita Franz-Wachtel**[4], **Boris Macek**[4], **Suayib Üstün**[2,5], **Yvon Jaillais**[1]*, **Marie-Cécile Caillaud**[1]*

[1]Laboratoire Reproduction et Développement des Plantes, Université de Lyon, Lyon, France; [2]University of Tübingen, Center for Plant Molecular Biology (ZMBP), Tübingen, Germany; [3]Division of Biological Science, University of Montana, Missoula, United States; [4]Interfaculty Institute for Cell Biology, Department of Quantitative Proteomics, University of Tübingen, Tübingen, Germany; [5]Faculty of Biology & Biotechnology, Ruhr-University Bochum, Bochum, Germany

**\*For correspondence:**
yvon.jaillais@ens-lyon.fr (YJ); marie-cecile.caillaud@ens-lyon. fr (MCC)

[†]These authors contributed equally to this work

**Present address:** [‡]Department of Molecular Genetics, Center for Research in Agricultural Genomics (CRAG), CSIC-IRTAUAB- UB, Barcelona, Spain; [§]Genetics Section, Universitat de Barcelona (UB), 08028 Barcelona, Catalonia, Spain and Centre for Research in Agricultural Genomics (CSIC-IRTA-UAB-UB). Edifici CRAG, Catalonia, Spain

**Competing interest:** The authors declare that no competing interests exist.

**Abstract** Membrane lipids, and especially phosphoinositides, are differentially enriched within the eukaryotic endomembrane system. This generates a landmark code by modulating the properties of each membrane. Phosphatidylinositol 4,5-bisphosphate [PI(4,5)P$_2$] specifically accumulates at the plasma membrane in yeast, animal, and plant cells, where it regulates a wide range of cellular processes including endocytic trafficking. However, the functional consequences of mispatterning PI(4,5)P$_2$ in plants are unknown. Here, we functionally characterized the putative phosphoinositide phosphatase *SUPPRESSOR OF ACTIN9* (SAC9) in *Arabidopsis thaliana* (*Arabidopsis*). We found that SAC9 depletion led to the ectopic localization of PI(4,5)P$_2$ on cortical intracellular compartments, which depends on PI4P and PI(4,5)P$_2$ production at the plasma membrane. SAC9 localizes to a subpopulation of *trans*-Golgi Network/early endosomes that are enriched in a region close to the cell cortex and that are coated with clathrin. Furthermore, it interacts and colocalizes with Src Homology 3 Domain Protein 2 (SH3P2), a protein involved in endocytic trafficking. In the absence of SAC9, SH3P2 localization is altered and the clathrin-mediated endocytosis rate is reduced. Together, our results highlight the importance of restricting PI(4,5)P$_2$ at the plasma membrane and illustrate that one of the consequences of PI(4,5)P$_2$ mispatterning in plants is to impact the endocytic trafficking.

## Editor's evaluation

Phosphoinositide phosphates (PIPs) are lipids that can convey distinct identities to different cellular membranes via different phosphorylation patterns. Here, Lebecq, Doumane, and co-authors document the effects of the previously-characterized sac9 mutant, affecting a putative PIP-5-phosphatase in Arabidopsis, on PIP localization and endocytic trafficking. This work confirms that disrupting PI(4,5)P2 localization or abundance can affect endocytic trafficking in plants and will be of interest to the plant and cell biology research fields.

## Introduction

Phosphoinositides constitute a family of low abundance lipids differentially enriched in the membranes of eukaryotic cells (*Platre and Jaillais, 2016*; *Balla, 2013*; *Noack and Jaillais, 2020a*). These versatile lipids can be interconverted into one another. For example, phosphatidylinositol 4,5-biphosphate [PI(4,5)P$_2$] is synthetized from phosphatidylinositol 4-phosphate (PI4P) by PI4P-5 kinases and is dephosphorylated into PI4P by PI(4,5)P$_2$ 5-phosphatase (*Noack and Jaillais, 2017*). Furthermore, PI(4,5)P$_2$ and PI4P are hydrolyzed by phospholipases C (PLC) into diacylglycerol and soluble phosphorylated inositol (*Balla, 2013*). PI(4,5)P$_2$ strictly localizes at the plasma membrane in plants and animal cells (*Van Leeuwen et al., 2007*; *Carim et al., 2019*; *Del Signore et al., 2017*; *Simon et al., 2014*; *Ben El Kadhi et al., 2011*) despite the plasma membrane being constantly turned-over by endocytosis and exocytosis. Throughout this paper, we define endocytosis in its broader sense, including the internalization step *sensus stricto* (i.e. recruitment of cargo and coat components of the 'clathrin-mediated endocytosis' machinery, formation of clathrin-coated pits, scission of the clathrin-coated vesicles and uncoating) followed by the subsequent transport of lipid and proteins through the endosomal system. As such, the endocytosis, or here after the endocytic trafficking, is the process that allows (1) cells to transport particles and molecules across the plasma membrane and (2) the termination of signaling through transport toward the vacuole for degradation. At the plasma membrane, PI(4,5)P$_2$ interacts with a variety of extrinsic membrane proteins such as endocytic protein adaptors (*Zhang et al., 2015*) and actin-regulatory proteins (*Paez Valencia et al., 2016*), which are recruited and/or activated by the binding to PI(4,5)P$_2$. Therefore, PI(4,5)P$_2$ subcellular patterning is likely critical to regulate the recruitment of proteins that act at the plasma membrane, and the cellular processes they mediate, including clathrin-mediated endocytosis.

Consistent with a critical role of PI(4,5)P$_2$ in the recruitment of early clathrin-mediated endocytosis factors, a *pip5k1 pip5k2* double mutant in *Arabidopsis thaliana* (*Arabidopsis*) lacking two ubiquitously expressed PI4P-5 kinases, has abnormal auxin distribution and defective endocytic trafficking of the transmembrane auxin efflux carriers PIN-FORMED 1 (PIN1) and PIN2 (*Tejos et al., 2014*; *Ischebeck et al., 2013*; *Mei et al., 2012*). Furthermore, the *pip5k1 pip5k2* double mutant has an altered dynamic of CLATRHIN LIGHT CHAIN2 (CLC2), with the density of CLC2 foci at the plasma membrane being reduced in the mutant (*Ischebeck et al., 2013*). Overexpression of *Arabidopsis* PI4P-5 kinase 6 in tip-growing pollen tubes induced massive aggregation of the plasma membrane in pollen tube tips due to excessive clathrin-dependent membrane invagination, supporting a role for PI(4,5)P$_2$ in promoting early stages of clathrin-mediated endocytosis (*Zhao et al., 2010*). The inducible overexpression of a highly active human PI4P-5 kinase leads to an increased PI(4,5)P$_2$ production, very strong developmental phenotypes, and heightened endocytic trafficking toward the vacuole (*Gujas et al., 2017*). In addition, we recently showed that inducible depletion of the PI(4,5)P$_2$ from the plasma membrane using the iDePP system leads to a decrease in the fraction of the clathrin adaptor protein AP2-μ and the Src Homology (SH3)-domain containing protein 2 (SH3P2) at the plasma membrane (*Doumane et al., 2021*). Furthermore, FM4-64 uptake experiments confirmed an impact of PI(4,5)P$_2$ depletion from the plasma membrane on bulk endocytic flow.

In animal cells, several PI(4,5)P$_2$ phosphatases are required for the late stages of clathrin-mediated endocytosis (*He et al., 2017*; *Pirruccello et al., 2014*). Many PI(4,5)P$_2$ phosphatases belong to the 5-phosphatase enzyme family, including OCRL and synaptojanins (Syn1/2) in animals, and synaptojanin-like proteins (Inp51p/Snjl1p, Inp52p/Sjl2p, and Inp53p/Sjl3p) in *Saccharomyces cerevisiae*. The *Arabidopsis* genome contains 15 genes encoding 5-phosphatases, but only a few are characterized. Mutation in the 5-phosphatase nine leads to osmotic stress tolerance, with reduced reactive oxygen species production and Ca$^{2+}$ influx (*Golani et al., 2013*). The 5-phosphatase 6/COTYLEDON VASCULAR PATTERN2 (CVP2) and 5-phosphatase 7/CVP2 LIKE 1 (CVL1) are specifically required for vascular differentiation (*Rodriguez-Villalon et al., 2015*; *Carland and Nelson, 2009*; *Carland and Nelson, 2004*). Finally, the *5-phosphatase 15/FRAGILE FIBER 3 (FRA3)* is expressed in developing fibers and vascular cells, which is consistent with the defective fiber and vessel phenotypes seen in the loss-of-function *fra3* mutant (*Zhong et al., 2004*).

Proteins containing SUPPRESSOR OF ACTIN (or Sac1-like) domains constitute another family of phosphoinositide phosphatases (*Zhong and Ye, 2003*). In *Arabidopsis*, there are nine SAC proteins, forming three clades (*Zhong and Ye, 2003*). The first clade is composed of SAC1, a PI(3,5)P$_2$ 5-phosphatase (*Zhong et al., 2005*), and its relatives SAC2 to 5, putative PI(3,5)P$_2$ 5-phosphatases (*Nováková et al.,*

*2014*). The second clade corresponds to SAC7/RHD4 a PI4P 4-phosphatase (*Thole et al., 2008*) and its relatives SAC6 and SAC8 putative PI4P 4-phosphatases (*Song et al., 2021*). The third clade is composed of a single member, a plant-specific protein called SAC9. SAC9 has a unique structure, with a SAC phosphoinositide phosphatase domain at its N-terminus, immediately followed by a putative protein/protein interaction domain (WW domain), and a long C-terminal region of 1104 amino acids where a putative coil-coiled domain is predicted (*Figure 1A*, *Zhong and Ye, 2003*). The *sac9* mutant is dwarf, it constitutively accumulates anthocyanins and it expresses genes from stress response pathways (*Williams et al., 2005*). Loss-of-function alleles of *SAC9* display a threefold increase in $PI(4,5)P_2$ content, together with a decrease in PI4P level, suggesting that it acts as a $PI(4,5)P_2$ 5-phosphatase in planta (*Williams et al., 2005*).

Using in vivo confocal microscopy, we found that SAC9 localizes in a subpopulation of trans-Golgi network/early endosomes (TGN/EEs) enriched in a region close to the cell cortex. Loss of SAC9 results into $PI(4,5)P_2$ mis-patterning at the subcellular level, leading to the accumulation of $PI(4,5)P_2$ in subcortical compartments associated with TGN/EEs. Similar cellular and developmental phenotypes were observed when SAC9 was mutated on its putative catalytic cysteine, suggesting that the phosphoinositide phosphatase activity of SAC9 is required for function. SAC9 interacts and colocalizes with the endocytic component SH3P2. In the absence of SAC9, and therefore when the patterning of the $PI(4,5)P_2$ is compromised, SH3P2 localization is affected and the clathrin-mediated endocytosis is significantly reduced. Thus, SAC9 is required to maintain efficient endocytic uptake, highlighting the importance of restricting the $PI(4,5)P_2$ pool to the plasma membrane.

## Results

### The cysteine 459 in the catalytic domain of SAC9 is required for SAC9 function

We investigated the root phenotype in the already described mutant alleles of SAC9 (*Vollmer et al., 2011*; *Williams et al., 2005*). As previously described, $sac9-1^{-/-}$ and $sac9-3^{-/-}$ knock-out mutants are two times shorter compared to the wild-type (WT) Col-0, at 12 days post-germination (dpg; *Figure 1A-C*). We also observed a three-time decrease in the lateral root density of $sac9-1^{-/-}$ and $sac9-3^{-/-}$ compared to WT plants (*Figure 1B-D*). To confirm that the phenotypes observed were due to the loss-of-function of SAC9, we generated *Arabidopsis* lines expressing SAC9 fused to a fluorescent protein (mCIT-SAC9 and TdTOM-SAC9) under SAC9 native promoter (*SAC9pro*) or the *UBIQUITIN10* promoter (*UBQ10pro*), respectively (*Figure 1*, *Figure 1—figure supplement 1A and B*). We found that both *SAC9pro:mCIT-SAC9* and *UBQ10pro:TdTOM-SAC9* rescued $sac9-3^{-/-}$ mutant phenotypes in two independent homozygous T3 transgenic lines for each construct (*Figure 1B and C*; *Figure 1—figure supplement 1A*). These results indicate that the root phenotypes described above are caused by *SAC9* loss-of-function, and that N-terminally tagged SAC9 proteins are functional.

Next, we mutated the cysteine in the conserved C-x(5)-R-[T/S] catalytic motif found in all SAC domain-containing phosphoinositide phosphatase (*Hsu et al., 2015*; *Hsu and Mao, 2013*; *Figure 1A* and *Figure 1—figure supplement 1A*). Such cysteine-to-alanine substitution was shown to block the catalytic phosphatase activity of other SAC domain-containing proteins (*Manford et al., 2010*; *Tani and Kuge, 2014*), and thus mCIT-SAC9$^{C459A}$ is a putative catalytically dead version of the enzyme. In contrast to wild-type mCIT-SAC9, we could not find any transgenic lines expressing *SAC9pro:mCIT-SAC9$^{C459A}$* that were able to rescue the *sac9-3* phenotype, out of 24 independent lines analyzed in T1 (*Figure 1B, C*, *Figure 1—figure supplement 1A, B*). Further analyses on two independent T3 homozygous lines confirmed these initial results and showed that mCIT-SAC9$^{C459A}$ fusions were stable and accumulated to similar extent as wild-type mCIT-SAC9 (*Figure 1B and C*, *Figure 1—figure supplement 1A, B*). Thus, the putative catalytic cysteine, C459, is required for SAC9 function but not for the stability of the protein, suggesting that the phosphatase activity of SAC9 is participating in the observed phenotypes.

### SAC9 localizes to a population of early endosomes close to the plasma membrane

Some phosphoinositide-phosphatases such as Metazoan Sac1 and yeast Sac1p are able to dephosphorylate in vitro several phosphoinositide species, but display a narrower specificity in vivo (*Hughes et al., 2000*; *Guo et al., 1999*; *Rivas et al., 1999*). Also, some enzymes involved in phosphoinositide

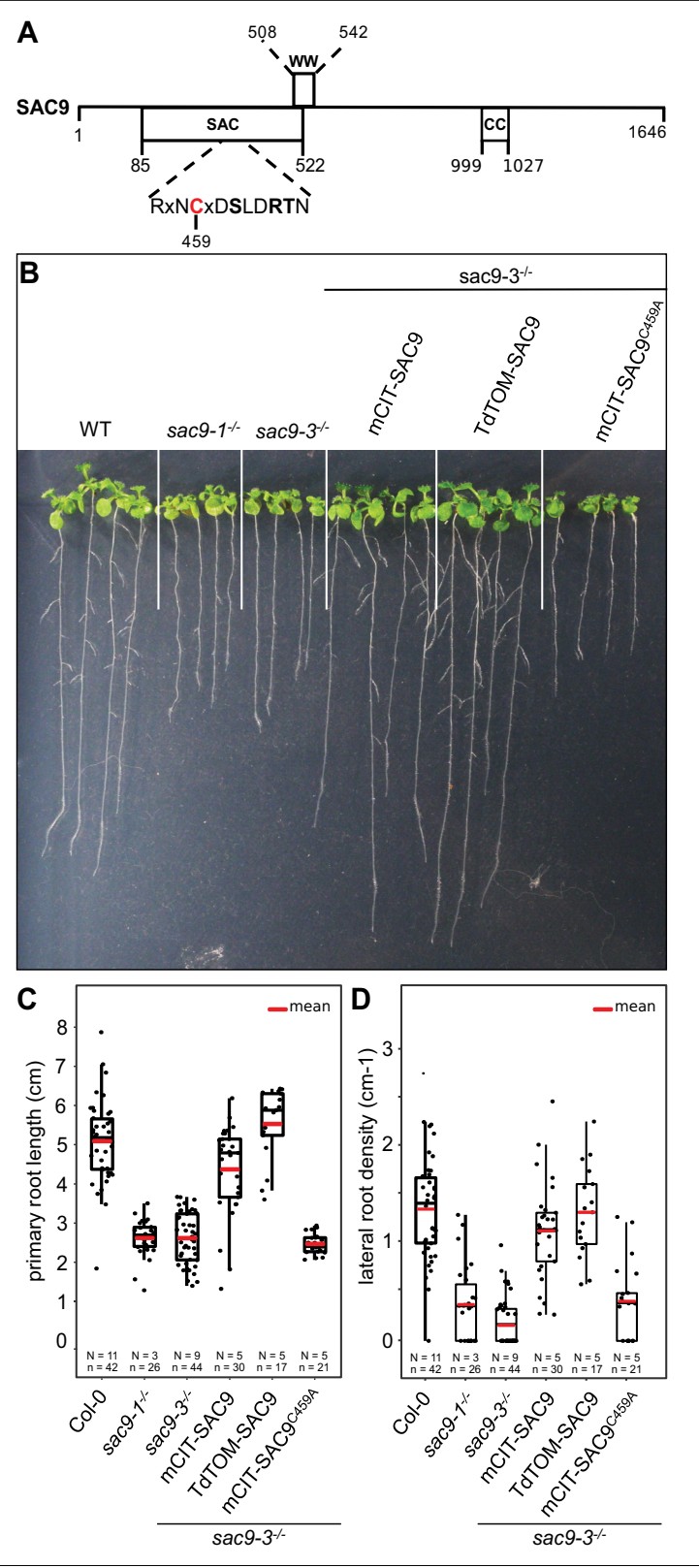

**Figure 1.** Structure-function analysis of SUPPRESSOR OF ACTIN9 (SAC9). (**A**) Schematic representation of SAC9 protein. The SAC catalytic domain, as well as the WW domain and the coil-coiled domain, are represented. (**B**) Representative images of the macroscopic phenotype observed in (**i**) wild-type (Col-0), (**ii**) *sac9-1⁻ᐟ⁻* and *sac9-3⁻ᐟ⁻* loss of function mutants, (**iii**) *sac9-3⁻ᐟ⁻* complemented lines expressing full-length genomic DNA encoding

*Figure 1 continued on next page*

*Figure 1 continued*

SAC9 fused to yellow (mCIT-SAC9, line #1000-9-1) or red (TdTOM-SAC9, line #987-5-4) fluorescent proteins and a mutated version of the putative catalytic cysteine residue within the C-x(5)-R-[TS] catalytic motif in the SAC domain (mCIT-SAC9$^{C459A}$, line #1354-12-14; right panel). Pictures are taken 12 days post germination (dpg). Note that a second independent transgenic line is presented for each construct in *Figure 1—figure supplement 1*. (**C**) Quantification of primary root length in *sac9-3*$^{-/-}$ homozygous mutants expressing mCIT-SAC9 and mCIT-SAC9$^{C459A}$ under the control of the native promoter (*SAC9prom*) and TdTOM-SAC9 under the expression of the Ubq10 promoter. Wild-type (Col-0) seedlings and two independent mutant alleles of *SAC9*, *sac9-1*$^{-/-}$, and *sac9-3*$^{-/-}$, are used as controls. (**D**) Same as (**C**) but for the quantification of the lateral root density (ratio of the number of lateral roots to primary root length). In the plots, middle horizontal bars represent the median, while the bottom and top of each box represent the 25th and 75th percentiles, respectively. At most, the whiskers extend to 1.5 times the interquartile range, excluding data beyond. For range of value under 1,5 IQR, whiskers represent the range of maximum and minimum values. Details for statistical analysis can be found in the Methods section and *Supplementary file 1C*. N=number of replicates; n=number of roots.

The online version of this article includes the following figure supplement(s) for figure 1:

**Figure supplement 1.** Complementation analysis of the sac9-3 mutant phenotype.

metabolism, such as the yeast PI 4-kinases Stt4p and Pik1p, specifically impact distinct pools of a given phosphoinositide species depending on their subcellular localization (*Roy and Levine, 2004*; *Yoshida et al., 1994*; *Flanagan et al., 1993*). Using homozygous T3 transgenic lines expressing the functional *SAC9pro:mCIT-SAC9* or *Ub10pro:TdTOM-SAC9* constructs, we assessed the subcellular distribution of SAC9 in rescued *sac9-3*$^{-/-}$ homozygous plants using live-cell fluorescence imaging. In meristematic epidermal cells of *Arabidopsis* roots, mCIT-SAC9 and TdTOM-SAC9 were mainly diffused in the cytosol and excluded from the nucleus (*Figure 2A-C*, *Figure 2—figure supplement 1A*). At the cortex of the cell, in close vicinity to the plasma membrane (Zi focal plane, *Figure 2A*; *Figure 2—figure supplement 1B*), mCIT-SAC9 localized to a cortical population of mobile dotty structures (*Figure 2B*, *Video 1*). To assess whether this discrete subcellular localization was relevant for SAC9 function, we generated transgenic lines expressing a mCIT-SAC9 variant in which the predicted coil-coiled motif was deleted, under the native promoter (*Figure 1A*, *SAC9pro:mCIT-SAC9*$^{ΔCC}$). Analyses of 24 independent T1 revealed that mCIT-SAC9$^{ΔCC}$ did not complement sac9-3$^{-/-}$ dwarf phenotype (*Figure 1—figure supplement 1A*). Similar to mCIT-SAC9$^{C459A}$, these results were confirmed on two independents homozygous T3 lines and western blot analyses showed that mCIT-SAC9$^{ΔCC}$ accumulated to similar extent as the wild-type mCit-SAC9 in each of these lines (*Figure 1—figure supplement 1B*). However, mCIT-SAC9$^{ΔCC}$ did not localize to endosome and remained entirely cytosolic (*Figure 2B*, *Figure 2—figure supplement 1C*). Thus, even though SAC9 localization in cortical intracellular compartments is discreet, and only slightly enriched compared to its cytosolic localization, it appears critical for function. Furthermore, this analysis excludes a scenario in which the mCit-SAC9-containing compartments are cytosolic densities, because they are not seen with a cytosolic mutant of SAC9.

The putative catalytically dead version, mCIT-SAC9$^{C459A}$, accumulated to a lesser extent in the cytosol compared to the native form. In addition, SAC9$^{C459A}$ showed a threefold increase in the number of labeled dotty structures compared with the signal collected for the wild-type fusion proteins mCIT-SAC9 and TdTOM-SAC9 (*Figure 2C, D*). This result suggests that the putative catalytic cysteine of SAC9 is required for the dynamic interaction of SAC9 with intracellular membranes. However, co-visualization of TdTOM-SAC9 and mCIT-SAC9$^{C459A}$ (*Figure 2E*) in Col-0 background demonstrated that all TdTOM-SAC9 dotty structures colocalized with mCIT-SAC9$^{C459A}$ at the cortex of the cell.

By comparing the signal observed at the cortex of the cell (Zi) with the signal collected at the center of the cell (Zii), we observed and quantified that mCIT-SAC9 labeled more intracellular compartments close to the plasma membrane than at a distal position, while this bias was less pronounced for FM4-64 labeled compartments (*Figure 2F-H*). Similarly, when imaging cells in their cortical part close to the plasma membrane (Zi), the mCIT-SAC9 and mCIT-SAC9$^{C459A}$ signal was concentrated in intracellular compartments at the close vicinity with the plasma membrane (*Figure 2F and I*). These results suggest a function for SAC9 in regulating phosphoinositide homeostasis either at or in the close vicinity of the plasma membrane.

We next investigated the nature of the intracellular structures labeled by mCIT-SAC9. Both mCIT-SAC9 and mCIT-SAC9$^{C459A}$ colocalized with the amphiphilic styryl dye (FM4-64) stained

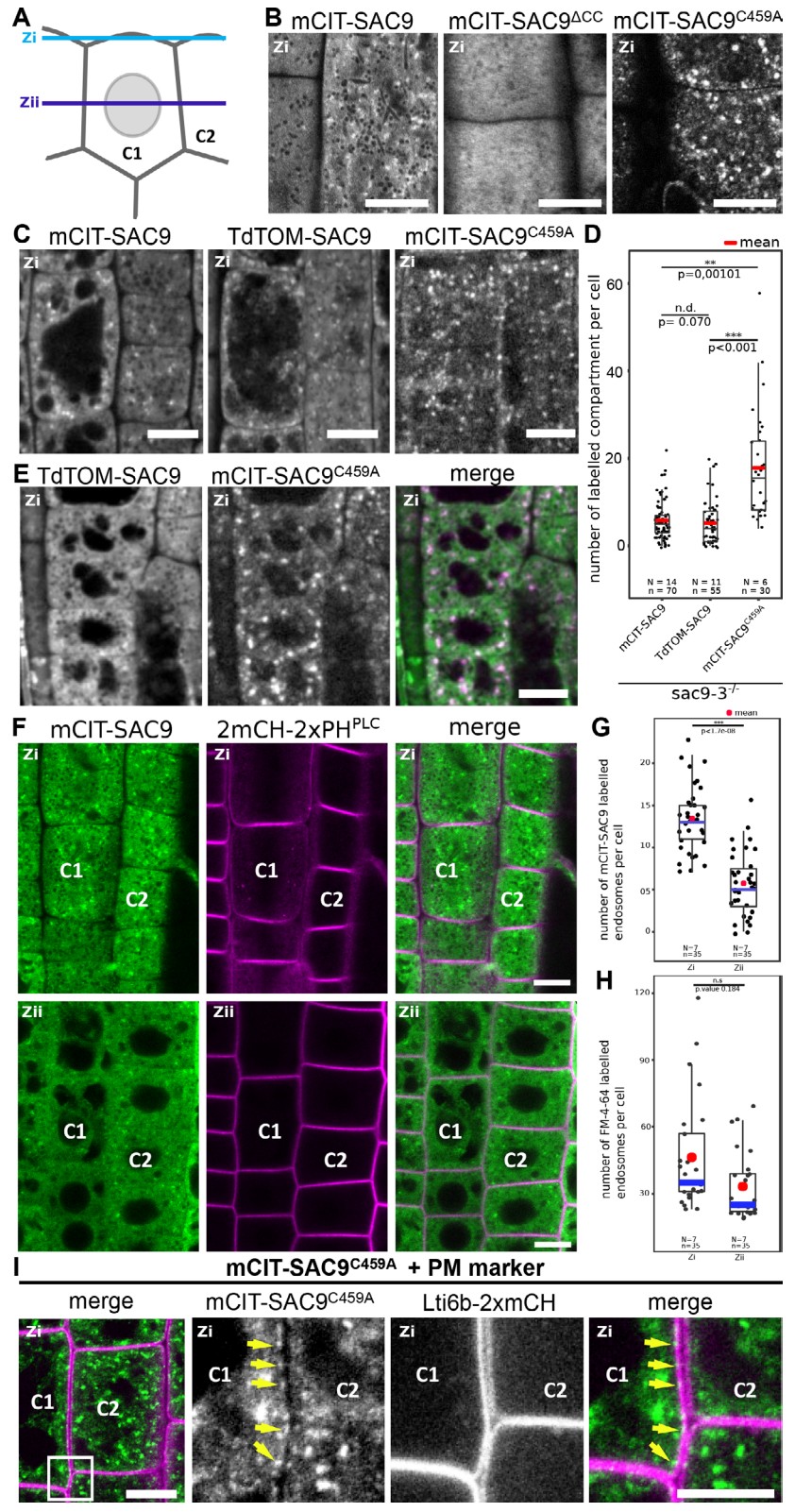

**Figure 2.** SUPPRESSOR OF ACTIN9 (SAC) localizes in the cytosol and in intracellular compartment enriched at the cell cortex. (**A**) Schematic representation of two root epidermal cells (cell #1 indicated as C1 and cell #2 indicated as C2) imaged at two different focal planes (cortical plane, close to the plasma membrane and designated as Zi and a median plane, in the middle of the cell designated as Zii). Note that the C1/C2 and Zi/Zii

*Figure 2 continued on next page*

*Figure 2 continued*

notations are used consistently according throughout the figures. (**B**) Confocal images of the subcortical part (i.e. Zi) of the *Arabidopsis* root epidermis expressing mCIT-SAC9, mCIT-SAC9$^{\Delta CC}$ mutated in the predicted coil-coiled domain, and mCIT-SAC9$^{C459A}$ mutated in its putative catalytic cysteine, under the control of SAC9 native promoter (*SAC9prom*). (**C**) Representative images of the fluorescent signal observe in *sac9-3*$^{-/-}$ mutant expressing mCIT-SAC9, TdTOM-SAC9, and mCIT-SAC9$^{C459A}$. (**D**) Comparison of the number of labeled intracellular compartments per cell in *sac9-3*$^{-/-}$ root epidermis expressing mCIT-SAC9 and TdTOM-SAC9, mCIT-SAC9$^{C459A}$. (**E**) Colocalization analysis on *Ub10pro:TdTOM-SAC9* (magenta) and *SAC9pro:CIT-SAC9*$^{C459A}$ (green). (**F**) Confocal images of the subcortical part of the *Arabidopsis* root epidermis (upper panels, Zi) or the center of the cell (lower panels, Zii) expressing *mCIT-SAC9* together with the PI(4,5)P$_2$ biosensor *2mCH-2xPH*$^{PLC}$ (note that the cells shown in Zi and Zii are the same, just on different focal plane). (**G**), Quantification of the number of endosomes labeled by mCIT-SAC9 observed per cell at two different focal planes (Zi and Zii). (**H**), Quantification of the number of endosomes labeled by FM4-64 observed per cell at two different focal planes (Zi and Zii). (**I**), Co-visualization of *SAC9pro:mCIT-SAC9*$^{C459A}$ (green) together with the plasma membrane marker Lti6b-2xmCH (magenta) in epidermal root cells taken in the Zi focal plane. The inset shows a magnification, with the yellow arrows indicating some of the intracellular compartments decorated by mCIT-SAC9$^{C459A}$ observed at the close vicinity to the labeled plasma membrane in cell C1. Scale bar: 10 µm in A-F; 5 µm in G (left panel) and 2 µm in the close-up of G (right panel). In the plot, middle horizontal bars represent the median, while the bottom and top of each box represent the 25th and 75th percentiles, respectively. At most, the whiskers extend to 1.5 times the interquartile range, excluding data beyond. For range of value under 1,5 IQR, whiskers represent the range of maximum and minimum values. Details of the statistical analysis could be found in *Supplementary file 1D*. The plane (Zi or Zii) in each image is mentioned, and the image display is representative for the plane used for the analysis. N=number of replicates; n=number of roots.

The online version of this article includes the following figure supplement(s) for figure 2:

**Figure supplement 1.** Subcellular localization of SAC9 alleles at different depths in the cell.

---

endosomal compartments at the cell's cortex, just beneath the plasma membrane, while the soluble mCIT-SAC9$^{\Delta CC}$ did not (*Figure 3A and B*, *Figure 2—figure supplement 1B, C*, *Figure 3—figure supplement 1A*). Early endosomes/TGN (EE/TGN) are sensitive to BFA while late endosomes (LE/MVB) are not (*Takagi and Uemura, 2018*). We observed that Brefeldin A (BFA) treatment led to the aggregation of mCIT-SAC9 and mCIT-SAC9$^{C459A}$ into BFA bodies (*Figure 3C, D*) suggesting that both functional and C459-mutated SAC9 fusion proteins may localize to EE/TGN compartments.

To get more precise insights into the SAC9's localization at the TGN, we crossed *Arabidopsis* lines expressing fluorescent tagged SAC9 and SAC9$^{C459A}$ proteins with endomembranes markers (*Geldner et al., 2009*). Using live-cell imaging in root meristematic cells at a Zi focal plane, we observed and quantified that mCIT-SAC9 colocalized with TGN markers (CLC2-RFP>79% colocalization; mCH-RabA1g>91%, mCH-VTI12 >67%; *Figure 3E and F*). The mCIT-SAC9 showed very weak colocalization with a late endosome (LE/MVB) marker (mCH-RabF2a<17%, *Figures 3E and 2F*), and with a Golgi marker (mCH-Got1*P*<1%, *Figure 3E and F*). Similarly, mCIT-SAC9$^{C459A}$ strongly colocalized with CLC2-RFP and mCH-RAB-A1g markers, whereas it did not colocalized with the Golgi marker (*Figure 3—figure supplement 2C*, *Figure 3G, H*). This confirmed that SAC9 and SAC9$^{C459A}$ fusion proteins localized to the TGN/EE, with the noticeable difference that SAC9 localization in endomembrane compartments was restricted to the cortex of the cell, suggesting a role of SAC9 in PI(4,5)P$_2$ homeostasis between the plasma membrane and early endosomal compartments.

## SAC9 is required to maintain the pool of PI(4,5)P$_2$ at the plasma membrane

It was previously reported that *Arabidopsis sac9-1* loss-of-function mutant had a diminution of PI4P and a threefold accumulation of PI(4,5)P$_2$, suggesting that SAC9 has a PI(4,5)P$_2$ 5-phosphatase catalytic activity (*Williams et al., 2005*). In order to visualize the effect of the perturbation of the phosphoinositide metabolism in the absence of SAC9, we introgressed two independent PI(4,5)P$_2$ biosensors (*Simon et al., 2014*) into *sac9-3*$^{-/-}$ genetic background. As previously described, both PI(4,5)P$_2$ biosensors labeled the plasma membrane and were excluded from intracellular compartments in wild-type cells (*Figure 4A and B*). In *sac9-3*$^{-/-}$, mCIT-2xPH$^{PLC}$ and mCIT-TUBBYc not only labeled the plasma membrane, but also decorated cortical intracellular dotty structures (*Figure 4A and B*). In addition to the absence of complementation observed macroscopically for *sac9-3* mutants expressing the putative catalytically-dead *SAC9*$^{C459A}$ variant (*Figure 1B and C*, *Figure 1—figure supplement 1A*),

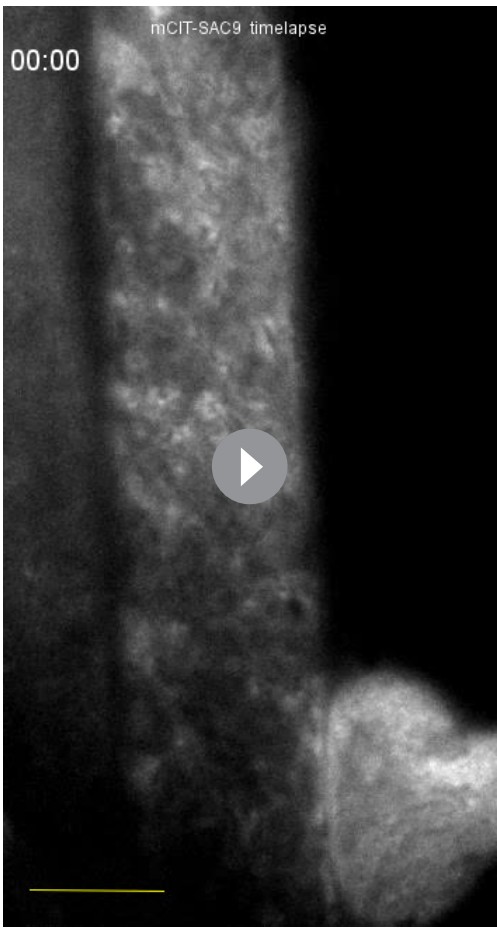

**Video 1.** Time-lapse imaging of mCIT-SAC9 using spinning disk confocal microscope (1 s per frame).
https://elifesciences.org/articles/73837/figures#video1

the same aberrant localization for the PI(4,5)$P_2$ biosensor mCIT-TUBBYc was observed in these lines (**Figure 4C**). These results demonstrate that C459 is required for the regulation of the pool of PI(4,5)$P_2$ and point out toward a role of SAC9 for the restriction of the PI(4,5)$P_2$ at the plasma membrane via its catalytic phosphatase activity. We also observed an increase in the number of intracellular compartments labeled by the PI4P biosensors (**Simon et al., 2014**; **Simon et al., 2016**) mCIT-PH$^{FAPP1}$ and mCIT-P4M$^{SidM}$, but not for mCIT-PH$^{OSBP}$ (**Figure 4A and B**). This result is consistent with a diminution of the PI4P pool at the plasma membrane and therefore the relocalization of the PI4P biosensor in intracellular compartments, as previously reported (**Simon et al., 2016**). The subcellular localization of PI3P sensors (mCIT-FYVE$^{HRS}$ and mCIT-PHOX$^{p40}$) was identical between Col-0 and sac9$^{-/-}$ cells (**Figure 4B**, **Figure 4—figure supplement 2A**). We detected a slight but significant decrease in the density of intracellular compartments decorated by mCIT-C2$^{Lact}$ phosphatidylserine biosensor in sac9-3$^{-/-}$ (**Figure 4B**; **Figure 4—figure supplement 1**). Taken together, these results indicate that SAC9-depletion leads to a massive change in PI(4,5)$P_2$ subcellular patterning, which is present on intracellular cortical structures instead of only being present at the plasma membrane.

Previous ultrastructural investigation using electron microscopy, reported a massive accumulation of vesicles, presumably containing cell wall, at the close vicinity to the plasma membrane in the sac9 mutant (**Vollmer et al., 2011**). When co-imaging 2xmCH-2xPH$^{FAPP1}$ together with mCIT-TUBBYc in sac9-3$^{-/-}$ (**Figure 4D**), the dotty structures decorated by the PI(4,5)$P_2$ biosensor were observed at the cortex of the cell (Zi), in the close vicinity of the plasma membrane (**Figure 4D** upper panel), whereas those structures were rarely observed in the internal part of the cell (**Figure 4D** lower panel). Moreover, dotty structures containing PI(4,5)$P_2$ were associated with PI4P-labeled compartments but they did not strictly overlap. Confocal imaging of sac9-3$^{-/-}$ mutant co-expressing mCIT-TUBBYc with endosomal markers revealed that the dotty structures containing PI(4,5)$P_2$ were found associated but strictly overlapping with the FM4-64 endocytic tracer as well as TGN markers (**Figure 4—figure supplement 1C**). Furthermore, BFA treatment efficiently induced 2xmCH-2xPH$^{FAPP1}$-positive BFA bodies in sac9-3$^{-/-}$ but did not affect the distribution of mCIT-TUBBYc compartments in the same cells (**Figure 4—figure supplement 1D**). Therefore, PI(4,5)$P_2$-containing intracellular compartments in sac9-3$^{-/-}$ are devoid of any ARF GTPase activated by BFA sensitive ARF-GEF, and are associated with TGN markers (**Figure 4—figure supplement 1E**). In vivo time-lapse imaging of PI(4,5)$P_2$ biosensor mCIT-TUBBYc and mCIT-2xPH$^{PLC}$ in sac9-3$^{-/-}$ mutant revealed that those intracellular structures were mobile in the cortex of root epidermal cells, hence, behaving like intracellular compartments (**Figure 4—figure supplement 1F**, **Video 2**). Moreover, in sac9-3$^{-/-}$ coexpressing the PI(4,5)$P_2$ biosensor mCIT-TUBBYc together with the non-functional SAC9pro:tdTOM-SAC9$^{C459A}$, mCIT-TUBBYc-labeled intracellular structures did not strictly colocalized but were observed at the same Z plan in close association with tdTOM-SAC9$^{C459A}$ (**Figure 4—figure supplement 1G**).

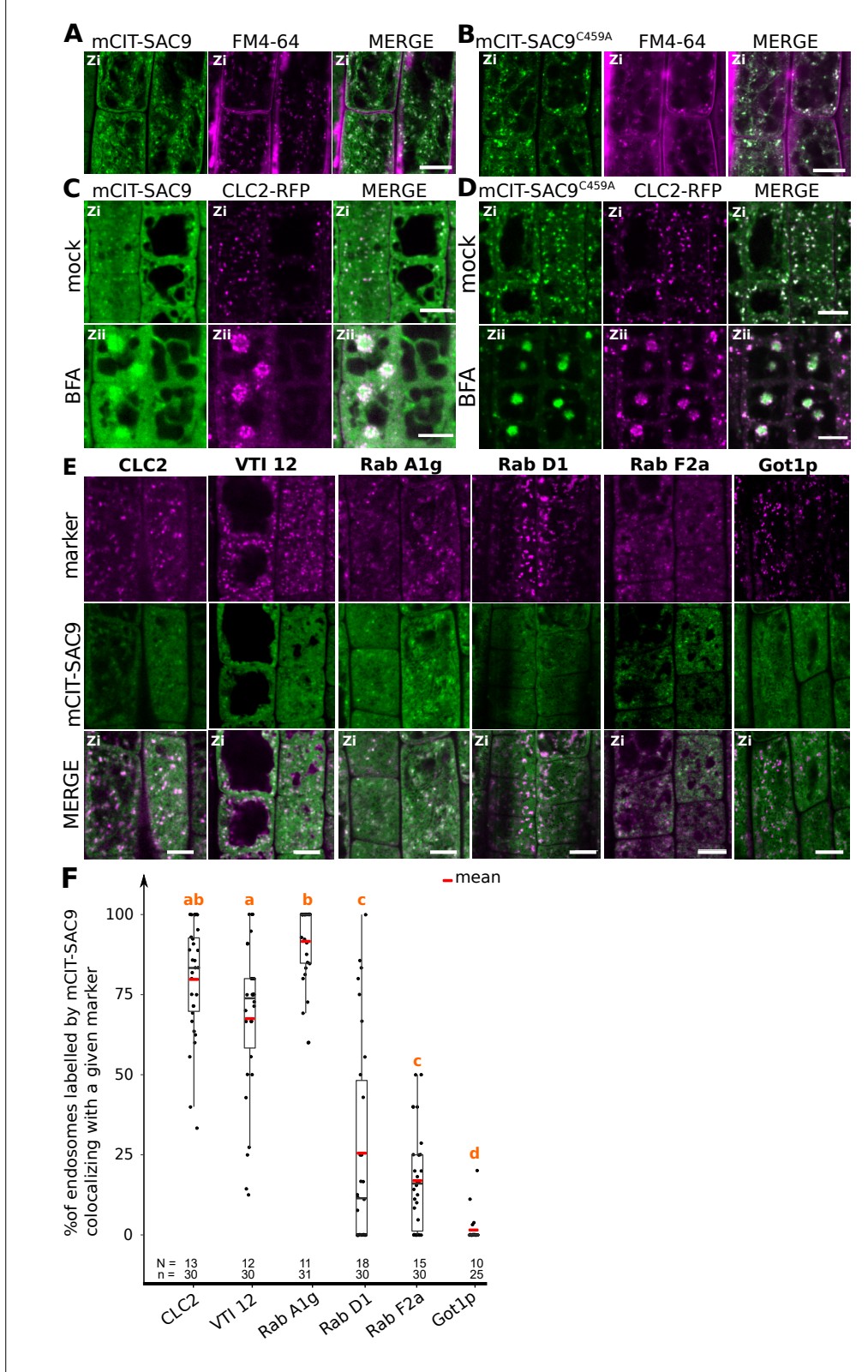

**Figure 3.** SUPPRESSOR OF ACTIN9 (SAC9) localizes to a subpopulation of TGN/EE. (**A, B**) Confocal images of *SAC9pro:mCIT-SAC9* (A, green) or *SAC9pro:mCIT- SAC9^C459A* (B, green) in vivo in *Arabidopsis* root epidermis together with the endocytic tracer FM4-64 (magenta). (**C, D**) Confocal images of *Arabidopsis* root epidermis co-expressing *SAC9pro:mCIT-SAC9* (C, green) or *SAC9pro:mCIT- SAC9^C459A* (D, green) together Clathrin Light Chain

*Figure 3 continued on next page*

*Figure 3 continued*

2 (CLC2) fused to RFP (*CLC2-RFP*; magenta). Upper panel: fluorescent signals observed in the mock treatment; Lower panel: fluorescent signals observed after 50 µM 60 min BFA treatment. (**E**) Confocal images of *Arabidopsis* root epidermis co-expressing *SAC9pro:mCIT-SAC9* (green) and TGN markers *CLC2-RFP, mCH-VTI12, mCH-RabA1g, mCH-RabD1*, the late endosome/pre-vacuolar compartment (LE/MVB) marker *mCH-RabF2a/Rha1* and the Golgi marker *mCH-Got1p* (magenta). (**F**) Percentage of colocalization between mCIT-SAC9 and a given endosomal compartment marker per cell, at zi. N=number of roots, n=number of cells. In the plots, middle horizontal bars represent the median, while the bottom and top of each box represent the 25th and 75th percentiles, respectively. At most, the whiskers extend to 1.5 times the interquartile range, excluding data beyond. For range of value under 1,5 IQR, whiskers represent the range of maximum and minimum values. The plane ($Z_i$ or $Z_{ii}$) in each image is mentioned, and the image display is representative for the plane used for the analysis. Scale bars: 10 µm.

The online version of this article includes the following figure supplement(s) for figure 3:

**Figure supplement 1.** Quantification of the number of endosomes per cell labeled by mCIT-SAC9 variants which colocalized with FM4-64 at Zi.

**Figure supplement 2.** Confocal images of SAC9C459A with endosomal markers.

We next addressed the turnover and the origin of the cortical intracellular PI(4,5)P$_2$ compartment observed in *sac9-3*[-/-]. We previously showed that short-term treatment (15–30 min) with phenyl arsine oxide (PAO), a pan PI 4-kinases inhibitor, significantly depletes PI4P (*Figure 4—figure supplement 2*) but not PI(4,5)P$_2$ pools at the plasma membrane of plant cells, whereas longer treatment (>60 min) affects the synthesis of both lipids (*Figure 4—figure supplement 2A*, *Platre et al., 2018*; *Simon et al., 2016*). We used this pharmacological approach to test the effect of the inhibition of either PI4P, or both PI4P and PI(4,5)P$_2$ synthesis, on *sac9-3*[-/-] anomalous PI(4,5)P$_2$ intracellular compartments (*Figure 4—figure supplement 2A–2E*). Solubilization of mCIT-PH$^{FAPP1-E50A}$ PI4P biosensor in *sac9-3*[-/-] cells treated for either 30 or 120 min with PAO confirmed the efficient PI4P depletion in both conditions (*Figure 4—figure supplement 2A–2E*). Solubilization of mCIT-TUBBYc PI(4,5)P$_2$ biosensor in WT after 120 min of PAO exposure, but not after 30 min, confirmed that an efficient PI(4,5)P$_2$ depletion occurred only for the longest treatment (*Figure 4—figure supplement 2A–2E*). 30 min PAO treatment did not affect anomalous *sac9-3*[-/-] PI(4,5)P$_2$ compartments, but 120 min PAO treatment significantly reduced the number of anomalous PI(4,5)P$_2$ compartments compared to both 120 min mock treatments or 30 min short treatment (*Figure 4—figure supplement 2A–2E*), showing that intracellular PI(4,5)P$_2$ compartments in *sac9-3*[-/-] are dependent on the PI4P synthesis, itself substrate for PI(4,5)P$_2$ production.

## SAC9 is required for efficient endocytic trafficking

Because of the specific localization of SAC9 at the cortex of the cell and its colocalization with TGN/EE markers, we wondered whether PI(4,5)P$_2$ defective patterning in *sac9-3*[-/-] correlates with endocytic defects. We counted the numbers of labeled endosomes following FM4-64 endocytic tracer treatment in cells from WT and *sac9-3*[-/-] seedlings (*Rigal et al., 2015*). We observed a significant near twofold decrease in the number of FM4-64-labeled endosomes per cells in *sac9-3*[-/-] compared to the WT (*Figure 5A and B*, *Figure 5—figure supplement 1A*), which was not caused by smaller cells in *sac9-3*[-/-] as the density of FM4-64-labeled endosomes was also strongly and significantly decreased compared to the WT (*Figure 5—figure supplement 1*). We inhibited recycling with BFA, and used FM4-64 tracer to monitor the endocytic trafficking by measuring the number of BFA bodies labeled by FM4-64 in Col-0 and *sac9-3*[-/-]. We observed significantly less FM4-64-labeled BFA bodies per cells in *sac9-3*[-/-] compared to the WT (*Figure 5C and D*, *Figure 5—figure supplement 1B*), confirming the lower rate of endocytic trafficking in this mutant. We then assessed whether SAC9 depletion affected the trafficking of cargo proteins. We, therefore, performed another BFA assay, but using the integral membrane protein PIN-FORMED2 fused with GFP (PIN2-GFP) which localizes at the plasma membrane and in intracellular organelles as it continuously recycles (*Armengot et al., 2016*). We observed significantly less PIN2-GFP-labeled BFA bodies per cell in *sac9-3*[-/-] compared to WT (*Figure 5E and F*). PIN2-GFP being partially located on intracellular organelles before BFA treatment, the effect observed may, therefore, indicate an endocytic defect as supported by the others experiments, and/or a more general defect in trafficking in *sac9-3*[-/-].

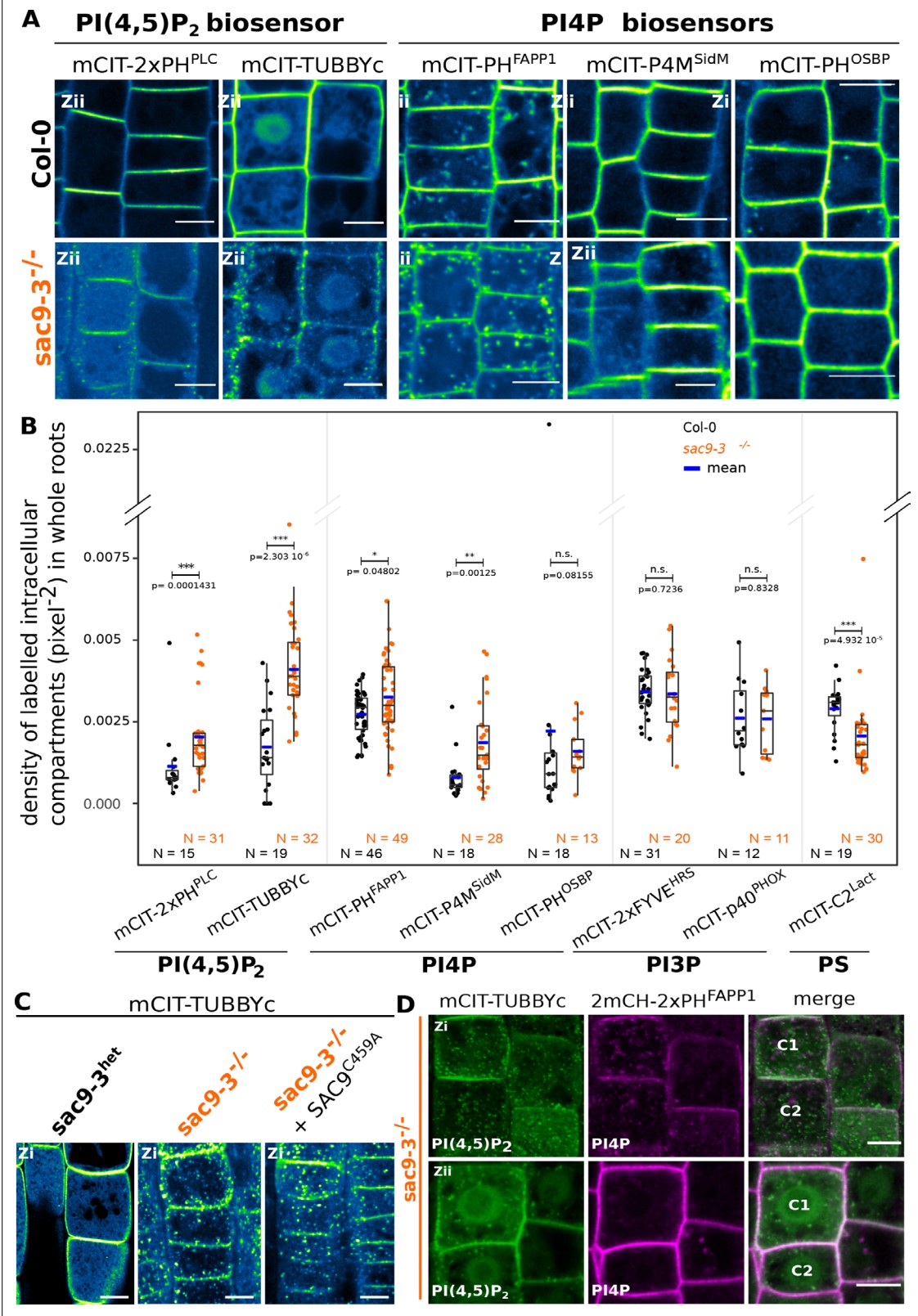

**Figure 4.** SUPPRESSOR OF ACTIN9 (SAC9) restricts PI(4,5)P$_2$ at the plasma membrane. (**A**) Confocal images of *Arabidopsis* root epidermis expressing mCIT-tagged sensors in WT (Col-0) and *sac9-3*$^{-/-}$ genetic backgrounds. Fluorescence intensity is color-coded (green fire blue scale). (**B**) Quantification of density of labeled intracellular compartments (pixel$^{-2}$) in whole roots epidermis expressing mCIT-tagged lipid sensors in wild-type (Col-0) and *sac9-3*$^{-/-}$. Non-parametric Wilcoxon rank sum tests with Bonferroni correction. In the plots, middle horizontal bars represent the median, while the bottom and

*Figure 4 continued on next page*

*Figure 4 continued*
top of each box represent the 25th and 75th percentiles, respectively. At most, the whiskers extend to 1.5 times the interquartile range, excluding data beyond. For range of value under 1,5 IQR, whiskers represent the range of maximum and minimum values. Details for statistical analysis can be found in the Methods section and *Supplementary file 1*. N=number of replicates; n=number of roots. (**C**) Confocal images of mCIT-TUBBYc in root epidermal cell of *sac9-3* heterozygous, *sac9-3⁻/⁻* (left panel) or *sac9-3⁻/⁻* expressing *TdTOM-SAC9^C459A* (right panel); N=3 replicates, n=47 roots. (**D**) Confocal images of *sac9-3⁻/⁻ Arabidopsis* root epidermal cell (**C1 and C2**) expressing *mCIT-TUBBYc* (green) and *2xmCH-PH^FAPP1* (magenta) at their cortex (upper panel, Zi) and at their center (bottom panel, Zii); N=3 biological replicates, n=49 roots. Scale bar: 10 µm. The plane (Zi or Zii) in each image is mentioned, and the image display is representative for the plane used for the analysis. Note that in panel D, the same cells are shown in the upper and lower panel, but at different focal planes.

The online version of this article includes the following figure supplement(s) for figure 4:

**Figure supplement 1.** Confocal images of Arabidopsis root epidermis expressing various phosphoinositide biosensors in WT and sac9-3-/-.

**Figure supplement 2.** The intracellular accumulation of mCITRINE-TUBBYc observed in *sac9* is sensitive to PI-4 kinase inhibition.

To gain further insights into the function of SAC9 in the endocytic trafficking, we investigated the sensitivity of the *sac9* mutant to pharmacological inhibition of endocytosis. To this end, we used the recently described ES9-17, which is a specific inhibitor of clathrin-mediated endocytosis (*Dejonghe et al., 2019*). We treated wild-type and *sac9-3⁻/⁻* seedlings with ES9-17 for 180 min and labeled the plasma membrane and endosomes with FM4-64. We observed dome-shaped plasma membrane invaginations in ES9-17 long-term treatment of Col-0 seedlings, almost exclusively in elongating or differentiated cells (epidermal or root cap cells), substantiating the possibility that these invaginations constitute read-outs of long-term disturb endocytic trafficking (*Figure 5G and H*, *Doumane et al., 2021*). Strikingly, we observed a much higher number of dome-shaped plasma membrane invagination decorated by the PI(4,5)P$_2$ biosensors mCIT-TUBBYc in cells from ES9-17 treated *sac9-3⁻/⁻* (*Figure 5I*), showing that SAC9 depletion causes over-sensitivity to inhibition of endocytic trafficking. Hypersensitivity to endocytic trafficking inhibition, decreased internalization of the bulk endocytic tracer FM4-64 and defects in PIN2 protein trafficking together indicate that endocytic trafficking is impacted in the absence of SAC9.

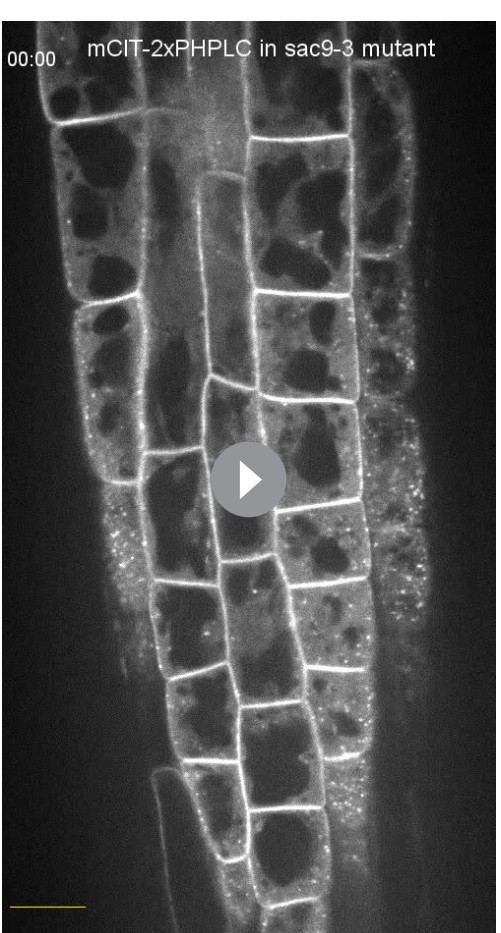

**Video 2.** Time-lapse imaging of mCIT-2xPH^PLC in *sac9* mutant (1 s per frame) using spinning disk confocal microscope.
https://elifesciences.org/articles/73837/figures#video2

## SAC9 is required for SH3P2 localization at the plasma membrane

To gain some insights into SAC9 function, we next screened for SAC9 interactors using a yeast two-hybrid assay against a universal *Arabidopsis* normalized dT library (ULTImate Y2H SCREEN, hybrigenics). As a bait, we used a portion of SAC9 (AA 499–966), which includes the WW domain, a putative protein-protein interaction platform (*Figure 6A*). Out of 59.6 million screened interaction, we recovered and sequenced 260 independent clones corresponding to 107 different proteins. However, among these proteins, most were classified as proven artifact or low confidence interaction and only three candidates (2.8%) were ranked as very high confidence in the interaction (*Supplementary file 1B*). Among the high confident interaction candidate, we identified 11 clones corresponding to SH3P2 (five

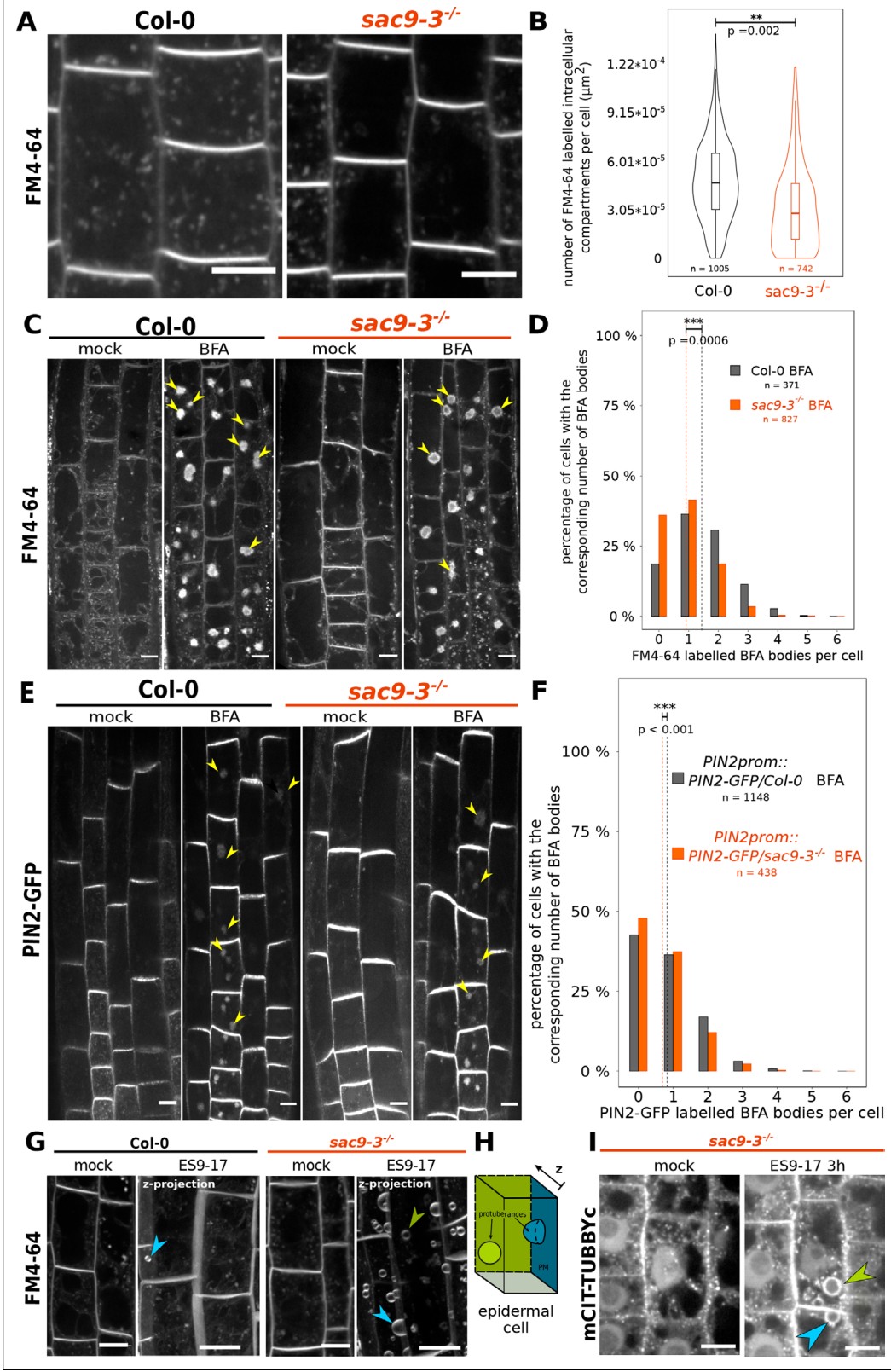

**Figure 5.** The endocytic flux is perturbed in the *sac9* mutant. (**A**) Representative images of Col-0 and *sac9-3*⁻/⁻ seedlings treated for 30 min with FM4-64, which is endocytosed and labels endocytic intracellular compartments. Scale bars: 10 μm. (**B**) Quantification from the experiment shown in (**A**). Violin and box plots quantifying the number of FM4-64 labeled intracellular compartments in Col-0 and *sac9-3*⁻/⁻. (**C**) Representative images of root

*Figure 5 continued on next page*

*Figure 5 continued*

epidermis following BFA and FM4-64 treatment of Col-0 and *sac9-3⁻/⁻* seedlings. Examples of FM4-64 labeled BFA bodies are pointed out (green arrowheads). (**D**) Quantification from the experiment shown in (**C**). For Col-0 and *sac9-3⁻/⁻*, the proportion (%) of cells containing from none to six BFA bodies is displayed. Dotted line: means. (**E**) Representative images following BFA treatment of PIN2-GFP/WT and PIN2-GFP/*sac9-3⁻/⁻* seedlings. Examples of PIN2-GFP labeled BFA bodies are pointed out (green arrowheads). Scale bars: 10 µm. (**F**) Quantification from the experiment shown in (**E**). For *PIN2-GFP*/WT and PIN2-GFP/*sac9-3⁻/⁻*, the proportion (%) of cells containing from none to six BFA bodies is displayed. Dotted line: means. (**G**) Over-sensitivity of *sac9-3⁻/⁻* to prolonged inhibition of endocytosis. Seedlings were treated 180 min with 30 µM ES9-17 or DMSO (mock), FM4-64 being added after 30 min (150 min of exposure). The picture shown after the ES9-17 treatment are the results of the projection of a z-stack. (**H**) ES9-17 treatments led to dome-shaped plasma membrane invagination. blue arrowheads: invaginations with an obvious connection to the plasma membrane; green arrowheads: invaginations without a clear connection to the plasma membrane (often connected to medullar plasma membrane). N=2 biological replicates, n=8 roots. (**I**) In *sac9-3* mutant, ES9-17 treatments led to dome-shaped plasma membrane invagination labeled by the PI(4,5)P$_2$ biosensor mCIT-TUBBYc; N=2 biological replicates, n=10 roots. Blue arrowheads: invaginations with an obvious connection to the plasma membrane; green arrowheads: invaginations without a clear connection to the plasma membrane (often connected to medullar plasma membrane). Scale bars: 10 µm. Details of the statistical analysis could be found in *Supplementary file 1C*. N=3 biological replicates, n=number of cells.

The online version of this article includes the following figure supplement(s) for figure 5:

**Figure supplement 1.** Example of FM4-64 labeling in Col-0 and *sac9-3* mutant.

---

independent clones). In the yeast-two-hybrid screen, the selected interaction domain identified for SH3P2, corresponds to the C-terminal part of the proteins (aa 213–368), which includes the SH3 domain (*Figure 6A*). We decided to focus on this candidate because, like SAC9, SH3P2 is linked to both clathrin-mediated endocytosis and membrane phosphoinositides. Indeed, SH3P2 (i) colocalizes with clathrin light chain, (ii) cofractionates with clathrin-coated vesicles (*Nagel et al., 2017*), and (iii) co-immunoprecipitates in planta with clathrin heavy chain (*Nagel et al., 2017*), clathrin light chain (*Adamowski et al., 2018*), and the dynamin DRP1A (*Ahn et al., 2017*). Furthermore, SH3P2 binds to various phosphoinositides in vitro (*Zhuang et al., 2013*; *Ahn et al., 2017*), and its plasma membrane localization is dependent on PI(4,5)P$_2$ in vivo (*Doumane et al., 2021*).

In an attempt to validate the reverse interaction, SH3P2-GFP was transiently expressed in *N. benthamiana*, then immunoprecipitation followed by mass spectrometry analysis (IP-MS) was done. Among the top interactors were two SAC9 protein isoforms in *N. benthamiana*, as determined by log fold-change and p-value in peptide count when compared to GFP control (*Figure 6B–C*). As SH3P2 was previously shown to be degraded via the proteasome (*Leong et al., 2022*), Bortezomib (BTZ) treatment was used to inhibit SH3P2 degradation in planta, and we observed a corresponding increase in the log fold change and p-value of SAC9 peptide counts compared to control (GFP/BTZ). Notably, AUXILIN-LIKE1/2 which are already described protein partners of SH3P2 (*Adamowski et al., 2018*), were found as among the top hits in both conditions, either with or without BTZ, thereby validating the approach (*Figure 6B–C*).

Using live cell imaging we observed and quantified that fluorescently tagged SAC9 and SAC9$^{C459A}$ colocalized with SH3P2 in a subcortical population of endomembrane compartments (*Figure 6B and C*), probably TGN/EEs (*Figure 3E and F*). Because SH3P2 was described to play a role in different steps of membrane trafficking, from the endocytosis at the plasma membrane to autophagy, we next addressed more precisely which pool of SH3P2 was affected in the absence of SAC9. Using confocal imaging, we observed a diminution of the signal corresponding to SH3P2-sGFP at the plasma membrane compared to the cytoplasm in *sac9-3*, while the amount of SH3P2-sGFP detected via western blot was similar between the two genotypes (*Figure 6D-F*, *Figure 6—figure supplement 1*). Moreover, in the absence of functional *SAC9*, SH3P2-TdTOM accumulates in a structure closely associated with the aberrant PI(4,5)P$_2$ pool decorated with mCIT-TUBBYc (*Figure 6H*). Altogether, the trafficking defects observed in the *sac9* mutant, the impaired localization of SH3P2 in the absence of a functional SAC9 and the formation of a SAC9-SH3P2 complex all point toward a role of SAC9 in endocytic trafficking.

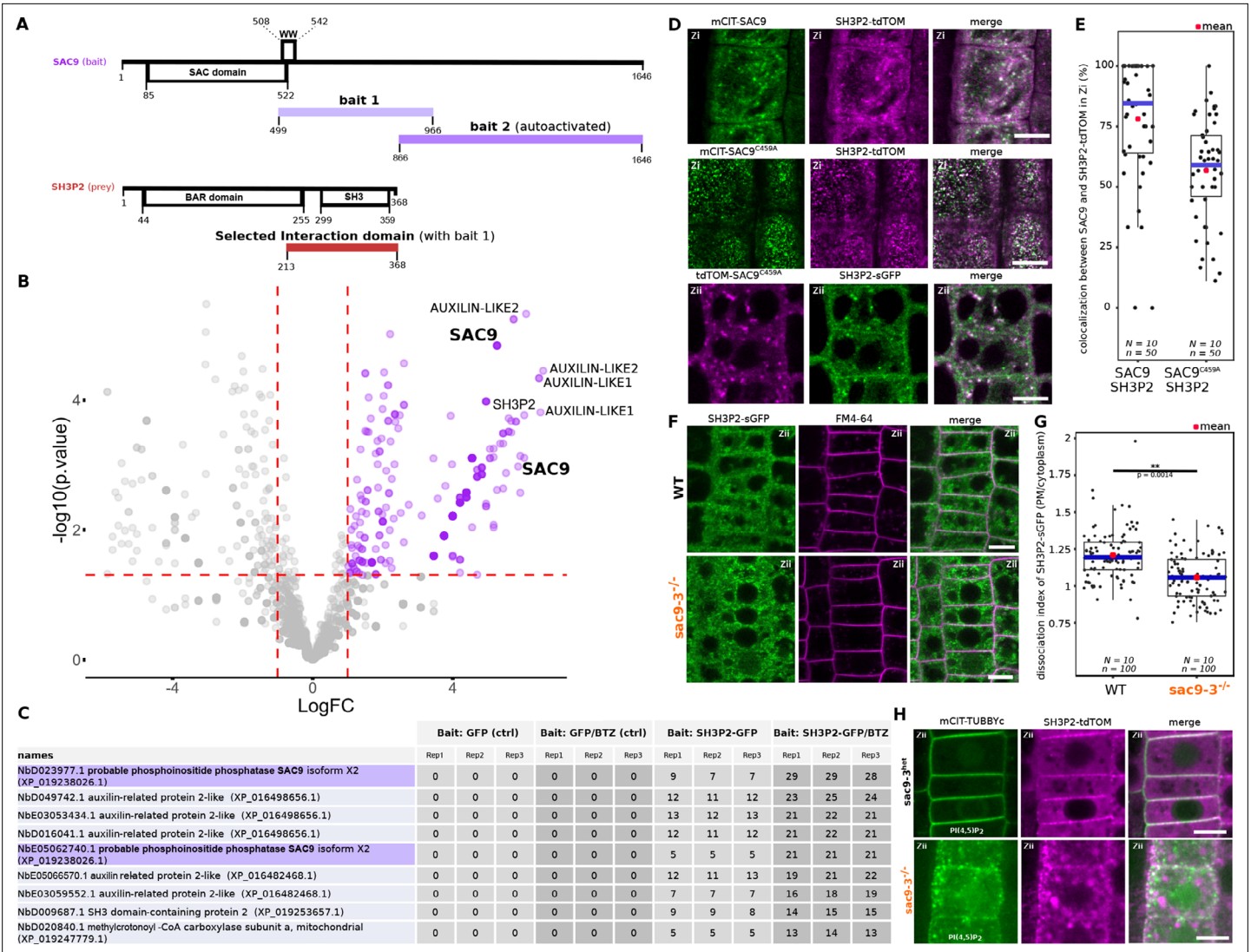

**Figure 6.** SUPPRESSOR OF ACTIN9 (SAC9) interact and colocalize with SH3P2 and regulates it subcellular distribution. (**A**) Schematic representation of the yeast two-hybrid screen using SAC9 as a bait, where SH3P2 was found as a protein partner. The selected interaction domain (interacting with bait 1) corresponds to the amino acid sequence shared by all the eleven prey fragments matching the same reference protein. Note that one other fragment of SAC9 was intended to be screened for interacting proteins (bait 2), but the construct was autoactive in yeast. For design purpose, the scale between SAC9 and SH3P2 was not respected. (**B**) Volcano plot of SH3P2-GFP IP-MS after transient expression in *N. benthamiana*. X-axis displays the log fold-change (log(FC)) of proteins in SH3P2-GFP compared to the control (GFP). The Y-axis shows the statistical significance, log10(p-value). Dashed lines represent threshold for log(FC)>1 and p-value <0.05. (**C**) Table showing the peptide counts of the top hits obtained in the IP-MS (n=3 independent pooled biological replicates), including the controls GFP and GFP/BTZ. Note that multiple protein isoforms of SAC9 are present in the *N. benthamiana* annotated proteome (*Kourelis et al., 2019b*), and two were found in the IP-MS as SAC9 interactors. (**D**), Representatives images of the colocalization between mCIT-SAC9 and SH3P2-tdTOM in a proximal (Zi) or distal (Zii) region from the plasma membrane of *Arabidopsis* epidermal root cells. Scale = 5 µm. (**E**) Quantification of the number of SAC9-labeled endosomes colocalizing with SH3P2 signal per cell. N=number of replicates; n=number of roots. (**F**), Representatives images of the SH3P2-sGFP localization (green) in WT and *sac9-3* mutant in which the plasma membrane was labeled using FM4-64 (magenta). (**G**) Quantification of the dissociation index of SH3P2-sGFP in WT and *sac9-3* mutant in which the plasma membrane was labeled using FM4-64. N=number of replicates; n=number of roots. (**H**), Representative images of the localization of *SH3P2pro:SH3P2-tdTOM* transformed in *sac9-3* heterozygous plant expressing the PI(4,5)P2 biosensor *mCIT-TUBBYc*. N=2 biological replicates, n=50 roots. In the plots, middle horizontal bars represent the median, while the bottom and top of each box represent the 25th and 75th percentiles, respectively. At most, the whiskers extend to 1.5 times the interquartile range, excluding data beyond. For range of value under 1,5 IQR, whiskers represent the range of maximum and minimum values. Details of the statistical analysis could be found in *Supplementary file 1H*.

The online version of this article includes the following figure supplement(s) for figure 6:

**Figure supplement 1.** Western blot analysis of TPLATE-GFP and SH3P2-GFP in WT and *sac9-3*[-/-].

## The endocytic protein TPLATE has an altered dynamic in the absence of SAC9

SAC9 is observed at the close vicinity to the plasma membrane and its absence causes a mislocalization of its protein partner SH3P2, as well as a reduction of the overall endocytosis process. We then assessed if the clathrin-mediated endocytosis at the plasma membrane was affected in the absence of SAC9. Using Total Internal Reflection Fluorescence (TIRF) microscopy, we determined the density and dynamic behavior of the endocytic protein from the TPLATE complex, TPLATE-GFP, in the *sac9* mutant compared with wild-type plants. Quantitative analysis revealed that the density of TPLATE-GFP was reduced in sac9-3$^{-/-}$ compared with the WT while the amount of TPLATE-GFP detected via western blot was similar between the two genotypes (*Figure 7A and B*; *Figure 6—figure supplement 1*; *Video 3*). Analysis of the dynamics of TPLATE-GFP at the plasma membrane of etiolated hypocotyl revealed a decrease in the dwell time of TPLATE-GFP at the plasma membrane in the *sac9* mutant compared to the WT (*Figure 7C and D*). This result is in line with a reduction of the endocytic flow in the absence of SAC9, and suggests that maintaining a strict plasma membrane accumulation of PI(4,5)P$_2$ is critical for clathrin-mediated endocytosis.

## Discussion

Here, we showed that SAC9 putative catalytic cysteine is required to complement the *sac9* phenotype, suggesting that SAC9 phosphoinositide phosphatase activity is key for its function. Fluorescent SAC9 protein fusions colocalize with TGN/EE markers in a subpopulation of endosomes close to the plasma membrane. We found that the subcellular patterning of PI(4,5)P$_2$ is defective in *sac9* mutants, consistent with the idea that SAC9 is a PI(4,5)P$_2$ 5-phosphatase. In planta, SAC9 interacts and colocalizes with the endocytic trafficking component SH3P2. In the absence of SAC9, and therefore when the patterning of the PI(4,5)P$_2$ at the plasma membrane and at its close vicinity is affected, SH3P2 localization at the plasma membrane is decreased and the rate of clathrin-mediated endocytosis is significantly reduced. Together, these findings underlie the importance of strictly restricting PI(4,5)P$_2$ to the plasma membrane during the endocytic process.

### SAC9 and phosphoinositide interconversion along the endocytic pathway

In animal cells, a phosphoinositide conversion cascade has been described through the successive action of phosphoinositide kinases and phosphatases. This cascade starts with PI(4,5)P$_2$ at the plasma membrane and ends-up with PI3P in the membrane of early endosomes (*Noack and Jaillais, 2017*; *Schmid and Mettlen, 2013*; *Abad et al., 2017*; *Posor et al., 2015*; *Posor et al., 2013*; *Shin et al., 2005*). Briefly, PI(4,5)P$_2$ is dephosphorylated at the plasma membrane or during the endocytic process by 5-phosphatase enzymes such as OCRL and synaptojanin (*De Matteis et al., 2017*; *Cauvin et al., 2016*; *Nández et al., 2014*; *Ben El Kadhi et al., 2011*; *Posor et al., 2013*). PI4P is then phosphorylated into PI(3,4)P$_2$ inside clathrin-coated pits by a PI3-kinase (PI3K C2α), before being converted into PI3P through the action of a PI4P phosphatase on clathrin-coated vesicles (*Schmid and Mettlen, 2013*; *Posor et al., 2013*). Because both PI(4,5)P$_2$ and PI3P are essential regulators of endocytic trafficking, this conversion cascade, and in particular the precise recruitment of dedicated enzymes at the right place and at the right time during endocytic trafficking, is of critical importance for clathrin-mediated endocytosis to proceed normally (*Noack and Jaillais, 2017*; *Schmid and Mettlen, 2013*).

The plants TGN collects endocytic vesicles and redirects cargo proteins either to the plasma membrane for recycling or to late endosomes/vacuole for degradation (*Dettmer et al., 2006*; *Narasimhan et al., 2020*; *Rodriguez-Furlan et al., 2019*). The plant TGN/EE is enriched in PI4P, not PI3P, which instead accumulates in late endosomes (*Noack and Jaillais, 2020a*; *Simon et al., 2014*). Interestingly, we found that (i) SAC9 localizes to clathrin-coated compartments close to the plasma membrane and (ii) the *sac9* mutant accumulates PI(4,5)P$_2$ in vesicular structures at the cortex of the cells. Together, we propose that SAC9 represents the long sought-after enzyme that performs the PI(4,5)P$_2$-to-PI4P conversion during the plant endocytic process (*Figure 8*). As such, SAC9 is required to erase PI(4,5)P$_2$ in endosomal membranes and thereby maintains this lipid strictly at the plasma membrane.

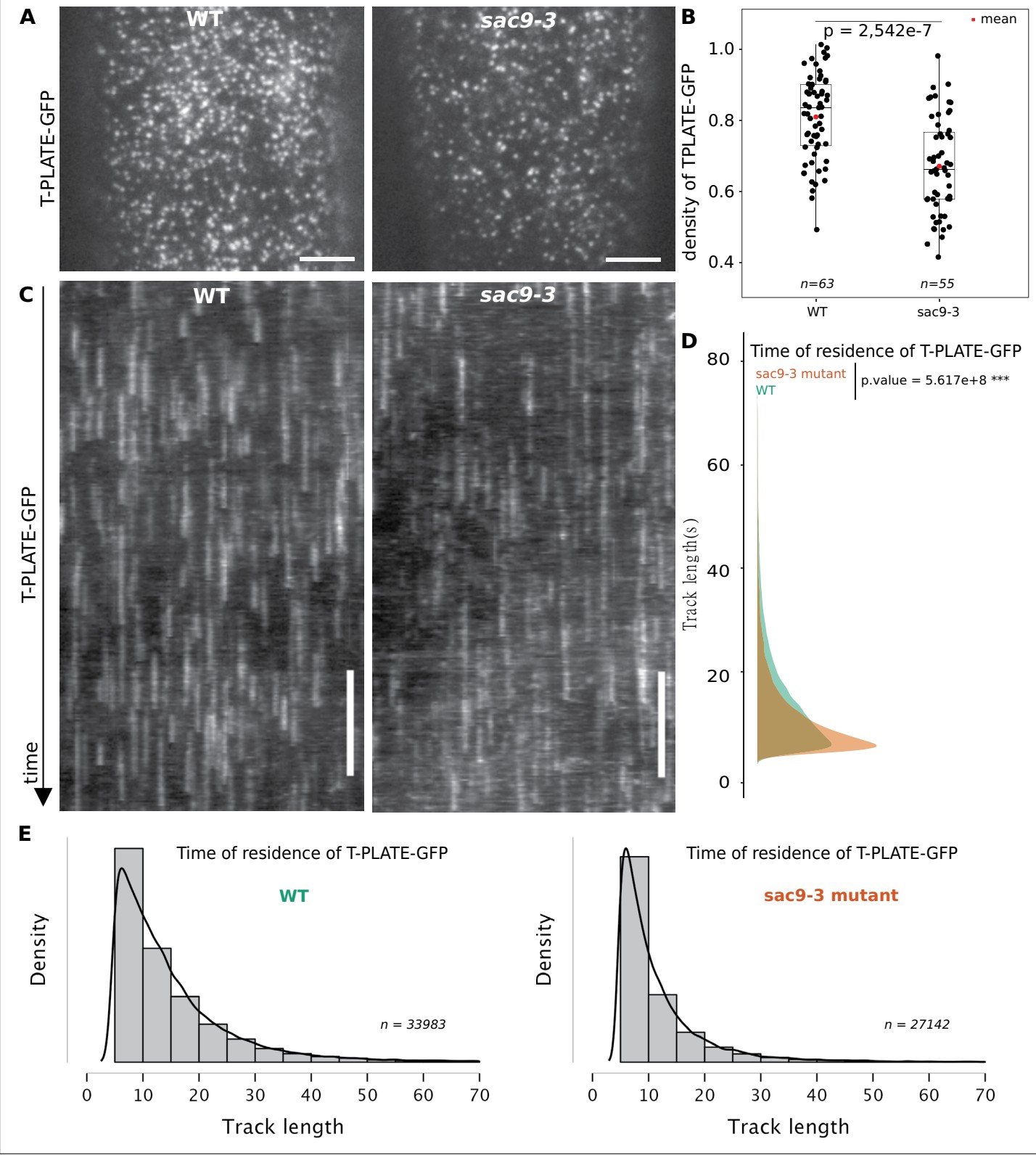

**Figure 7.** TPLATE dynamics and density at the plasma membrane is disturbed in the *sac9* mutant. (**A**) Representatives images of the TPLATE-GFP localization in WT and *sac9-3* mutant at the plasma membrane observed by TIRF microscopy in etiolated hypocotyl. Scale = 5 μm. (**B**) Quantification of the density at the plasma membrane of TPLATE-GFP in WT and *sac9-3* mutant observed by TIRF microscopy in etiolated hypocotyl. Only one of the three replicates are represented. In the plot, middle horizontal bars represent the median, while the bottom and top of each box represent the 25th

*Figure 7 continued on next page*

*Figure 7 continued*

and 75th percentiles, respectively. At most, the whiskers extend to 1.5 times the interquartile range, excluding data beyond. For range of value under 1,5 IQR, whiskers represent the range of maximum and minimum values. N=13 plants n=63 cells for the WT; N=11, n=55 for *sac9-3*. (C) Representatives kymograph of the TPLATE-GFP dynamics at the plasma membrane in WT and *sac9-3* mutant observed by TIRF microscopy in etiolated hypocotyl over 5 min; scale = 60 s. (D), Frequency distribution of the lifetimes of TPLATE-GFP tracks in WT (green) and *sac9-3* mutant (oranges). Results from an independent samples T-Test followed by Mann-Whitney U test is presented (see *Supplementary file 1L* for details). (E), Histograms of median normalized fluorescence of TPLATE-GFP in WT (left) and *sac9-3* mutant (right) representing the density of tracks per track length. Details of the statistical analysis could be found in *Supplementary file 1J*. For WT, 21 cells from N=10 plants were used, n=33,983 tracks; For *sac9*, N=13 n=27,142 tracks.

SAC9 being a putative PI(4,5)P$_2$ phosphatase, we could have expected that SAC9 overexpression may have induced gain-of-function phenotypes caused by ectopic PI(4,5)P$_2$ dephosphorylation. However, transgenic lines that express SAC9 under the expression of the *UBQ10* promoter did not show any obvious phenotypes, and instead complemented the *sac9* loss-of-function phenotype like the endogenous promoter. This absence of overexpression phenotypes may be explained by several hypotheses. First, it is possible that SAC9 is not sufficiently overexpressed when under the control of the *UBQ10* promoter compared to the endogenous SAC9 promoter. Second, it is possible that SAC9 overexpressing lines compensate for the excess SAC9 activity by adjusting their PI4P and PI(4,5)P$_2$ balance. However, our data point toward a third possibility. Most lipid phosphatase studied to date require membrane association to dephosphorylate their cognate phosphoinositide, while they do not act on their target lipid when they are in the cytosol. We found that SAC9 strongly accumulates in the cytoplasm when it is expressed under the control of either promoter and in particular when it is overexpressed. Thus, SAC9 membrane association, rather than its expression level, is likely to be the limiting factor preventing SAC9 gain-of-function phenotype in over-expression lines.

## Accumulation of PI(4,5)P$_2$-containing vesicles in the *sac9* mutant

In the absence of SAC9, PI(4,5)P$_2$ accumulates inside the cell in what appears to be abnormal compartments that are associated with but independent from TGN/EEs. We can speculate that the prolonged accumulation of PI(4,5)P$_2$ on clathrin-coated compartments, and perhaps the lack of PI4P production on these structures, impairs the function of these vesicles. We also observed that PI4P biosensors accumulate in TGNs in the *sac9* mutants. We previously found that PI4P biosensors relocalize to PI4P-containing TGNs upon specific depletion of PI4P at the plasma membrane (*Simon et al., 2016*; *Platre et al., 2018*). PI4P being the putative product of SAC9 (*Williams et al., 2005*), it is possible that there is a reduced amount of PI4P at the plasma membrane in this mutant. In that case, the balance of PI4P between the plasma membrane and the TGN would be affected between these two compartments, which could explain the relocalization of the PI4P sensors to TGN compartments.

Ultrastructural analyses previously revealed that *sac9* massively accumulates vesicles, presumably containing cell wall precursors, in close proximity to the irregular cell wall deposition (*Vollmer et al., 2011*). The finding of these unknown subcortical compartments in *sac9* resonates with our confocal microscopy observations of ectopic PI(4,5)P$_2$-containing vesicles in this mutant. Importantly, we found that these vesicles are not extensively labeled after a short-term FM4-64 treatment. This suggests that these vesicles are accumulating slowly over time in the *sac9* mutants and are not actively exchanging membrane materials with the TGN/EEs or the plasma membrane, at least not in the 30 min time frame of the FM4-64 pulse labeling experiment. Moreover, recent proteomic characterization of isolated *Arabidopsis* clathrin-coated vesicles identified SAC9 as one of the

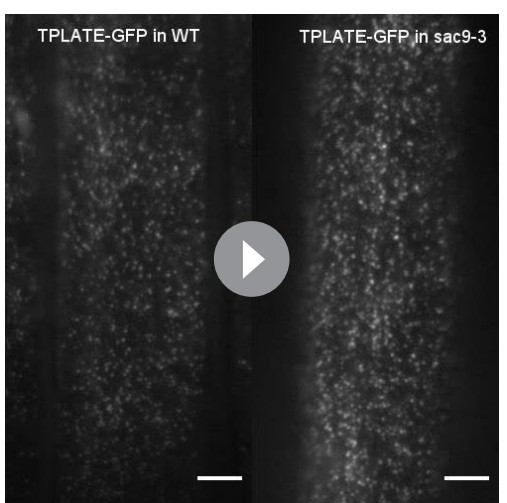

**Video 3.** Time-lapse imaging of TPLATE-GFP at the plasma membrane of WT and in *sac9-3* mutant using TIRF microscopy. Time lapses were acquired during 300 time-points for 300 s (acquisition time 500 ms).
https://elifesciences.org/articles/73837/figures#video3

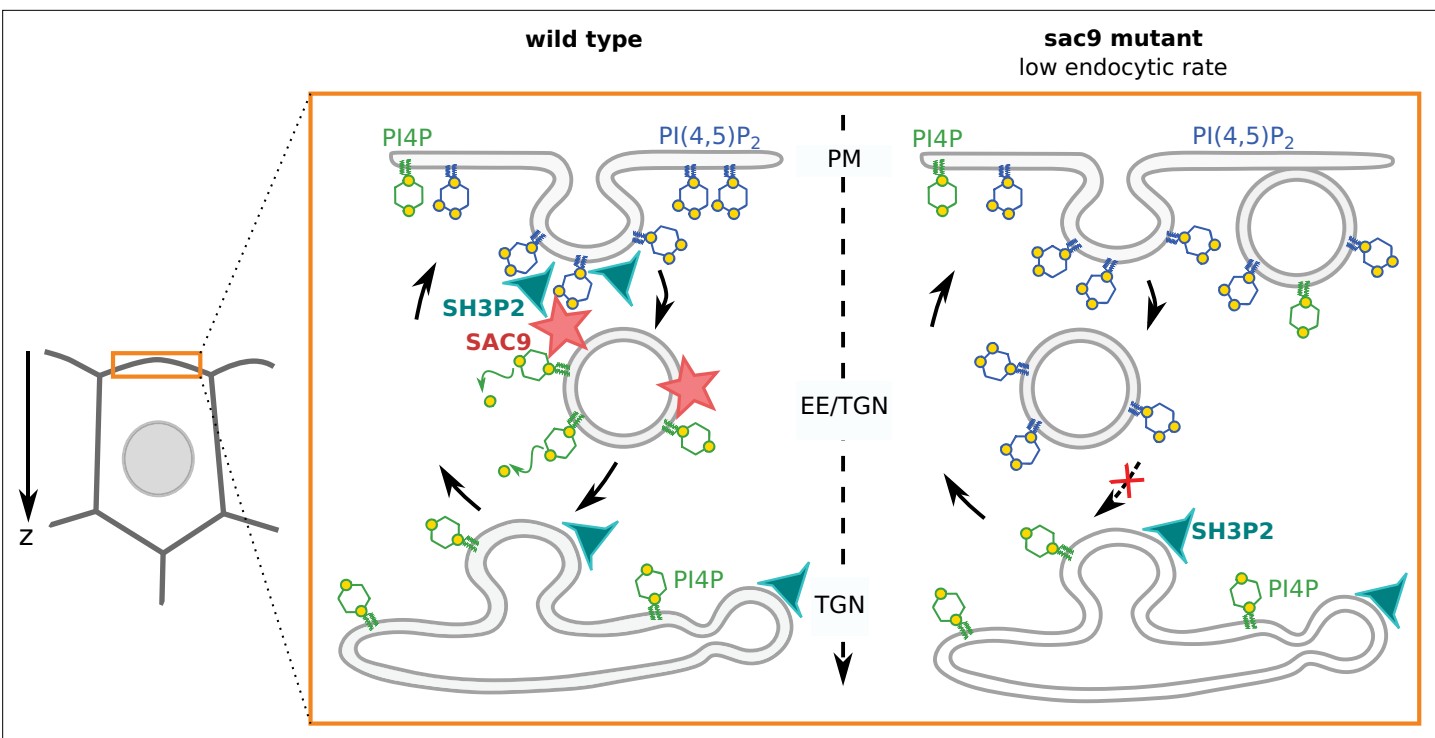

**Figure 8.** Model for the mode of action of SUPPRESSOR OF ACTIN9 (SAC9) in regulating PI(4,5)P$_2$ subcellular patterning. In wild-type plants, SAC9 restricts the localization of PI(4,5)P$_2$ at the plasma membrane allowing endocytic processes to occur. In the absence of SAC9, endocytosis PI(4,5)P$_2$ accumulates in atypical endomembrane compartments that are no longer able to fuse with the TGN possibly because of their abnormal anionic lipid signature. SAC9 interacts with SH3P2 close to the plasma membrane. The defects in PI(4,5)P$_2$ patterning in absence of SAC9 leads to decreasing in the endocytic rate and the formation of membrane protuberances in contact with the plasma membrane. Note that in this model, the observed increase in intracellular PI4P levels as measured by probes is not included.

most abundant clathrin-coated vesicles proteome components (*Dahhan et al., 2022*). Interestingly, in that study, no other 5-phosphatase was identified to be associated with clathrin-coated vesicles. Together, we propose that PI(4,5)P$_2$ accumulation on endocytic vesicles after scission could lead to an altered dynamics of these vesicles, which may be unable to fuse correctly with TGN/EEs, ultimately leading to their accumulation in the cell cortex (*Figure 8*).

We observed that the endocytic pathway is partially impaired in the absence of SAC9, and that *sac9* was oversensitive to ES9-17-mediated inhibition of endocytosis. In animal cells, membrane protuberances have been reported in cultured cells knocked-out or knock-down for enzymes of PI(4,5)P$_2$ metabolism (*Gurung et al., 2003*; *Terebiznik et al., 2002*; *Mochizuki and Takenawa, 1999*). These protuberances could therefore be a conserved process affected by disturbed PI(4,5)P$_2$ homeostasis in both plant and animal cells leading to such plasma membrane distortion. It is not the first observation of plasma membrane protuberances in cells where PI(4,5)P$_2$ homeostasis is affected: upon PIP5K6 over-expression in pollen tubes, which increases PI(4,5)P$_2$ levels, plasma membrane invaginations also occurs. The authors elegantly demonstrated that these plasma membrane distortions are due to clathrin-mediated endocytosis defects (*Zhao et al., 2010*). These structures were also reported when inducible depletion of PI(4,5)P$_2$ at the plasma membrane of root cells was performed using the iDePP system (*Doumane et al., 2021*), suggesting that the tight regulation of the PI(4,5)P$_2$ metabolism is important for the integrity of the plasma membrane. Previous study on the cellular organization of the primary roots of *sac9-1* seedlings at ultrastructural level reported the excessive membrane material either in direct contacts or close to the wall forming the protuberances (*Vollmer et al., 2011*). The irregular cell wall deposition in the *sac9* mutant leads to cell wall ingrowths, coined 'whorl-like structures' (*Vollmer et al., 2011*). We propose that such protuberances could be a readout of long-term endocytic defects, for instance, caused by an excess of plasma membrane due to unbalanced exocytosis. If so, the appearance of plasma membrane protuberances upon PI(4,5)P$_2$ metabolism perturbation could be an additional indication of disturbed endocytosis.

## How could SAC9 regulate endocytosis?

We observed that the endocytic pathway is partially impaired in the absence of SAC9. We envision several scenarios to explain these endocytic defects. They are not mutually exclusive and ultimately, it would not be surprising if the *sac9* endocytic phenotype results from a combination of altered cellular pathways relying, directly or indirectly, on the precise spatio-temporal regulation of anionic lipid homeostasis. First, as mentioned in the paragraph above, the *sac9* mutant likely accumulates ectopic vesicle following endocytosis. These vesicles trap membrane components inside the cell, which may overall impact the endocytic process. In addition, it is possible that the localization or dynamics of PI(4,5)$P_2$ binding proteins, which regulate clathrin-mediated endocytosis, could be impacted, as we observed for SH3P2 and TPLATE (*Zhuang et al., 2013*; *Yperman et al., 2021b*; *Yperman et al., 2021a*). Other proteins from the TPLATE complex also interact with anionic lipids, including PI(4,5)$P_2$, and could be affected in *sac9* (*Yperman et al., 2021b*; *Yperman et al., 2021a*). Furthermore, the dynamics of dynamin-related proteins, which contain a lipid-binding plekstrin homology domain, and the AP2 adaptor complex, which relies on PI(4,5)$P_2$ for localization in planta (*Doumane et al., 2021*), could also be altered in *sac9*. It was also recently reported that perturbation of clathrin-containing vesicles at the TGN/EEs also impacts clathrin-mediated endocytosis (and conversely) (*Yan et al., 2021*). It is thus possible that perturbing phosphoinositides metabolism after the scission of clathrin-coated vesicles from the plasma membrane impacts the dynamics of clathrin-mediated endocytosis at the cell surface in an indirect way, for example, because of slowed recycling of the clathrin pool or other endocytic components at the surface of TGN/EEs.

It is also important to note that bulk endocytosis in general and clathrin-mediated endocytosis in particular are still going-on in the *sac9* mutant, albeit at a reduced rate. Thus, while PI(4,5)$P_2$ strict exclusion from intracellular membranes is important for the endocytic process, the plant membrane trafficking system appears to be extremely resilient, as it manages to operate despite the accumulation of these membranes of mixed identity inside the cell. Here, we can speculate that SAC9 may not be the only 5-phosphatase enzyme that can control PI(4,5)$P_2$ homeostasis during endocytosis. In addition, while PI(4,5)$P_2$ is the main phosphoinositide regulating the recruitment of endocytic regulators in animals, it is likely not the case in plants (*Marković and Jaillais, 2022*). Indeed, PI4P, not PI(4,5)$P_2$, is the major anionic lipids that powers the plasma membrane electrostatic field (*Simon et al., 2016*), and is very likely key to recruit endocytic proteins in plants (*Yperman et al., 2021b*; *Yperman et al., 2021a*). In addition, phosphatidic acid (PA) is also important to drive the electrostatic field of the plasma membrane (*Platre et al., 2018*) and PA could be involved in the localization of AtEH1/Pan1, a component of the TPLATE complex (*Yperman et al., 2021a*; *Dragwidge et al., 2022*).

## Limitation of the study

SAC9 was previously proposed to act as a PI(4,5)$P_2$ 5-phosphatase in planta, mainly because the *sac9* mutant accumulates PI(4,5)$P_2$ (*Williams et al., 2005*). We found an accumulation of PI(4,5)$P_2$ sensors inside the cells in *sac9*, which is fully consistent for a 5-phosphatase enzyme. However, we could not confirm this activity in vitro. This is mainly due to problems in purifying SAC9 catalytic domain in sufficient quantity. Yet, it represents a weakness of our study that we should keep in mind. Indeed, although this is not the most parsimonious explanation, it is possible that SAC9 is not a 5-phosphatase and that instead, it controls the localization or activity of another enzyme bearing this catalytic activity. Nonetheless, the fact that the putative catalytically inactive mCIT-SAC9$^{C459A}$ fusion protein does not rescue *sac9* macroscopic phenotype and PI(4,5)$P_2$ localization suggests that SAC9 phosphatase activity is required for its function. We showed that the number of BFA bodies labeled with FM4-64 in *sac9-3* vs wild-type is reduced. Such reduction may be due to (i) a lower amount of BFA bodies formed per cell (i.e. the number per cell area or volume), (ii) reduced FM4-64 internalization from the cell surface, (iii) defects in the balance between endocytic and exocytic trafficking which alter TGN/EE function, and (iv) a combination of those. In either case, such decrease suggests that membrane trafficking flux through the endosomal system is impacted in *sac9*.

To explain the endocytic defects observed in the *sac9* mutant, we mainly focused on PI(4,5)$P_2$ mis-patterning. However, biochemical measurements also showed that the *sac9* mutant accumulates less PI4P than its wild-type counterpart. Given the importance of PI4P for plant cell function (*Marković and Jaillais, 2022*; *Noack et al., 2022*; *Noack et al., 2020b*; *Simon et al., 2016*), it is possible that PI4P rather than (or in combination with) PI(4,5)$P_2$ defects are involved in the *sac9* phenotypes. Given that

phosphoinositide metabolism is highly intricate, we recognize that it is difficult to fully untangle the specific involvement of each lipid in the observed phenotypes. In addition, the SAC9 protein may carry specific functions outside of its catalytic activity. For example, SAC9 directly interacts with SH3P2, yet the physiological relevance of this interaction for both SAC9 and SH3P2 function remains unclear. Structure-function analyses aiming at dissecting this interaction will be instrumental to clarify this point. Based on confocal imaging, we concluded that the level of plasma membrane-associated SH3P2-sGFP is reduced in the *sac9* mutant. In addition, the overall levels of SH3P2 are not affected in the mutant. Subcellular fractionations and quantitative colocalization analyses between SH3P2-GFP and compartment markers in the *sac9* mutant background will be needed to fully understand how SAC9 impacts SH3P2 localization. In addition, the exact function of SH3P2 in endocytosis is also elusive and it will be important to analyze it in detail in the future (*Adamowski et al., 2022*). Finally, one of the clear limitations in interpreting the *sac9* phenotype, which is common to most genetic approaches, is the accumulation of defects over a long-time period in the mutant. As such, it is impossible to pin-point toward direct or indirect effects of $PI(4,5)P_2$ mis-patterning on the cellular and developmental phenotypes of this mutant. Future studies, aiming at rapidly manipulating SAC9 function and localization in an inducible manner will be instrumental in disentangling the direct and indirect effects of SAC9 on lipid dynamics and endocytosis regulation.

## Materials and methods

### Cloning

*SAC9* promoter (*SAC9pro*; 800 bp) was amplified from Col-0 genomic DNA using *SAC9prom_fwd_gb* ttgtatagaaaagttgctattgaaaaaagatagaggcgcgtg and *SAC9prom_rev_gb* TTTTTTGTACAAACTT GCCTGAGCTCAGGACCAAGCGG primers and the corresponding PCR product was recombined into *pDONR-P1RP4* (Life Technologies https://www.thermofisher.com/in/en/home.html) vector by BP reaction to give *SAC9pro/pDONR-P1RP4*. The *mCIT* and *TdTOM* containing vectors *cYFPnoSTOP/pDONR221* and *TdTOMnoSTOP/pDONR221* were described before (*Simon et al., 2014*; *Jaillais et al., 2011*).

The genomic sequence of *SAC9* (At3g59770) was amplified by PCR using 7-day-old *Arabidopsis* seedlings gDNA as template and the *SAC9-B2R* GGGGACAGCTTTCTTGTACAAAGTGGCTATGG ATCTGCATCCACCAGGTTAGT and *SAC9-B3wSTOP* GGGGACAACTTTGTATAATAAAGTTGCT CAGACACTTGAAAGGCTAGTCCAT primers. The corresponding PCR product was recombined into *pDONR-P2RP3* vector by BP reaction to give *SAC9g/pDONR-P2RP3*.

SAC9$^{\Delta CC}$ was amplified from *SAC9g/pDONR-P2R-P3* with SAC9g_deltaCC_C-F /5phos/ggaattgat ccagctacc SAC9g_deltaCC_N-R /5phos/AAGCTTCTTTGCCTGTATTATG primers and then ligated to give *SAC9g$^{\Delta CC}$ /pDONR-P2R-P3*.

SAC9$^{C459A}$ mutation was obtained by site-directed mutagenesis using the partially overlapping SAC9-C459A-F tacgttttaacgctgctgattccttggatcgaacaaatgc and SAC9-C459A-R AGGAATCAGCAG CGTTAAAACGTATCACCCCATTTGATGTG primers on *SAC9g/pDONR-P2RP3*.

The gateway expression vector SH3P2shortprom (0.4 kb): SH3P2gDNA-tdTOM was obtained from pDONR P4-P1R SH3P2 short prom amplified with the primers: attb4-sh3p2promshortF GGGGACAA CTTTGTATAGAAAAGTTGCTGGTTAAGGGTCTTCTAGATGGTGTAG, attb1-sh3p2promR GGGG ACTGCTTTTTTGTACAAACTTGCTCTTCACCAGATCAAGAGCTATTCACAAA, pDONR221/SH3P2g amplified with the primers: attB1_SH3P2g_F GGGGACAAGTTTGTACAAAAAAGCAGGCTTAACC_ ATGGATGCAATTAGAAAACAAGCTAG, attB2_SH3P2g_R GGGGACCACTTTGTACAAGAAAGC TGGGTA_GAAAACTTCGGACACTTTGCTAGCAAGAAC and pDONR-P2R-P3 TdTOM together with pDEST pH7m34GW by LR GW Final destination vectors were obtained using three fragments LR recombination system (Life Technologies, https://www.thermofisher.com/in/en/home.html) using pB7m34GW destination vector (*Karimi et al., 2007*). The following reactions were set-up to generate the corresponding destination vectors: UBQ10prom: tdTOM-SAC9g/pH, SAC9prom: mCIT-SAC9g/ pB, pAtSAC9: mCIT-SAC9g$^{DEAD}$/pB, SAC9prom: TdTOM-SAC9g$^{DEAD}$/pH, pAtSAC9: mCIT-SAC9g$^{\Delta CC}$/ pB, SH3P2shortprom: SH3P2gDNA-tdTOM, pUb10: SH3P2gDNA-tdTOM (*Supplementary file 1*).

### Growth condition and plant materials

*Arabidopsis thaliana* Columbia-0 (Col-0) accession was used as wild-type (WT) reference genomic background throughout this study. *Arabidopsis* seedlings in vitro on half Murashige and Skoog (½ MS)

basal medium supplemented with 0.8% plant agar (pH 5.7) in continuous light conditions at 21 °C. Plants were grown in soil under long-day conditions at 21 °C and 70% humidity 16 hr daylight. Wild-type Col-0 and heterozygous (or homozygous) *sac9-3* were transformed using the dipping method (*Clough and Bent, 1998*).

For each construct generated in this paper (UBQ10prom: tdTOM-SAC9g/pH, SAC9prom: mCIT-SAC9g/pB, pAtSAC9: mCIT-SAC9g$^{DEAD}$/pB, SAC9prom: TdTOM-SAC9g$^{DEAD}$/pH, pAtSAC9: mCIT-SAC9g$^{ΔCC}$/pB, SH3P2shortprom: SH3P2gDNA-tdTOM, pUb10: SH3P2gDNA-tdTOM), between 20 and 24 independent T1 were selected on antibiotics (Basta or hygromycin) and propagated. In T2, all lines were screened using confocal microscopy for fluorescence signal and localization. Between 3 and 5 independent lines with a mono-insertion and showing a consistent, representative expression level and localization were selected and grown to the next generation. Each selected line was reanalyzed in T3 by confocal microscopy to confirm the results obtained in T2 and to select homozygous plants. At this stage, we selected one representative line for in-depth analysis of the localization and crosses and two representative lines for in-depth analysis of mutant complementation.

## FM4-64 staining and drug treatments

Seedlings were incubated in wells containing 1 µM FM4-64 (Life Technologies T-3166; from a stock solution of 1.645 mM=1 mg/ml in DMSO) in half Murashige and Skoog (½ MS) liquid basal medium without shaking for 30 min and in dark. Seedlings were then mounted in the same medium and imaged within a 10 min time frame window (1 hr ± 5 min).

Seedlings were incubated in wells containing Brefeldin A (BFA; Sigma B7651) applied at 50 µM (from a stock solution of 30 mM in DMSO), or a corresponding volume of DMSO as 'mock' treatment, dissolved in liquid ½ MS for 1 hr in dark without shaking before mounting in the same medium and imaging. For co-treatment with 50 µM BFA and 1 µM FM4-64, FM4-64, and BFA were added at the same time. Imaging was performed within a 14 min time frame window (1 hr ± 7 min).

For PAO treatment, seedlings were incubated in wells containing 30 µM PAO (Sigma P3075, https://www.sigmaaldrich.com/IN/en, PAO stock solution at 60 mM in DMSO), or a volume of DMSO as mock treatment, during the indicated time. Roots were imaged within a 10 min time frame window around the indicated time. The PAO effects on the localization of the biosensors were analyzed by counting manually the number of labeled compartments per cell.

ES9-17 was stored at –20 °C as a 30 mM stock solution in DMSO. Seedlings were incubated in dark without shaking for the indicated duration (± 7 min) in liquid ¼ MS (pH 5.7) in wells containing 1% DMSO to help solubilization of ES9-17, and either 1 µM ES9-17 or the corresponding additional volume of DMSO (mock treatment). For 1 µM ES9-17 and 1 µM FM4-64, FM4-64 was added 30 min after the ES9-17 (the indicated time corresponds to ES9-17 exposure).

## Western blot

To verify that both TPLATE and SH3P2 fusion proteins were both identically expressed in wild-type and *sac9*, 10-day-old seedlings were grind in liquid nitrogen, weighed, and resuspended in 1 ml of extraction buffer (100 mM Tris–HCl pH 7.5, 150 mM NaCl, 5 mM EDTA, 5% glycerol, 10 mM DTT, 0.5% Triton X-100, 1% Igepal, and 1% protease inhibitors (Sigma) into milliQ water) and centrifuged at 5000 rpm for 10 min at 4 °C to obtain the total protein extract in the supernatant. The total protein extract was then filtered on column and 100 µl of the filtrate denatured with 50 µl of Lamelli buffer 3 x at 95 °C for 5 min. 40 µl of the total protein extract was loaded onto 7.5% polyacrylamide gel, run for 120 min at 120 V and blotted on nitrocellulose membranes 2 hr, 100 V on ice. Membrane was then incubated 5 min in red-ponceau buffer, washed with milliQ water, and imaged. Then membranes were blocked in 5% milk dissolved in TBST buffer (10 mM Tris-HCL, 150 mM NaCL, 0.05% Tween20) for 1 hr. mCIT tagged proteins were revealed by using, respectively, GFP monoclonal antibody (anti-GFP mouse monoclonal, Roche; at 1/1000 in 5% milk over-night) as primary antibodies and anti-mousse IgG-HRP conjugated secondaries antibodies (Mouse IgG, HRP conjugate W402B, Promega; 1/5000 in TBST, 4 hr). As a loading control, we used anti-tubulinα antibodies as primary antibodies (1/1000 in 5% milk over-night). Finally tagged proteins were detected by chemiluminescence using ECL, substrate to HRP enzyme, revelation.

## Transient expression in *N. benthamiana*

The AtSH3P2-GFP constructs were described previously (*Leong et al., 2022*). The binary plasmid was transformed into *Agrobacterium tumefaciens* strain C58C1 and transient expression of the construct was performed by infiltration of *N. benthamiana* leaves at the four-to six-leaf stage.

## Immunoprecipitation

To verify that SAC9 fusion proteins were present in the plant cells, leaves from 1-month-old plants were ground in liquid nitrogen, weighed, and resuspended in 1 ml of extraction buffer (100 mM Tris–HCl pH 7.5, 150 mM NaCl, 5 mM EDTA, 5% glycerol, 10 mM DTT, 0.5% Triton X-100, 1% Igepal, 100 µM MG132 and 1% protease inhibitors (Sigma) into milliQ water) and centrifuged at 5000 rpm for 10 min at 4 °C to obtain the total protein extract in the supernatant. The total protein extract was then filtered on column, and incubated 40 min with 50 µl of magnetic protein G Dynabeads (Millipore) fused to a GFP monoclonal antibody (1/500 in PBS 0.002% tween). Beads, antibodies, and antibodies bund proteins were magnetically precipitated on columns, eluted and denatured in 40 µl of Laemmli buffer 2x at 95 °C for 5 min. 20 µl of the immunoprecipitation was loaded, blotted on nitrocellulose membranes, incubated in red-ponceau buffer and blocked in 5% milk as previously described. Tagged SAC9 fusion proteins were revealed by using GFP monoclonal antibody (anti-GFP mouse monoclonal, Roche) and detected by chemiluminescence using ECL revelation as for TPLATE and SH3P2 fusion proteins.

Pull-down assays in *N. benthamiana* using GFP-Trap assays were performed as previously described (*Leong et al., 2022*). For immunoblot analysis, protein extracts were separated by SDS-PAGE, transferred to PVDF membranes (Biorad), blocked with 5% skimmed milk in TBS, and incubated with primary antibodies anti-GFP-HRP (SantaCruz) antibody using 1:2000 dilutions in TBS containing 0.1% Tween 20. The immunoreaction was developed using an ECL Prime Kit (GE Healthcare) and detected with Amersham Imager 680 blot and gel imager.

## Mass spectrometry data analysis

Proteins in the *N. benthamiana* proteome corresponding to the peptide hits were identified according to the annotated proteome (*Kourelis et al., 2019a*). The proteins were submitted to BLAST (*Sayers et al., 2022*) to identify their isoform(s) in the *A. thaliana* proteome (*Cheng et al., 2017*). The dataset containing the *A. thaliana* protein ID and peptide counts were submitted to the R package IPinquiry4 (https://github.com/hzuber67/IPinquiry4; *Zuber, 2020*) which uses a Genewise Negative Binomial Generalized Linear Model developed by EdgeR to calculate the log fold-change (log(FC)) and log10(p-value).

## NanoLC-MS/MS analysis and data processing

Proteins were purified on an NuPAGE 12% gel (Invitrogen) and Coomassie-stained gel pieces were digested in gel with trypsin as described previously (*Borchert et al., 2010*) with a small modification: chloroacetamide was used instead of iodoacetamide for carbamidomethylation of cysteine residues to prevent the formation of lysine modifications isobaric to two glycine residues left on ubiquitinylated lysine after tryptic digestion. After desalting using C18 Stage tips peptide mixtures were run on an Easy-nLC 1200 system coupled to a Q Exactive HF-X mass spectrometer (both Thermo Fisher Scientific) as described elsewhere (*Kliza et al., 2017*) with slight modifications: the peptide mixtures were separated using a 87-min segmented gradient from 10-33-50 to 90% of HPLC solvent B (80% acetonitrile in 0.1% formic acid) in HPLC solvent A (0.1% formic acid) at a flow rate of 200 nl/min. The seven most intense precursor ions were sequentially fragmented in each scan cycle using higher energy collisional dissociation (HCD) fragmentation. In all measurements, sequenced precursor masses were excluded from further selection for 30 s. The target values were 105 charges for MS/MS fragmentation and $3 \times 106$ charges for the MS scan.

Acquired MS spectra were processed with MaxQuant software package version 1.5.2.8 with integrated Andromeda search engine. Database search was performed against a *Nicotiana benthamiana* database containing 74,802 protein entries. Endoprotease trypsin was defined as protease with a maximum of two missed cleavages. Oxidation of methionine, phosphorylation of serine, threonine and tyrosine, GlyGly dipeptide on lysine residues, and N-terminal acetylation were specified as variable modifications. Carbamidomethylation on cysteine was set as fixed modification. Initial maximum

allowed mass tolerance was set to 4.5 parts per million (ppm) for precursor ions and 20 ppm for fragment ions. Peptide, protein, and modification site identifications were reported at a false discovery rate (FDR) of 0.01, estimated by the target-decoy approach (Elias and Gygi). The iBAQ (Intensity Based Absolute Quantification) and LFQ (Label-Free Quantification) algorithms were enabled, as was the 'match between runs' option (*Schwanhäusser et al., 2011*).

### Live cell imaging

Most images (see exceptions below) were acquired with the following spinning disk confocal microscope set up: inverted Zeiss microscope (AxioObserver Z1, Carl Zeiss Group, http://www.zeiss.com/) equipped with a spinning disk module (CSU-W1-T3, Yokogawa, https://www.yokogawa.com/) and a ProEM +1024 B camera (Princeton Instrument, http://www.princetoninstruments.com/) or Camera Prime 95B (https://www.photometrics.com/) using a 63 x Plan-Apochromat objective (numerical aperture 1.4, oil immersion). GFP and mCITRINE was excited with a 488 nm laser (150 mW) and fluorescence emission was filtered by a 525/50 nm BrightLine! single-band bandpass filter (Semrock, http://www.semrock.com/), mCHERRY en TdTOM was excited with a 561 nm laser (80 mW) and fluorescence emission was filtered by a 609/54 nm BrightLine! single-band bandpass filter (Semrock, http://www.semrock.com/). For quantitative imaging, pictures of epidermal root meristem cells were taken with detector settings optimized for low background and no pixel saturation. Care was taken to use similar confocal settings when comparing fluorescence intensity or for quantification.

Colocalization experiments were performed on an inverted Zeiss CLSM710 confocal microscope or an inverted Zeiss CLSM800 confocal microscope and inverted Zeiss CLSM980 confocal microscope using a 63 x Plan-apochromatic objective. Dual-color images were acquired by sequential line switching, allowing the separation of channels by both excitation and emission. In the case of colocalization, we also controlled for a complete absence of channel crosstalk. GFP was excited with a 488 nm laser, mCIT was excited with a 515 nm laser and mCH/tdTOM were excited with a 561 nm laser. Imaging was performed in the root epidermis in cells that are at the onset of elongation. Only cells imaged at their Zi were considered for the colocalization analysis.

### Yeast two-hybrid screen

The yeast two-hybrid screen was performed by hybrigenics services (https://www.hybrigenics-services.com/contents/our-services/interaction-discovery/ultimate-y2h-2), using the ULTImate Y2H screen against their Universal *Arabidopsis* Normalized library obtained using oligo_dT. A codon optimized residue (aa 499–966) of SAC9 gttaataatcaggggggatataacgctccccttccaccgggatgggaaaaaagagctgat gccgtaactggaaaatcatattatatagatcacaatacaaagacaacaacatggagtcatccatgtcctgataaaccatggaagagac ttgacatgaggtttgaggaatttaagagatcaactatcttatctcctgtgtcagaacttgccgatctttttctgcaacaaggtgatatccat gcaaccctctatactggctcgaaagctatgcacagccaaattctcaacatcttcagtgaagaatcaggagcatttaaacagttttctgcagc acagaaaaacatgaagattacactacagagaagatataaaaatgctatggttgatagttcacggcaaaaacagctcgagatgtttctg ggaatgaggcttttcaagcatcttccatcaattcctgtccagcctttacatgtactttctcgaccatctggtttctttctgaaacctgtacctaac atgtccgaaagttccaatgatgggtccagtctgctgagtatcaagaggaaggacataacttggctatgtccacaagctgcagatattg ttgaattatttatctatctcagtgagccttgccatgtatgtcaacttctactgaccatatcacacggtgcggatgatttgacatgtccatccactg tggacgtgagaactggacgccacatagaggaccttaaattagttgttgagttagttcaactggattaccgattacctgtaattatgtttttct ggacagggtgcttcaataccacgctgtgcaaatggtacaaatcttctggtacccttaccagggccaattagttctgaggatatggctgttac tggagctggtgcacgtcttcatgaaaaagatacgtcaagtctttcactgctatatgattttgaagaactagaaggacagttggatttcttaa cccgtgtagttgctgttacatttttatccagctggtgctgttagaattcctatgactcttggtcagatagaagtccttggaatttctcttccatgga aaggaatgtttacttgtgaacgtactggaggaagattagctgaacttgcaaggaaaccagatgaagatggaagtcctttttcatcttgttct gacttgaatccgtttgctgcaacaacatctttacaggctgaaactgtttccacaccagtacaacagaaggatcccctttcccagtaatctgct tgacctttttgacaggagaggactcttcttctgacccccttcccacaaccagtggtggaatgtattgcaagtggaggcaatgacatgcttgatt tcttagacgaagcagttgttgaatatcgcggctctgacactgttcctgacgggtct was cloned in a pB66 vector (N-GAL4-bait-C fusion). The screen was performed on 0.5 mM 3AT. 59 million interactions were analyzed, and 260 positive clones were sequenced (ATNOR_dT_hgx4515v1 _pB66, *Supplementary file 1B*).

### TIRF imaging

3 days old etiolated seedlings were used for hypocotyl epidermal cells observations. TIRF-VAEM imaging was made using an ilas2 TIRF microscope (Roper Scientific) with 100 x Apo NA 1.46 Oil objective and a Prime 95B camera (Photometrics, https://www.photometrics.com/) and 1.5 coverslips were

used (VWR 61–0136). Time lapses were acquired during 300 time-points for 300 s (acquisition time 500 ms). Spot density was measured using Spot_detector ImageJ macro (*Bayle et al., 2017*). Since the data below 5 s and beyond 70 s exceeds the typical lifetime of the clathrin-coated vesicle at the plasma membrane, we removed them from the analysis.

Because manual verification of TPLATE-GFP lifetimes is greatly limited by the number of CCVs that can be detected, particle identification and tracking were performed using ImageJ plugin Trackmate (https://research.pasteur.fr/fr/software/trackmate/). Trajectories were reconstructed following a three-stage workflow: (i) detection of peaks potentially associated with fluorescent emitters, (ii) quality test and estimation of the subpixel position, and (iii) track reconnection. To discriminate between signal and background, particular attention has been paid to the size and shape of the observable objects. Particle of minimum size 0.5 with a threshold of 50 and a contrast >0.04 was filtered, to capture as many spots as possible without background. For many reasons, such as variation in fluorescent intensity, loss of focus or photobleaching, the emitter can be missing for several time points causing the premature stop of tracks. Therefore, the maximum number of frames separating two detections was set to three frames (*Bayle et al., 2021*). As a final verification, a visual inspection of the tracks can be performed on a reconstituted image, where all the tracks from a movie are represented.

2000 tracks were selected per acquisition starting from frame 5 to avoid segmenting truncated tracks. Acquisition was made on seven hypocotyl cells from three different plants per genotype and per replicate.

## Dissociation indexes

Dissociation indexes of membrane lipid fluorescent biosensors were measured and calculated as previously described (*Platre et al., 2018*). Briefly, we calculated 'indexNoDex, defined as the ratio between the fluorescence intensity (Mean Grey Value function of Fiji software) measured in two elliptical regions of interest (ROIs) from the plasma membrane region (one at the apical/basal plasma membrane region and one in the lateral PM region) and two elliptical ROIs in the cytosol in the mock condition. For quantification, we used FAPP-E50A as a positive control, since the delocalization of a biosensor from the plasma membrane to the cytoplasm helps us to use automatic tools.

## Measures, counting, and statistical analysis

Primary root length and number of lateral roots were manually measured from pictures. For comparing the primary root length and lateral root density between each genotype, we used a one-way ANOVA and post hoc Tukey HSD pairwise tests (95% confidence level).

For quantitative imaging, pictures of epidermal root meristem cells were taken with detector settings optimized for low background and no pixel saturation. Care was taken to use similar confocal settings when comparing fluorescence intensity. Pseudo-colored images were obtained using the 'Green Fire Blue' look-up-table (LUT) of Fiji software (http://www.fiji.sc/). The intracellular compartments ('spots') per cell were automatically counted.

We automatically measured the density of intracellular compartments labeled per root using the procedure described in *Bayle et al., 2017* for each biosensor, and we used two-sided non-parametric Wilcoxon rank-sum tests to compare Col-0 and *sac9-3*[-/-] genotypes in *Figure 4*. To account for multiple testing, we used a Bonferroni correction and lowered the significance level of each test at alpha = 0.05/11=0.004.

To assess the effect of the inhibitor PAO on anomalous mCIT-TUBBYc intracellular compartments, we manually counted their number per cell. We tested PAO effect and the effect of treatment duration (30 min and 2 hr) on the number of marked intracellular compartments using a generalized linear mixed-effect model (Poisson family) to account for image ID as a random factor. Two-sided post-hoc tests were performed using the R package 'lsmeans' (*Lenth, 2016*). We compared the number of FM4-64 positive compartments in Col-0 and *sac9-3*[-/-] using a generalized linear mixed model (Poisson family) to account for image ID (*id est* root) as a random factor.

The density of FM4-64 labeled compartments was also compared in Col-0 and *sac9-3*[-/-] using a linear mixed model accounting for image ID as a random factor, followed by a Wald $\chi^2$ test (function *Anova* in R package 'car').

To compare the effects of BFA on FM4-64, we tried to automatically count the number and size of the BFA bodies in Col-0 and *sac9-3*[-/-] seedlings, but the analysis was not optimal to treat the images

acquired for sac9-3. We, therefore, manually counted the number of BFA body per cell in multiple samples, using the same region of the root. We then compared the results of the BFA treated Col-0 and *sac9-3*$^{-/-}$ seedlings using a generalized linear mixed model (Poisson family) with image ID (*id est* root) as a random factor (Wald $\chi^2$ test: $\chi^2$ = 33.8, p<0.001). To compare the effects of BFA on Col-0 and *sac9-3*$^{-/-}$ seedings expressing PIN2-GFP, we counted manually and compared the treatments BFA-Col-0 with BFA-Sac9-3 using a generalized linear mixed model (Poisson family) with image ID (*id est* root) as a random factor (Wald $\chi^2$ test: $\chi^2$ = 42.1, p<0.001). For the dissociation index analysis, we performed all our statistical analyses in R v. 3.6.1, (*R Development Core Team, 2019*), using R studio interface and the packages ggplot2 (*Wickham, 2016*), lme4 (*Bates et al., 2014*), car (*Fox and Weisberg, 2018*), multcomp (*Hothorn et al., 2008*) and lsmeans (*Lenth and Lenth, 2018*). To compare TPLATE-GFP density between Col-0 and sac9-3$^{-/-}$, we used a two-sided non-parametric Kruskal Wallis rank-sum tests for each replicate and obtained each time a statistical difference (p value <0.05 between the two genotypes). Graphs were obtained with R and R-studio software using the package 'ggplot2,' and customized with Inkscape (https://inkscape.org).

## Quantification of mCIT-SAC9 compartment densities and colocalization with compartments markers

Because the signal of mCIT-SAC9 is mainly diffused in the cytosol, no automatic spot detection could be used for quantification of densities and colocalization analyses in *Figures 2, 3 and 6*. Therefore, for comparing the number of intracellular compartments containing mCIT-SAC9 or mCIT-SAC9$^{C459A}$ protein-fusions per cell across conditions, we manually counted them and used either a generalized linear mixed-effect model (Poisson family) for counting comparisons or a linear mixed effect model (and associated ANOVAs) for density comparisons, accounting for image ID (id est root) as a random factor.

Since the localization of the marker for the membrane compartment was larger in z compared to the restricted localization of SAC9 (only present close to the surface of the cell), we counted the number of mCIT-SAC9 labeled structures which were also labeled by the compartment markers in the cell cortex (Zi plane). The percentage of endosomes labeled by mCIT-SAC9 or mCIT-SAC9C459A colocalizing with a given marker, counted manually are presented in the graphs. Positive colocalization was called when the compartment marker was present as a dotted structure overlaying the mCit-SAC9 signal.

The same approach was used to deduce the localization of the mutated version of SAC9. After running our mixed models, we subsequently computed two-sided Tukey post hoc tests (function glht in R package 'multcomp,' *Hothorn et al., 2008*) to specifically compare each pair of conditions.

## Acknowledgements

We are grateful to the SiCE group (RDP, Lyon, France), Dr Yohann Boutté (LBM, Bordeaux, France), Dr Fabrice Besnard and Dr Nicolas Doll (RDP, Lyon, France), Hugo Ducuing (INMG, Lyon, France), Sébastien This (CIRI, Lyon, France) and Augustin Le Bouquin (IGFL, Lyon, France), Dr Sophie Piquerez (I2BC, Paris-Saclay, France) for comments and discussions. We thank Patrice Bolland, and Alexis Lacroix from our plant facility, and Claire Lionnet (RDP, Lyon, France). We are also grateful to Dr Daniël van Damme (VIB, Ghent, Belgium) for sharing with us TPLATE-GFP and for discussions. We would like to thanks Pr Dr Erika Isono (University of Konstanz, Konstanz, Germany) for sharing SH3P2-sGFP transgenic line with us. We thank Hybrigenics for the yeast two hybrid screen. We are also grateful to E Russinova (VIB, Ghent, Belgium) for kindly providing us ES9-17; S Bednarek for sharing markers and for discussions. We acknowledge the contribution of SFR Biosciences (UAR3444/CNRS, US8/Inserm, ENS de Lyon, UCBL) facilities, notably the LBI-PLATIM-MICROSCOPY for assistance with imaging. This work was supported by two Seed Fund ENS LYON-2016 and LYON-2019 (to MCC), a Junior Investigator grant ANR-16-CE13-0021 (to MCC), ERC no. 3363360-APPL under FP/2007–2013 (to YJ). This work was supported by an Emmy Noether Fellowship GZ: UE188/2-1 from the Deutsche Forschungsgemeinschaft (DFG; to SÜ). MD and AL were funded by Ph.D. fellowships from the French Ministry of Research and Higher Education.

## Additional information

### Funding

| Funder | Grant reference number | Author |
| --- | --- | --- |
| Agence Nationale de la Recherche | ANR-16-CE13-0021 | Marie-Cécile Caillaud |
| European Research Council | 3363360-APPL | Yvon Jaillais |
| Deutsche Forschungsgemeinschaft | UE188/2-1 | Suayib Üstün |

The funders had no role in study design, data collection and interpretation, or the decision to submit the work for publication.

### Author contributions

Alexis Lebecq, Data curation, Formal analysis, Validation, Investigation, Visualization, Methodology; Mehdi Doumane, Data curation, Formal analysis, Validation, Investigation, Visualization, Methodology, Writing - original draft; Aurelie Fangain, Formal analysis, Validation, Methodology; Vincent Bayle, Data curation, Formal analysis, Methodology; Jia Xuan Leong, Conceptualization, Resources, Data curation, Formal analysis, Methodology, Writing – review and editing; Frédérique Rozier, Data curation, Formal analysis; Maria del Marques-Bueno, Laia Armengot, Romain Boisseau, Mathilde Laetitia Simon, Formal analysis; Mirita Franz-Wachtel, Resources, Data curation, Software, Formal analysis, Methodology; Boris Macek, Data curation, Formal analysis, Supervision, Methodology; Suayib Üstün, Data curation, Formal analysis, Supervision, Funding acquisition, Validation, Investigation, Methodology, Writing – review and editing; Yvon Jaillais, Conceptualization, Resources, Supervision, Investigation, Methodology, Project administration, Writing – review and editing; Marie-Cécile Caillaud, Conceptualization, Resources, Data curation, Formal analysis, Supervision, Funding acquisition, Investigation, Writing – review and editing

### Author ORCIDs

Mehdi Doumane http://orcid.org/0000-0002-5711-3626
Laia Armengot http://orcid.org/0000-0002-3790-9838
Romain Boisseau http://orcid.org/0000-0003-4317-1064
Boris Macek http://orcid.org/0000-0002-1206-2458
Yvon Jaillais http://orcid.org/0000-0003-4923-883X
Marie-Cécile Caillaud http://orcid.org/0000-0002-0348-7024

### Decision letter and Author response

Decision letter https://doi.org/10.7554/eLife.73837.sa1
Author response https://doi.org/10.7554/eLife.73837.sa2

## Additional files

### Supplementary files

• MDAR checklist

• Supplementary file 1. Ressources and statistics. (A) Reagent and resources. (B) Raw data from ULTImate Y2H SCREEN *Arabidopsis thaliana* - SAC9 (aa 499–966) vs Universal Arabidopsis Normalized_dT. (C) Details of the statistics corresponding to *Figure 1C, D*. (D) Details of the statistics corresponding to *Figure 2D, G*. (E) Details of the statistics corresponding to *Figure 3G-F*. (F) Details of the statistics corresponding to *Figure 4B*. (G) Details of the statistics corresponding to *Figure 5*. (H) Details of the statistics corresponding to *Figure 6E, G*. (I) Details of the statistics corresponding to *Figure 7B*. (J) Details of the statistics corresponding to *Figure 1—figure supplement 1*. (K) Details of the statistics corresponding to *Figure 3—figure supplement 2*.

### Data availability

The mass spectrometry data from this publication will be made available at the PRIDE archive (PXD033585).

The following dataset was generated:

| Author(s) | Year | Dataset title | Dataset URL | Database and Identifier |
|---|---|---|---|---|
| Franz-Wachtel M, Üstün S | 2022 | Functional characterization of SH3P2 from *Arabidopsis thaliana* | https://www.ebi.ac.uk/pride/archive/projects/PXD033585 | PRIDE, PXD033585 |

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
