## [Editor Report]

Phosphoinositide phosphates (PIPs) are lipids that can convey distinct identities to different cellular membranes via different phosphorylation patterns. Here, Lebecq, Doumane, and co-authors document the effects of the previously-characterized sac9 mutant, affecting a putative PIP-5-phosphatase in Arabidopsis, on PIP localization and endocytic trafficking. This work confirms that disrupting PI(4,5)P2 localization or abundance can affect endocytic trafficking in plants and will be of interest to the plant and cell biology research fields.

---

## [Decision Letter]

**Decision letter after peer review:**

Thank you for submitting your article "The Arabidopsis SAC9 Enzyme defines a cortical population of early endosomes and restricts PI(4,5)P_2_ to the Plasma Membrane" for consideration by *eLife*. Your article has been reviewed by 3 peer reviewers, one of whom is a member of our Board of Reviewing Editors, and the evaluation has been overseen by Jürgen Kleine-Vehn as the Senior Editor. The following individuals involved in review of your submission have agreed to reveal their identity: Sebastian Y. Bednarek (Reviewer #2); Clara Sanchez-Rodriguez (Reviewer #3).

Essential revisions:

1) The claim that SAC9 is indeed a 5-phosphatase could be supported by imaging PI(4,5)P2 markers sac9 mutants carrying CIT-SAC9(C459A) to document similar changes to π distribution in the "catalytically dead" version as in the sac9 loss of function mutant. Please also revise the text surrounding the claims that SAC9 is a phosphatase, for example by referring to the C459A mutant "putative catalytically inactive"

2) Additional imaging data need to be presented to document that SAC9 is localized to a "cortical subpopulation" of early endosomes. To support this claim, the authors would need to carefully compare the distribution of SAC9 relative to other TGN/EE markers (e.g internalized FM4-64, SPY61, VHAa1 etc.) throughout the entire volume of cells. Alternatively, the authors may remove these claims from the manuscript. In either case, it will be important to clearly indicate which focal plane is being presented in each image, as noted by Reviewer 3.

3) At least one additional method is required to verify the interaction between SAC9 and SH3P2: CoIP, BiFC FRET, proximity labelling are all appropriate, assuming rigorous controls are also presented.

4) For the colocalization experiments, there seems to be a disconnect between the images and the quantification. Additional information in the methods could help clarify this. Reviewer 2 has suggested a control colocalization experiment between a soluble protein (e.g. untagged mCIT) and several markers. This is feasible within the timeline for revisions since imaging could be conducted within the F1 generation.

5) For the claims that SAC9 is involved in endocytosis, it is essential to eliminate the trivial explanation that endocytosis-related proteins are downregulated in sac9, rather than specifically depleted from the PM (e.g. via western blot of TPLATE and SH3P2, as Reviewer 3 suggests). This claim could be further supported by investigating the possibility that other SH3P2-related processes (MBV formation, autophagy, cell plate formation) are affected (e.g. marker line imaging in the sac9 mutant), as suggested by Reviewer 1 or testing whether loss of sac9 might generally affect intracellular trafficking by assessing TGN/EE function (e.g. PIN2 localization/recycling, secGFP imaging), as suggested by Reviewer 2.

6) The authors must present evidence of analyzing multiple independent transgenic lines for complementation experiments, as raised by Reviewer 1.

7) The FM4-64 uptake experiments need to be re-analyzed (or perhaps re-performed) on samples that show comparable PM staining between wild type and sac9 mutants. This is essential to eliminate the trivial explanation for reduced FM4-64 uptake into cells, since it is possible FM4-46 integration into the plasma membrane is affected in sac9 mutants due to changes in PM composition.

*Reviewer #1 (Recommendations for the authors):*

1. The authors repeatedly claim that "In this study, we show that the phosphoinositide phosphatase activity of SAC9 is required for its function." (line 75, line 100, line 282, etc) but they have not documented any biochemical activity of SAC9. They say that these experiments were unsuccessful; attempts to purify SAC9 and determine its activity should be presented in the supplemental data to support these claims. As a proxy, the authors rely on the mCIT-SAC9(C459A) mutant, which they assume is catalytically inactive. They provide a reasonable discussion of the caveats to their data, but this paper is weakened without any evidence of SAC9 activity. It would support their claims that mCIT-SAC9(C459A) is catalytically inactive to document similar changes to PI(4,5)P2 distribution in sac9 mutants carrying CIT-SAC9(C459A). It's also not appropriate to call C459 the "catalytic cystine" (line 94) without documenting activity.

2. Furthermore, only one independent transgenic line is presented for complementation experiments and the C459A line clearly expresses much less protein than wild type SAC9 complementation line (Supplemental Figure 1A), so it is quite possible that the lack of sac9 rescue by mCIT-SAC9(C459A) is simply due to less (but potentially fully functional) SAC9 protein in this line. Please present complementation data from at least three independent transgenic lines for each construct and document SAC9 protein levels in each line, especially the CIT-SAC9(C459A) lines.

3. The authors make statements about the localization of SAC9 such as "fluorescent SAC9 protein fusions localize…in a subpopulation of endosomes close to the plasma membrane" (line 149, 284, 311, 312). There are clearly SAC9-labelled puncta not at the cell cortex (for example, Figure 2D, the lefthand cell is an endoplasmic plane of section through the vacuole and many bright SAC9 puncta are visible quite far from the PM). The authors could remove these unsupported claims; however, this would substantially dimmish their central claim that SAC9 is somehow specific to endocytosis and specifically labelling new endocytic vesicles.

4. The only data presented to support the interaction between SAC9 and SH3P2 is Y2H results from truncated versions of both proteins. This unverified interaction must be supported by at least one other method to document protein-protein interactions.

5. The authors claim that SAC9 function is specific to endocytosis (line 331). However, they have not tested any other processes in sac9 mutants, including SH3P2-mediated processes such as multivesicular body formation (Nagel et al., 2017 PNAS), cell plate formation (Ahn et al., 2017 Plant Cell), autophagy (Zhuang et al., 2013 Plant Cell), so it is inappropriate to claim that SAC9 has such a specific function. The authors could either assess MVB formation, autophagy, and cell plate formation in sac9 mutants or they could remove these claims.

*Reviewer #2 (Recommendations for the authors):*

1. The authors show that the number of BFA bodies labeled with FM4-64 in sac9-3 vs wild-type is reduced, consistent with a decrease in endocytosis of the tracer dye. However, this assumes that the formation of BFA bodies (i.e. the number per cell area or volume) in sac9-3 mutants is similar to that of wild-type. Quantitation of BFA body formation, using markers of the TGN/EE in wild-type and mutant cells should thus be presented to rule out that formation of BFA bodies is not altered in the sac9 mutant cells.

2. The authors should explain their rationale for using the enzymatically inactive SAC9 variant in co-localization experiments with SH3P2 (Figure 7) rather than wild-type SAC9. Based on the representative images shown in Figure 3H (which do not necessarily correspond with the quantitative data shown in Figure 3G; see comment above) SAC9C459A appears to be associated with the TGN/EE, Golgi and late endosomal compartments raising questions as to which compartment(s) wild-type SAC9 and SH3P2 colocalize with.

3. In lines 157-158, the authors state, "As expected, both PI(4,5)P2 biosensors strictly labeled the plasma membrane in wild-type cells (Figures 4A and 4C)." However, while this appears to be the case for 2xPHPLC probe, the TUBBYc probe shows significant labeling in the cytoplasm and nucleus (Figure 4A, Col-0 background). Please modify the statement and/or explain the observed pattern.

4. It is interesting that the intracellular levels of PI(4)P appear to increase in the sac9 mutant. This is particularly evident in sac9 cells expressing the PHFAPP1 probe (Figure 4B). Is the enhanced intracellular labeling associated with changes in the districution of membrane associated ARF GTPase? Please discuss as the model presented in Figure 9 does not address the increased levels of PI(4)P in the sac9 mutant.

5. The authors should describe their rationale for using the ARF protein binding defective mutant form of the PI(4)P probe, FAPP-E50A in Figure 5, as opposed to the WT variant of FAPP1 marker used elsewhere in this study.

6. The authors should describe whether the image shown at higher magnification in Figure 4F is from the Zi or Zii plan of focus.

7. The authors need to provide more information in the manuscript text or methods section to explain how they calculated/quantitated the 'density' of intracellular puncta in the various backgrounds. Does density refer to the number of endosomes labeled by FM4-64, e.g. per cell? Or, does it refer to the number of intracellular puncta relative to the area of the cell imaged? Similarly, how were the number of BFA bodies quantitated (Figure 6)?

8. The authors should discuss their assignment of RabD1 as a post-Golgi endosomal marker (Figure 3). Based on the findings of Pinheiro et al., 2009 J Cell Sci., YFP-RabD1 colocalizes with internalized FM4-64 and VHA-a1 markers of the TGN/EE. However, while SAC9 appears to colocalize with FM4-64 it does colocalize significantly with RabD1. Please discuss or explain this apparent discrepancy.

9. Based on confocal imaging, the authors conclude that the level of plasma membrane associated SH3P2-sGFP is partially reduced in the sac9 mutant. Additional experiments including quantitative analysis of the total levels and distribution of SH3P2 in sac9 mutant and wild type subcellular fractionations (e.g. enriched plasma membrane, microsomal, and cytosolic fractions) would address whether loss of SAC9 affects the levels of SH3P2 and provide complementary data supporting the authors' conclusions.

10. The authors use TIRF microscopy to quantitate TPLATE abundance at the plasma membrane and describe a decreased density of puncta labeled by TPLATE in sac9 backgrounds. Is it biologically significant to show the data beyond 100 seconds in panel 8D? This exceeds the typical lifetime of the clathrin coated vesicle at the plasma membrane. Here, I would also ask the authors to demonstrate by immunoblotting that the total protein level of TPLATE does not change in sac9 backgrounds, to ensure that the decrease in the levels of plasma membrane-associated TPLATE is not due to a decrease in the total abundance of TPLATE.

*Reviewer #3 (Recommendations for the authors):*

Important aspects of image acquisition and data analysis, among others, need to be clarified and extended:

1) Figure 1B-D. How the authors can explain that the overexpression of SAC9 under the pUBQ10 promoter is not translated in a plant phenotype considering its function? Did the authors check whether in the pUBQ10::TdTOM-SAC9 line there is an increase of the SAC9 gene expression or protein level (is not included in the WB analysis of Supp Figure 1A)?

2) Figures 2, and 3: Images shown present cells imaged in different focal planes (for example, cells C1 and C2 in Fig2G, where C1 image is not at the subcortical focal plane indicated by the blue line in Figure 2A). We recognize the difficulty of imaging cells in the same focal plane and the need of recording and analyzing different z-stacks to quantify the structures in the same focal plane in different cells. Please clarify how these images were analyzed. If required, show only cells imaged in the required focal plane.

3) Figure 2F. The co-expression of the WT version of SAC9 and the mutated SAC9C459A. It is expected by the authors that the mutation in SAC9 affects mainly the function but not the localization of SAC9. In that case, expressing (overexpressing?) SAC9, would not prevent SAC9C459A to localize in the vesicular structures since WT SAC9 would allow proper endocytic trafficking? Please discuss.

4) Figure 2G: The localization of the native SAC9 at the PM by co-localization with a PM reported, as done for SAC9C459A is required.

5) Line 112-113: "mCIT-SAC9 and TdTOM-SAC9 were mainly soluble and excluded from the nucleus"; and Line 114-115: "We observed that mCIT-SAC9C459A was less soluble with a three-fold increase in the number of mCIT-SAC9C459A labeled dotty structures". To employ soluble to describe the localization of a protein does not seem to be the most appropriate term. SAC9 seems to be localized mainly diffused in the cytosol, but the mutated version accumulates in vesicular compartments.

6) Figure 3A. For the FM4-64 staining, there is a clear difference in the portion of the plasma membrane (PM) stained by FM4-64 in SAC9 WT compared to mutated SAC9. Please, discuss: are they performed using the same conditions? Could the absence of SAC9 (mutant) or the expression of the non-functional SAC9 alter the PM composition (as a complementary information of the Figure 4)?

7) Figure 3C-H: The authors quantified the colocalization of SAC9 and its mutant version with different organelles markers, suggesting that SAC9 colocalized mainly with EE/TGN structures. However, in the presented pictures seems obvious that SAC9 also colocalizes with LE/MVB structures, although in the quantification it is scored with low colocalization numbers. It is not very clearly explained in the Material and Methods section how the colocalization quantification was performed, manually counting or mediating other methods, i.e., Pearson coefficient? How the z-stack was selected? Always to the same distance to the PM? Are the vesicles that are transported to the vacuole excluded from the PI(4,5)P2 to PI4P conversion (it is explained in the text that they are richer in PI3P)? Please include RabF2a in the quantification in Figure 3G.

8) Figure 3C-D: The CLC2-RFP labeled structures look significantly different in the SAC9 C459A compared to the SAC9 WT version. Please, discuss.

9) Figure 4: In the text it is described an increase of the number of intracellular compartments label by the PI4P biosensor in the sac9-3 mutant (Figure 4B-C), suggesting that it happens due to a depletion of the PI4P pool at the PM and a relocation to the vesicles. How can the authors explain that this can happen? How could you demonstrate that this increase of the PI4P in the intracellular compartments is from the PM? And why is not happening in the mCIT-PHOSB sensor? Also, in the figure is not clear that there is any increase in the intracellular compartments using the mCIT-P4MSidM sensor. In the cited paper was quantified the number of intracellular compartments labeled with the mCIT-P4MSidM and mCIT-PHOSB, which was practically 0, being very surprising the quantification of the intracellular particles in both cases in the present paper. In these sensors, there is not an obvious decrease (is not quantified) of the PM signal that could explain the increase of the intracellular compartments signal.

10) Line 174-177: "When co-imaging 2xmCH-2xPHFAPP1 together mCIT-TUBBYc in sac9-3-/- (Figure 4D), the dotty structures decorated by the PI(4,5)P2 biosensor -but not with the PI4P biosensor- were observed at the cortex of the cell, at the close vicinity with the plasma membrane (Figure 4D upper panel and 4E), whereas those structures were rarely observed in the internal part of the cell". In that case, what are the structures decorated with PI4P? Co-expression with the different organelle markers or/and SAC9C459A could be explanatory.

11) Are the intracellular compartments labeled by the PI(4,5)P2 sensors the same where SAC9C459A is accumulating (or where WT SAC9 can be also found)?

12) Line 186-188: in vivo time-lapse imaging of PI(4,5)P2 biosensor mCITTUBBYc and mCIT-2xPHPLC in sac9-3-/- mutant revealed that those intracellular structures were mobile in the cortex of root epidermal cells, hence, behaving like intracellular compartments (Supplemental Figure 3D, Supplemental video 2). Are they more or less mobile than in WT? This parameter could already point to the alteration of the endocytic dynamics.

13) Figure 6: The FM4-64 staining of PM is not homogenous even in the same plant. Same for the amount of a certain protein, like PIN2. Therefore, the endocytosis should be quantified using the ratio of internal signal/PM signal.

14) Figure 6A. There is a decrease in the number of FM4-64 labeled endosomes in the sac9-3 mutant (Figure 6A). Could that be explained due to a possible alteration of the PM density in the sac9-3 mutant? Is that a direct effect or a consequence of an increase of non-labeled FM4-64 vesicles (observed for the PI4,5P2 and PI4P biosensors)? In the case that PM is affected in the sac9-3 mutant, in order to calculate the density of intracellular compartments would be convenient to normalize it with the signal of "available lipids" at the PM to avoid indirect effects.

15) Figure 6G: It is already published (Vollmer et al., 2011) that sac9-3 mutant has wall protuberances that are randomly distributed and can be extended as a consequence of the PM accumulation. In that case, is the appearance of these protuberances an indirect effect of the lower endocytosis rate in the sac9-3 mutant? It seems to appear even in absence of inhibitors of the endocytosis. In that case, would these PM-protuberances be enriched with PI(4,5)P2 lipids?

16) Figure 7: SH3P2-sGFP and SAC9C459A are partially co-localizing (although colocalization quantification is missing and required). Are they also colocalizing when WT SAC9 version is used? Additionally, colocalization does not always imply interaction, and Line 285-286 "In planta, SAC9 interacts and colocalizes with the endocytosis component SH3P2" is an overstatement since no interaction assay has been done in planta (Only yeast two-hybrid). To further confirm the interaction of both proteins, other methods like Co-immunoprecipitation (with or without crosslinker) or biotin proximity labeling (PL) (Mair et al., 2019) are recommendable.

17) Lines 312-314: "Together, we propose that SAC9 represent the long-sought-after enzyme which performs the PI(4,5)P2-to-PI4P conversion during the plant endocytic process (Figure 9)." How is SAC9 recruited to the PM-vesicles for performing its function? It is clear that this question could be difficult to address for this publication, but could this recruitment be through the interaction with other proteins, i.e., SH3P2? Are they mutually needing each other to localize properly? In that case, what is the localization of SAC9 in the sh3p2 mutant background?

18) Figure 8A: the representative image shown for sca19-3 TPLATE-GFP is not homogeneously in focus.

19) Figure 2D: The reduction of TPLATE-GFP dwell time at the PM in sca9-3 is not obvious based on this figure. Include average, SD, and statistics. Also indicate in methods how these data were obtained. If a significant lower TPLATE-GFP dwell time in sac19-3 is confirmed, please, discuss this result in context with the published data showing an inverse correlation between TPLATE dwell time and endocytosis (longer time>less endocytosis; Wang et al., 2020, https://pubmed.ncbi.nlm.nih.gov/32321842/)

20) Please, re-check the methodology to describe the experimental set-up and data analysis in the detail required to be repeated by other colleagues. Among others, indicate what is "N" and "n" in the graphs, cite the published lines used in the study, and indicate how the particles were chosen for their analysis.

21) Avoid over conclusions not supported by data like

– Lines 11-12: «it interacts (only shown by Y2H) and colocalizes (not quantified) with the endocytic component Src 11 Homology 3 Domain Protein 2 (SH3P2)»

– Lines 132-133: "catalytically dead SAC9 fusion proteins localize to endosomes and are likely part of the early steps of endocytic trafficking pathway» No data at this point indicate that SAC9 can be part of the early steps of the endocytosis.

– Line 311: "SAC9 localizes to clathrin-coated vesicles close to the plasma membrane» This is not shown in the results.

[Editors’ note: further revisions were suggested prior to acceptance, as described below.]

Thank you for resubmitting your work entitled "The Arabidopsis SAC9 Enzyme is enriched in a cortical population of early endosomes and restricts PI(4,5)P_2_ at the Plasma Membrane" for further consideration by *eLife*. Your revised article has been evaluated by Jürgen Kleine-Vehn (Senior Editor) and a Reviewing Editor.

The manuscript has been improved but there are some remaining issues that need to be addressed, as outlined below:

1) Explain discrepancies in the quantification of cortical vs endoplasmic SAC9-labelled particles (Reviewer 1 points 1 and 2 and Reviewer 2 point 2).

2) Clarify details about the number of transgenic lines (Reviewer 1, point 3 and Reviewer 2 point 6).

3) Clarify details of statistical analysis (Reviewer 1, point 4).

4) Clarify details of the colocalization experiments (Reviewer 2 points 4 and 8 and Reviewer 3 points 1 and 2).

Please also consider the detailed comments from reviewers, but a point-by-point response to all of their comments will not be strictly necessary for a revised version.

*Reviewer #1 (Recommendations for the authors):*

The authors have substantially revised the manuscript to address many of my previous comments, including adding several new experiments. They have analyzed π biosensors in the SAC9(C459A) to provide the support that this mutation affects SCA9 enzymatic activity. The evidence documenting SAC9 interactions with SH3P2 is now much stronger with the addition of SH3P2-GFP IP-MS experiments. The localization data are now much better aligned with the authors' claims and much more clearly communicated. The authors have also provided a very detailed response to the previous reviewer comments. However, several of my previous comments have not been adequately addressed:

1. Quantification of cortical vs endoplasmic SAC9-labelled particles has been added to the manuscript in the figure on P. 45 G (the figures are not numbered in the document I was sent for review, so I refer to them by page number in the pdf). However, there is a flaw with the approach: density per cell is a misleading measurement since, in the zii plane, there are of course fewer puncta per area since there is less cytoplasm because the nucleus and vacuole take up about half of the area. Please present data as punta per area of cytoplasm. If the differences in SAC9 puncta density in cortical/endoplasmic cytoplasm do not hold when properly quantified, please revise the text and title accordingly.

2. Why is p. 45 G "density of SAC9 puncta per cell" in a range of 0.001-0.003, but presented as "number of SCA9 puncta per cell" in p 53 C in a range of 5-20? Why present two different measures? Why are there 10-fold more SAC9 puncta per cell than FM4-64 puncta in the figure on p 35 C vs D, when the authors described these markers as colocalized (line 160)?

3. The number of independent transgenic lines analyzed is still not indicated. The text says "multiple independent lines" (line 103) and evidence from only one or two lines is presented in the supplemental figures. Please present data from at least three independent transgenic lines for each new construct.

4. It is not clear in the main figure legends or text what N and n are in the graphs. If this means seedlings and cells, please clarify how statistical analyses are being conducted (i.e. which is being used as sample size). Inappropriately identifying N drastically affects p-values, and therefore conclusions from statistical analysis. N is the number of independent biological replicates (e.g. plants), not the number of measurements taken (Lord et al., 2020 J Cell Biol). For example, it’s unlikely that >1000 independent plants were analyzed in the figure on p. 48 B. Please revise accordingly.

5. Introduction line 27: why redefine endocytosis, rather than just calling this “endocytic trafficking” throughout (i.e. as you do on line 41)? This will be less confusing to the broad readership of *eLife*.

6. The article requires careful proofreading, particularly for tense use/agreement, article use, and number agreement. As just a few examples: intro line 18 should read “abundance” not “abundant”, in intro line 31: remove “the” from “the endocytosis”, intro line 54: “FM4-64 experiment” should be plural, intro line 78: “the sac9 mutants is dwarf” should be “sac9 mutants are dwarf”

7. Timestamp and scale bars are missing in supplemental videos.

*Reviewer #2 (Recommendations for the authors):*

In my opinion, the authors have overall satisfactorily addressed the editor's and my major comments/concerns. As detailed below I have only a few remaining issues (that do not require further experimentation) that I feel the authors should address.

Response to Editor's comments/concerns:

1. Imaging PIP2 marker in a sac9 C459A mutant background to see changes in PI distribution as in loss-of-function sac9 background

This is supported. The marker mCIT-TUBBYc is imaged in WT, sac9, and sac9C459A lines. The images of both mutant backgrounds look identical, but this is not quantitated. It should be noted that imaging of mCIT-TUBBYc in the WT was done at the zii level (4A) while in sac9 is at zii and zi (4A and 4C) but C459A is in zi (4C).

2. Confirm cortical nature of SAC9: image this relative to TGN/EE markers in multiple layers of the cell AND clarify when each image is taken in the cortex or interior of the cell

Overall the analysis is improved. Most images indicate whether the plane of focus is at the cell cortex or interior using the appreciated zi vs zii notation. The images and quantitation in Figures 2F and 2G indicate that there is a statistically significant difference in the number of endosomes labeled by SAC9 between the cortex and interior.

I remain somewhat concerned that the analysis would have been more convincing had the authors compared the distribution of wild-type SAC9 relative to intracellular FM4-64. Supplemental Figure 2 imaging comparison was conducted between mCIT-Sac9delta 999-1027 and FM4-64. This is confusing as in figure 1 the authors show that the SAC9 mutant variant lacking the coiled coil region is cytosolic. More informative would have been the comparison of wild-type SAC9 and internalized FM4-64 rather than PH domain or Lti6b reporters in Zi and Zii focal planes.

3. Verification of SH3P2 and SAC9 interaction by additional method (e.g. coIP or FRET)

I feel that the additional reciprocal co-IP data presented in Figure 6 showing that SH3P2-GFP interacts with tobacco SAC9 addresses this concern. Additionally, confocal microscopy shows that localization of SH3P2 to the plasma membrane is strongly affected by the loss of SAC9.

4. Address disconnect between images and quantitation of images and/or image mCIT (or other tags) relative to markers used.

This is somewhat supported. Harmonization of the majority of images in the manuscript is appreciated but is not totally consistent (e.g. mCIT-TUBBYc imaging in Figure 4A and 4C; imaging of SAC9 between mock and BFA treatments occurs in zi and zii, respectively). Figure 3 remains the same as in the previous manuscript draft, where the authors had included images of RabF2 colocalization with C459A SAC9 but not the corresponding quantitation and the authors had included the quantitation of VTI12 with WT SAC9 but did not show the image. In this revision, the authors have supplemented the quantitation in Figure 3F with 'representative' colocalization image of WT SAC9 and VTI12 in Figure 3-supplemental Figure 1, but it is not apparent if the image shown in the supplement is actually quantitated in the main manuscript figures. (Note: the panel in Figure 3-supplemental Figure 1 is not labeled as VTI12 but instead as W13R - is this the same? Authors need to make it clear, as the figure legend for Figure 3-supplemental Figure 1 refers to VTI12.) The authors do not address the fact that C459A SAC9 colocalization with CLC2 is quantitated but not shown by images or that RabF2 colocalization with C459A SAC9 is not quantitated (Figures 3G and 3H). I would ask that the authors confirm that the quantitation of the colocalization between VTI12 and WT SAC9 directly corresponds to the image shown or otherwise replace the quantitation in Figure 3 with that directly corresponding to Supplemental Figure 3.

In response to the suggestion from reviewer 2, there is no imaging of mCIT alone relative to the other markers used.

The additional data corresponding to the loss of the coiled-coil domain (SAC9-deltaCC) resulting in the loss of endosomal localization pattern is interesting to note. And, while the inability of this variant to rescue the sac9 mutant indeed supports that this feature is important for the function of the protein, it does not necessarily indicate that the coiled-coil region mediates membrane association. But, this is asserted only in the response to reviewers and not in the manuscript itself.

5. Confirm that loss of endocytic proteins at PM in sac9 is not due to their downregulation. Possibly also address whether sac9 affects other TGN/EE-related processes, e.g. post-Golgi trafficking.

Overall the authors have addressed this concern however the quality of the immunoblot in Supplementary data figure 6 is low and the data is not quantitated. Immunoblot analyses show that with equal loading (as assessed by Ponceau and anti-tubulin) of total protein extracts from WT and sac9 plants, the GFP signal of TPLATE-GFP or SH3P2-GFP does not change in the sac9 mutant. Curious - what are the roughly 67 kDa bands present between the columns where WT and sac9 total protein extract were loaded? The data in Figures 6F and 6G show better than the ratio of SH3P2 in the cytosol relative to the PM is increased in sac9 relative to WT and is actually more convincing in showing that downregulation contributes less to decreased abundance of SH3P2 at the PM than does the change in SH3P2 re-distribution to the cytosol.

The authors do have the tools to assess disruption of post-Golgi trafficking in sac9 backgrounds, as they already have a PIN2::PIN2-GFP in sac9 line which was used for a PIN2-GFP localization to BFA body assay in Figure 5. This assay has been used to show that endocytosis is disrupted in sac9 background because the distribution of cells with BFA bodies labeled by PIN2-GFP is decreased/shifted to the left compared to WT. Problematically, cycloheximide has not been used in this BFA assay. The internalization of FM4-64 in Figures 5A-5D is more appropriate to show that endocytosis is impaired, and perhaps the BFA/PIN2-GFP internalization assay could be moved to the supplement of the manuscript. But, ultimately, I am satisfied by the authors' statement in the Discussion that likely multiple explanations exist for why sac9 displays impaired endocytosis independently of/concomitant with PIP-related factors.

6. Analyze multiple independent transgenic rescues.

This is supported. Supplemental Figures 1A and 1B demonstrate multiple sac9 alleles, a full rescue of sac9 by two different fluorescent tag fusions of SAC9, and the inability of multiple transformants of delta CC SAC9 to rescue sac9 (but only one transformant of C459A inability to rescue sac9 is shown). The additional language regarding independent transformants is helpful.

7. Quantitate FM4-64 internalization defects of sac9 lines using images where FM4-64 staining at the PM is comparable to WT.

This is supported. Supplemental Figure 5 shows that FM4-64 staining at the PM in sac9 is comparable to WT.

Response to Reviewer #2 comments/concerns:

1. Validate the use of FM4-64 labeling of BFA bodies in sac9 by showing that formation of BFA bodies in this mutant is similar to WT (e.g. by showing that BFA body formation labeled by TGN/EE markers is unaltered in sac9).

Although the author did not directly address this concern the authors effectively argue that endocytosis is impaired in sac9 mutants due to impaired internalization of FM4-64 and altered dynamics of endocytic protein players, and they also agree with the reviewer that, as is, their experiment is insufficient to show whether a combination of impairment of endocytosis and/or post-Golgi trafficking occurs in sac9. They have included a discussion about the interpretation of the BFA results (lines 474-476). Note, the authors should consider an additional alternative that BFA body formation is affected in the sac9 mutant due to defects in endocytic/exocytic which alter TGN/EE function as shown in the study by Yan et al Plant Cell 2021

2. Explain why colocalization between SH3P2 and SAC9 was not performed using WT SAC9.

This is mostly resolved by Figures 6D and 6E which show colocalization between SH3P2 and WT SAC9 as well as between SH3P2 and SAC9 C459A variant. However, a minor concern is that the imaging experiment with WT SAC9 is performed in the zi plane while the experiment with the C459A variant has been performed in the zii plane, and both are quantitated in panel 6E where the y-axis shows SAC9/SH3P2 colocalization in the zi plane. Authors should consider addressing this.

3. Address localization of PIP2 probe, mCIT-TUBBYc, to the cytoplasm/nucleus as well as PM in comparison to 2xPH probe which localizes only to PM.

Resolved by the new language.

4. Address apparent increase in intracellular PI4P levels in sac9 and how changes in PI4P levels fit into the model (Figure 8).

The authors state that we will not be able to resolve the interplay between the effects of PI4P and PIP2 in mediating sac9 and acknowledge that the effect on ARF1 GTPase is unknown. But, they do effectively argue that the observed increase in intracellular PI4P levels as measured by probes that do not localize concomitantly to the TGN/EE due to interaction with ARF1, e.g. mCIT-P4M, provides evidence that the ARF1 effect is not critical here. But, the authors do not satisfyingly address the role of PI4P in their model in this response.

5. Validate the use of ARF protein binding defective mutant, FAPP-E50A, as opposed to WT variant used elsewhere.

Satisfactorily, addressed by moving figure to supplemental materials.

6. Indicate whether the image shown at higher magnification in Figure 4F is from the Zi or Zii plane of focus.

Resolved by removing the image.

7. The authors need to provide more information in the manuscript text or methods section to explain how they calculated/quantitated the 'density' of intracellular puncta in the various backgrounds. Does density refer to the number of endosomes labeled by FM4-64, e.g. per cell? Or, does it refer to the number of intracellular puncta relative to the area of the cell imaged? Similarly, how was the number of BFA bodies quantitated (Figure 6)?

Satisfactorily addressed by the addition of quantitation methodology to Methods

8. Validate choice of RabD1 as a post-Golgi endosomal markers. Pinheiro et al. support the role of RabD1 as a post-Golgi marker as it colocalizes with FM4-64 and VHAa1.

The authors argue that the Pinheiro paper does not quantify these interactions, and so they have used as support the Geldner et al. Plant J 2009 paper. However, in the Geldner paper, the assignment of wave25 (RabD1) as a post-Golgi/endosomal marker protein appears arbitrary. Indeed in Table 2 Remarks that Geldner and colleagues state that RabD1 (wave25) is similar to wave lines 29 and 33 (i.e. RabD2a and D2b) which are assigned as Intermediate Golgi/endosomal. This is more similar to what was reported by Pinheiro and thus I feel that the authors are not justified in relying on RabD1 as a post-Golgi/endosomal marker.

9. Support decrease in PM associated SH3P2 with data showing that total levels of SH3P2 are not changed.

Overall satisfactory

See the response to the editor's comments/concerns point #5.

*Reviewer #3 (Recommendations for the authors):*

Doumane and colleagues have addressed most of the reviewer comments. Overall we are satisfied with the revised version of the manuscript. Most of the questions/comments have been answered and appear to support the authors' findings as written in the manuscript.

Nevertheless, for better clarification, and to fully support the publication of the manuscript, it will be beneficial to address some points not answered during the first revision:

1) It is not clarified yet in the material and methods how the co-localization was quantified. Please detail this point.

2) Co-localization of SAC9 and organelle markers. It was not discussed why mCit-SAC9-C459A seems to co-localize more importantly with LE-MVB markers, compared with the SAC9 wt version. Quantification is not included in Figure 3G.

3) For the PI4P biosensors (Fig 4), it would be recommendable to use pictures that are representative of the quantification of Fig4B. For instance, for mCIT-P4MSidM, 0 intracellular compartments are visible for sac9-3, but in the quantification, it is shown a clear increase of the intracellular compartments (two stars).

With regard to the clarification of the focal plane (commented in the first revision), would be recommendable to use replace the picture of sac9-3 x mCIT2xPHPLC (that is Zi) for a picture of the Zii plane, to be consistent with the rest of the pictures present in the panel (all of them in the plane Zii).

4) The manuscript would benefit from adding all the explanations included in the "response to reviewers". Ex. from our comment #9, among others.

Suggestion:

The classification of C1 and C2 (Fig 2A), it is not used in the rest of the paper, so it would be recommendable to erase it.

---

## [Author Response]

Essential revisions:1) The claim that SAC9 is indeed a 5-phosphatase could be supported by imaging PI(4,5)P2 markers sac9 mutants carrying CIT-SAC9(C459A) to document similar changes to π distribution in the “catalytically dead” version as in the sac9 loss of function mutant. Please also revise the text surrounding the claims that SAC9 is a phosphatase, for example by referring to the C459A mutant “putative catalytically inactive”

As suggested, to document if similar changes to phosphoinositides distribution in the C459A mutated version of SAC9 were observed (as it does in the sac9 loss-of-function mutant), we monitored the localization of the PI(4,5)P2 biosensor in sac9-3 background expressing SAC9^C459A^. We found that the intracellular structures igherd by mCIT-TUBBYc were still visible in sac9-3 expressing SAC9pro:tdTOM-SAC9^C459A^, confirming that C459A mutant is not functional both in terms of controlling normal plant growth at the whole plant levels and PI(4,5)P2 homeostasis at the cellular level. The data are now presented in Figure 4. In igherd, we carefuly reworded the manuscript and systematically refer to the C459 as the putative catalytic igherd and the C459A mutant as a putative catalytically inactive variant.

2) Additional imaging data need to be presented to document that SAC9 is localized to a “cortical subpopulation” of early endosomes. To support this claim, the authors would need to carefully compare the distribution of SAC9 relative to other TGN/EE markers (e.g internalized FM4-64, SPY61, VHAa1 etc.) throughout the entire volume of cells. Alternatively, the authors may remove these claims from the manuscript. In either case, it will be important to clearly indicate which focal plane is being presented in each image, as noted by Reviewer 3.

We harmonized the pictures in the manuscript, systematically showing the cortical view of the cell (Zi), and for some key experiments, the localization in the median plane of the cell, (Zii, plane going through the nucleus).

We quantified the number of endosomes stained by mCIT-SAC9 in these two focal planes (Zi and Zii) and quantified that mCIT-SAC9 localization in endosomes is more visible in the cortex of the cell than in its center, and igherd then to FM4-64 labelling. The data are now presented in Figure 2 and Figure 2-supplemental Figure 1. This data strongly support our observation about SAC9 subcortical localization.

We included additional quantifications, in particular between FM4-64/mCIT-SAC9 and SH3P2/SAC9. The data are presented in Figures 3, 7, and Figure 2-supplemental Figure 1.

3) At least one additional method is required to verify the interaction between SAC9 and SH3P2: CoIP, BiFC FRET, proximity labelling are all appropriate, assuming rigorous controls are also presented.

We encountered difficulties working with the SAC9 protein in vitro or with western blots extracted from Arabidopsis. To confirm the interaction, we thus collaborated with Suayb Üstün (ZMBP Tübingen) and his lab. They expressed SH3P2-GFP in Nicotiana benthamiana leaves and performed IP-MS experiments in the absence (3 replicates) and presence (3 replicates) of a proteasome inhibitor (to increase SAC9 stability). SAC9 was found as a top 10 interactor in all 6 experimental replicates (never found in the GFP only controls) and as the top SH3P2 interactor after proteasome inhibition. We also now include colocalization analysis (representative pictures and quantification) between SH3P2 and wild-type SAC9. In addition, we performed new experiments showing that SH3P2 plasma membrane association is compromised in the sac9 mutant. Together, with our yeast-two hybrid data, we believe that our new IP-MS and in planta colocalization analyses strengthen our conclusion on the SH3P2/SAC9 complex. – The data are presented in Figure 6.

4) For the colocalization experiments, there seems to be a disconnect between the images and the quantification. Additional information in the methods could help clarify this. Reviewer 2 has suggested a control colocalization experiment between a soluble protein (e.g. untagged mCIT) and several markers. This is feasible within the timeline for revisions since imaging could be conducted within the F1 generation.

We added information in the material and method explaining how the images were quantified, as well as in the legend of the figures.

We harmonized the pictures in the manuscript, systematically showing the cortical view of the cell (Zi), and for some key experiments, the localization in the median plane of the cell, (Zii, plane going through the nucleus).

We added representative images for the colocalization analysis of mCIT-SAC9 with the TGN marker VTI12 in Figure 3-supplemental Figure 1.

We are now including in the revised manuscript a mutated form of SAC9 in its predicted Coil-Coiled domain (SAC9-deltaCC, Figure 2B), which is no longer able to complement the mutant sac9-3 and is localized only in the cytoplasm. With this experiment, we:

1) confirm that the punctate localization observed with wild-type full length SAC9 is not due to dense cytoplasmic spots (since they cannot be observed with the cytosolic SAC9-deltaCC version),

2) uncover a molecular determinant in the SAC9 protein required for endosomal targeting.

3) further support that membrane association is required for SAC9 function.

5) For the claims that SAC9 is involved in endocytosis, it is essential to eliminate the trivial explanation that endocytosis-related proteins are downregulated in sac9, rather than specifically depleted from the PM (e.g. via western blot of TPLATE and SH3P2, as Reviewer 3 suggests). This claim could be further supported by investigating the possibility that other SH3P2-related processes (MBV formation, autophagy, cell plate formation) are affected (e.g. marker line imaging in the sac9 mutant), as suggested by Reviewer 1 or testing whether loss of sac9 might generally affect intracellular trafficking by assessing TGN/EE function (e.g. PIN2 localization/recycling, secGFP imaging), as suggested by Reviewer 2.

We are now presenting in Figure S5 by immunoblotting that the total protein level of SH3P2-GFP and T-PLATE-GFP do not dramatically change in the sac9 backgrounds. We can therefore conclude that the decrease in the levels of plasma membrane-associated SH3P2-sGFP is not due to a decrease in the total abundance of SH3P2. Similarly, the decreased density of T-PLATE-GFP in plasma membrane foci, as well as its alterned dynamics cannot simply be attributed to a igherdt expression level in sac9.

The plasma membrane localization of SH3P2 is specifically affected in sac9, while it still associates in intracellular compartments in this mutant. Data are presented in Figure 6

For the interest of time and the focus of the manuscript, we did not investigate other SH3P2-related processes in sac9. We think this is outside of the scope of this manuscript. We do not claim that SAC9 is involved only in endocytosis and in fact, given the widespread function of phosphoinositides, it is likely that it has additional functions outside of endocytosis. We have added a sentence to explain this point better in the revised version of the manuscript:

“We observed that the endocytic pathway is partially impaired in the absence of SAC9. We envision several scenarios to explain these endocytic defects. They are not mutually exclusive and ultimately, it would not be surprising if the sac9 endocytic phenotype results from a combination of altered cellular pathways relying, directly or indirectly, on the precise spatio-temporal regulation of anionic lipid homeostasis.”

In addition, as pointed out, SH3P2 is a igherdtional protein involved in many cellular processes and it is still unclear whether those SH3P2-related functions are independent or interrelated.

6) The authors must present evidence of analyzing multiple independent transgenic lines for complementation experiments, as raised by Reviewer 1.

We apologize because it was not clearly stated in the material and methods, but all over the study, and for each construct, at least 20 primary T1 transformants were selected. Independent T2 and T3 plants were obtained for subsequent analysis. We, therefore, selected multiple independent lines for the complementation and we obtained similar results. The data are presented in supplemental figure 1A, B.

7) The FM4-64 uptake experiments need to be re-analyzed (or perhaps re-performed) on samples that show comparable PM staining between wild type and sac9 mutants. This is essential to eliminate the trivial explanation for reduced FM4-64 uptake into cells, since it is possible FM4-46 integration into the plasma membrane is affected in sac9 mutants due to changes in PM composition.

FM4-64 itself is a lipid that intercalates in membrane. In addition, PI(4,5)P2 and PI4P represent less than a percent of the total phospholipids, which themselves are only about a third of total lipids. Moreover, PI4P and PI(4,5)P2 are embedded in the cytosolic leaflet of the plasma membrane while FM4-64 insert in the outer leaflet of the membrane. Altogether, we thus have no reason to believe FM4-64 labeling itself would be affected in sac9, and this is in line with our confocal observations.

Indeed, we did not observe a clear difference in between the WT and sac9-3 regarding the portion of the plasma membrane (PM) stained by FM4-64.

We added in supplemental figure 4 representative images used for the quantification with FM4-64.

Reviewer #1 (Recommendations for the authors):1. The authors repeatedly claim that “In this study, we show that the phosphoinositide phosphatase activity of SAC9 is required for its function.” (line 75, line 100, line 282, etc) but they have not documented any biochemical activity of SAC9. They say that these experiments were unsuccessful; attempts to purify SAC9 and determine its activity should be presented in the supplemental data to support these claims. As a proxy, the authors rely on the mCIT-SAC9(C459A) mutant, which they assume is catalytically inactive. They provide a reasonable discussion of the caveats to their data, but this paper is weakened without any evidence of SAC9 activity. It would support their claims that mCIT-SAC9(C459A) is catalytically inactive to document similar changes to PI(4,5)P2 distribution in sac9 mutants carrying CIT-SAC9(C459A). It’s also not appropriate to call C459 the “catalytic cystine” (line 94) without documenting activity.

In order to document if similar changes to phosphoinositides distribution in the sac9-3 expressing the C459A version of SAC9 was observed (as it does in the sac9 loss of function mutant), we used the PI(4,5)P2 biosensor localization in sac9-3 complemented line as a read-out for functional complementation. We showed that while the PI(4,5)P2 biosensor TUBBYc-mCIT was located in intracellular structure in sac9-3 mutants plants, the localization of the biosensor was fully restored in sac9-3 complemented with Ub10pro:tdTOM-SAC9 and SAC9pro:mCIT-SAC9. By contrast, the intracellular structures igherd by mCIT-TUBBYc were still visible in sac9-3 expressing SAC9pro:tdTOM-SAC9^C459A^, confirming that mutation in the putative catalytic cysteine abolished the function of the enzyme. The new data is now presented in Figure 4D.

2. Furthermore, only one independent transgenic line is presented for complementation experiments and the C459A line clearly expresses much less protein than wild type SAC9 complementation line (Supplemental Figure 1A), so it is quite possible that the lack of sac9 rescue by mCIT-SAC9(C459A) is simply due to less (but potentially fully functional) SAC9 protein in this line. Please present complementation data from at least three independent transgenic lines for each construct and document SAC9 protein levels in each line, especially the CIT-SAC9(C459A) lines.

We apologize because it was not clearly stated in the material and methods: All over the study, and for each construct, at least 20 primary T1 transformants were selected. Independent T2 and T3 plants were obtained for subsequent analysis. We, therefore, selected multiple independent lines for the complementation and we obtained similar results. Multiple independent lines are now presented in Figure 1-supplemental Figure 1.

We never noticed that CIT-SAC9(C459A) accumulated to a lesser extent than wild type SAC9 in our confocal analysis. We thus performed additional western blot in two independent lines expressing CIT-SAC9(C459A). This confirmed that CIT-SAC9(C459A) accumulates to similar extent (or perhaps slightly higher) than wild-type SAC9. Thus, SAC9(C459A) is indeed non-functional despite being expressed and stable. We have now included these new results in Figure 1-supplemental Figure 1.

3. The authors make statements about the localization of SA“9 such as "fluorescent SAC9 protein fusions localize…in a subpopulation of endosomes close to the plas”a membrane" (line 149, 284, 311, 312). There are clearly SAC9-labelled puncta not at the cell cortex (for example, Figure 2D, the lefthand cell is an endoplasmic plane of section through the vacuole and many bright SAC9 puncta are visible quite far from the PM). The authors could remove these unsupported claims; however, this would substantially dimmish their central claim that SAC9 is somehow specific to endocytosis and specifically labelling new endocytic vesicles.

By comparing the signal observed at the cortex of the cell (Zi focal lane) with the signal collected at the center of the same cell (Zii focal plane, Figure 2A), we observed and quantified that mCIT-SAC9 labeled more intracellular compartments close to the plasma membrane that a distal position (Figure 2F, G). This is clearly not a all or nothing localization, and for this reason we prefer to speak about “enrichment in the cortex of the cell”. We change the title accordingly to reflect this: “The Arabidopsis SAC9 Enzyme is enriched in a cortical population of early endosomes and restricts PI(4,5)P2 at the Plasma Membrane”

We agree that the proximity of the vacuole from the PM could be misleading: However, we regularly see the vacuole close to the PM, at Zi (see panel A and B figure 3 as example).

4. The only data presented to support the interaction between SAC9 and SH3P2 is Y2H results from truncated versions of both proteins. This unverified interaction must be supported by at least one other method to document protein-protein interactions.

We are now presenting set of independent evidences showing the interaction of SAC9 with SH3P2, obtained in collaboration with the laboratory of Dr. Suayb Üstün (ZMBP Tübingen) and his lab. They expressed SH3P2-GFP in Nicotiana benthamiana leaves and performed IP-MS experiments in the absence (3 replicates) and presence (3 replicates) of a proteasome inhibitor (to increase SAC9 stability). SAC9 was found as a top 10 interactor in all 6 experimental replicates (never found in the GFP only controls) and as the top SH3P2 interactor after proteasome inhibition. Together, with our yeast-two hybrid data, we believe that our new IP-MS and in planta colocalization analyses strengthen our conclusion on the SH3P2/SAC9 complex.

The data are presented in Figure 6.

5. The authors claim that SAC9 function is specific to endocytosis (line 331). However, they have not tested any other processes in sac9 mutants, including SH3P2-mediated processes such as multivesicular body formation (Nagel et al., 2017 PNAS), cell plate formation (Ahn et al., 2017 Plant Cell), autophagy (Zhuang et al., 2013 Plant Cell), so it is inappropriate to claim that SAC9 has such a specific function. The authors could either assess MVB formation, autophagy, and cell plate formation in sac9 mutants or they could remove these claims.

We remove this claim from the discussion as we don’t think thatSAC9 play a role only in endocytosis since PI(4,5)P2 is important for a plethora of processes in plants (see our response to the editor above). We however think that SAC9 dephosphorylates PI(4,5)P2 during endocytosis to remove it from the surface of nescent endocytic vesicles and/or early endosomes. Accumulation of PI(4,5)P2 inside the cell in sac9 in turn is likely to have pleiotropic effects, one such effect being a feedback on endocytosis itself.

Reviewer #2 (Recommendations for the authors):1. The authors show that the number of BFA bodies labeled with FM4-64 in sac9-3 vs wild-type is reduced, consistent with a decrease in endocytosis of the tracer dye. However, this assumes that the formation of BFA bodies (i.e. the number per cell area or volume) in sac9-3 mutants is similar to that of wild-type. Quantitation of BFA body formation, using markers of the TGN/EE in wild-type and mutant cells should thus be presented to rule out that formation of BFA bodies is not altered in the sac9 mutant cells.

We agree that a reduction of FM4-64 staining in BFA bodies could be due to (i) a lower amount of BFA bodies formed per cell (i.e., the number per cell area or volume), (ii) reduced FM4-64 internalization from the cell surface, or (iii) a combination of both. In either case, such decrease suggests that membrane trafficking flux through the endosomal system is impacted in sac9. Alone, such experiment would certainly not be enough to conclude on the impact of SAC9 on internalization from the plasma membrane. However, these data should be interpreted together with our additional endocytosis experiments, including: less FM4-64 internalization (no BFA), heightened sensitivity to endocytosis inhibition and the reduction in T-PLATE foci density and altered dynamics in sac9. To be more cautious, we have now included a discussion about the interpretation of the BFA result.

We also added in supplemental figure 4 representatives images used for the quantification of the BFA bodies stained with FM4-64.

2. The authors should explain their rationale for using the enzymatically inactive SAC9 variant in co-localization experiments with SH3P2 (Figure 7) rather than wild-type SAC9. Based on the representative images shown in Figure 3H (which do not necessarily correspond with the quantitative data shown in Figure 3G; see comment above) SAC9C459A appears to be associated with the TGN/EE, Golgi and late endosomal compartments raising questions as to which compartment(s) wild-type SAC9 and SH3P2 colocalize with.

As explained to Referee #1, the absence of this result was due to difficulties in obtaining the relevant genetic material by crossing (that we have since then obtained by transformation of the sac9-3+/- background). In the revised version of the manuscript, we are now presenting the colocalization between SAC9pro-mCIT-SAC9 x Ub10pro:TdTOM-SH3P2. With this additional experiment, we showed and quantifyed a significant colocalization between mCIT-SAC9 and SH3P2-tdTOM at the cortex of the cell. This new data is presented in the figure 6.

3. In lines 157-158, the authors state, "As expected, both PI(4,5)P2 biosensors strictly labeled the plasma membrane in wild-type cells (Figures 4A and 4C)." However, while this appears to be the case for 2xPHPLC probe, the TUBBYc probe shows significant labeling in the cytoplasm and nucleus (Figure 4A, Col-0 background). Please modify the statement and/or explain the observed pattern.

We rephrased accordingly:

“As previously described, both PI(4,5)P_2_ biosensors labeled the plasma membrane and were excluded form intracellular compartments in wild-type cells (Figure 4A and 4B)”.

4. It is interesting that the intracellular levels of PI(4)P appear to increase in the sac9 mutant. This is particularly evident in sac9 cells expressing the PHFAPP1 probe (Figure 4B). Is the enhanced intracellular labeling associated with changes in the distribution of membrane associated ARF GTPase? Please discuss as the model presented in Figure 9 does not address the increased levels of PI(4)P in the sac9 mutant.

We agree with the reviewer, and we discuss the effect on PI4P in the following section:

“Given the importance of PI4P for plant cell function (Noack et al., 2020; Simon et al., 2016), it is possible that PI4P rather than (or in combination with) PI(4,5)P_2_ defects are involved in the sac9 phenotypes. Given that phosphoinositide metabolism is highly intricate, we fully recognize that it is difficult to fully untangle the specific involvement of each lipid in the observed phenotypes. ”

We don’t know whether the ARF1 GTPase is affected. It could indeed impact the localization of the PHFAPP1 probe. However, because we have a similar trend with the P4M probe, which localizes independently of ARF1, we don’t think this is the major cause for the delocalization of those probes.

5. The authors should describe their rationale for using the ARF protein binding defective mutant form of the PI(4)P probe, FAPP-E50A in Figure 5, as opposed to the WT variant of FAPP1 marker used elsewhere in this study.

Figure 5 documents that long-term PAO treatment (which depletes PI(4,5)P2 as a consequence of its direct effect on PI4P) reduced the number of PI(4,5)P2-positive compartments in sac9 mutants. For quantification purposes, we used FAPP1-E50A as a positive control, since the delocalization of a biosensor from the PM to the cytoplasm helps us to use automatic tools (see material and methods). However, we are expecting to find similar results with other biosensors, as we described in Simon et al., 2016. As suggested by reviewer 1, this experiment which does not add to the main claims of the paper was moved to the supplemental material.

6. The authors should describe whether the image shown at higher magnification in Figure 4F is from the Zi or Zii plan of focus.

For clarity and space matters, we removed the image shown at higher magnification in Figure 4F.

7. The authors need to provide more information in the manuscript text or methods section to explain how they calculated/quantitated the 'density' of intracellular puncta in the various backgrounds. Does density refer to the number of endosomes labeled by FM4-64, e.g. per cell? Or, does it refer to the number of intracellular puncta relative to the area of the cell imaged? Similarly, how were the number of BFA bodies quantitated (Figure 6)?

We now provide more information in the manuscript text or methods section to explain how they calculated/quantitated the 'density' of intracellular puncta. “ To compare the effects of BFA on FM4-64 we tried to automatically count the number and size of the BFA bodies in Col-0 and sac9-3^-/-^ seedlings, but the analysis was not optimal to treat the images acquired for sac9-3. We therefore manually counted the number of BFA body per cell in multiple samples, using the same region of the root (see Supplemental Figure 4). We then compared the results of the BFA treated Col-0 and sac9-3^-/-^ seedlings using a generalized linear mixed model (Poisson family) with image ID (id est root) as a random factor (Type II Wald χ^2^ test : χ^2^ = 33.8, p < 0.001). To compare the effects of BFA on Col-0 and sac9-3^-/^seedings expressing PIN2-GFP, we counted manually and compared the treatments BFA-Col-0 with BFA-Sac9-3 using a generalized linear mixed model (Poisson family) with image ID (id est root) as a random factor. (Type II Wald χ^2^ test : χ^2^ = 42.1, p < 0.001). For dissociation index we performed all our statistical analyses in R v. 3.6.1, (R Core Team, 2019), using R studio interface and the packages ggplot2 (Wickham 2016), lme4 (Bates et al., 2014), car (Fox and Weisberg 2011), multcomp (Hothorn et al., 2008) and lsmeans (Lenth and Lenth 2018). To compared TPLATE-GFP density between Col-0 and sac9-3^-/-^ we used a two-sided non-parametric Kruskal Wallis rank-sum tests for each replicate and obtained each time a statistical difference (p.value < 0,05 between the two genotypes). Graphs were obtained with R and R-studio software, and customized with Inkscape (https://inkscape.org).

8. The authors should discuss their assignment of RabD1 as a post-Golgi endosomal marker (Figure 3). Based on the findings of Pinheiro et al., 2009 J Cell Sci., YFP-RabD1 colocalizes with internalized FM4-64 and VHA-a1 markers of the TGN/EE. However, while SAC9 appears to colocalize with FM4-64 it does colocalize significantly with RabD1. Please discuss or explain this apparent discrepancy.

We used the Geldner et al., 2009 reference paper, which is quantified. In the Pinheiro et al., 2009 J Cell Science paper, there is no quantification in figure 1 for the localization of YFP-RabD1, which makes comparison difficult.

9. Based on confocal imaging, the authors conclude that the level of plasma membrane associated SH3P2-sGFP is partially reduced in the sac9 mutant. Additional experiments including quantitative analysis of the total levels and distribution of SH3P2 in sac9 mutant and wild type subcellular fractionations (e.g. enriched plasma membrane, microsomal, and cytosolic fractions) would address whether loss of SAC9 affects the levels of SH3P2 and provide complementary data supporting the authors' conclusions.

We are now presenting in Figure S5 by immunoblotting that the total protein level of SH3P2-GFP does not dramatically change in sac9 backgrounds. We can therefore conclude that the decrease in the levels of plasma membrane-associated SH3P2 is not due to a decrease in its total abundance. We added in the result section:

“Using confocal imaging, we observed a diminution of the signal corresponding to SH3P2-sGFP at the plasma membrane compared to the cytoplasm in sac9-3, while the amount of SH3P2-sGFP detected via Western blot was similar between the two genotypes (Figure 6D-F, Figure 6-supplemental Figure 1)”.

We agree that subcellular fractionation of SH3P2 would have provided additional support for an altered localization of SH3P2 in the sac9 background. However, we struggled to obtain enough mutant material for such an experiment. Nonetheless, we do not think that fractionation is required for our conclusion, since the effect of the sac9 loss-of-function on SH3P2-GFP localization as observed by confocal microscopy and the additional western blot analysis is extremely clear.

Reviewer #3 (Recommendations for the authors):Important aspects of image acquisition and data analysis, among others, need to be clarified and extended:1) Figure 1B-D. How the authors can explain that the overexpression of SAC9 under the pUBQ10 promoter is not translated in a plant phenotype considering its function? Did the authors check whether in the pUBQ10::TdTOM-SAC9 line there is an increase of the SAC9 gene expression or protein level (is not included in the WB analysis of Supp Figure 1A)?

The use of ubiquitous promoter UBQ10 to drive the expression of SAC9, did not lead to ectopic mislocalization. We learned using SAC9 native promoter or UBQ10pro, which both complement the sac9-3 phenotype, that the SAC9 enzyme strongly accumulates in the cytoplasm, where it is presumably not active because not in contact with its lipid substrate. Therefore, the absence of phenotype due to the over expression of SAC9 might be linked to a limiting factor which is responsible for the addressing of SAC9 to the membrane to fulfill its function. Moreover the UBq10 promoter is quite mild, so maybe they are not overexpression. Indeed when imaging the line, no difference in the fluorescent intensity could be observed between these to constructs.

2) Figures 2, and 3: Images shown present cells imaged in different focal planes (for example, cells C1 and C2 in Fig2G, where C1 image is not at the subcortical focal plane indicated by the blue line in Figure 2A). We recognize the difficulty of imaging cells in the same focal plane and the need of recording and analyzing different z-stacks to quantify the structures in the same focal plane in different cells. Please clarify how these images were analyzed. If required, show only cells imaged in the required focal plane.

We are now showing in each picture in which focal planes the picture were taken, (Zi or Zii, see figure 2A) and we added the information about which images was used for the quantification in the material and method

By comparing the signal observed at the cortex of the cell (Zi focal lane) with the signal collected at the center of the cell (Zii focal plane, Figure 2A), we observed and quantified that mCIT-SAC9 labeled more intracellular compartments close to the plasma membrane that a distal position (Figure 2F, G).

3) Figure 2F. The co-expression of the WT version of SAC9 and the mutated SAC9C459A. It is expected by the authors that the mutation in SAC9 affects mainly the function but not the localization of SAC9. In that case, expressing (overexpressing?) SAC9, would not prevent SAC9C459A to localize in the vesicular structures since WT SAC9 would allow proper endocytic trafficking? Please discuss.

We apologize if this was not clear. We indeed expected SAC9C459A to affect function but we had no preconceived idea on localization, because both could be linked (which was indeed the case). We expected C459 to be catalytically inactive, because this residue is predicted to be the catalytic residue within the SAC domain and similar mutations in phosphoinositide phosphatases are well known to affect their catalytic activity. This assumption is consistent with our results showing that SAC9C459A is not functional (unable to rescue the sac9 mutant) and that PI(4,5)P2 misslocalization is affected when C459 is mutated (new data presented in the revised manucript). However, we agree that we haven’t shown formally that this mutant is indeed catalytically dead and we now refer to this mutant as a putative dead version throughout the manuscript.

Concerning the localization of SAC9C459A, we did find an effect of the mutation on the localization: SAC9C459A membrane association is increased compared to that of the wild type SAC9 (increased ratio of fluorescence in intracellular compartments compared to the cytosolic localization). This is likely because the mutation of the catalytic site impacts the on/off interaction between SAC9 and membranes, as seen for other lipid phosphatases in animal systems. Note that we did not find any obvious differences in the localization of SAC9C459A when expressed in a WT background (expressing wild type functional SAC9) or in a sac9 mutant (see Figure 2B and 2H). Thus, there is no direct effect of SAC9 on SAC9C459A localization. In any case, this result aligns well with the scenario listed by referee #3 (i.e. expression of SAC9 does not prevent SAC9C459A to localize in the vesicular structures). We haven’t tried the experiment in which we overexpress wild type SAC9 and analyze the localization of SAC9C459A, because this is a complicated genotype to produce and given the absence of phenotype following overexpression we believe this would not be very informative.

4) Figure 2G: The localization of the native SAC9 at the PM by co-localization with a PM reported, as done for SAC9C459A is required.

We added the colocalization of mCIT-SAC9 with the PM-labelled PI(4,5)P2 biosensor in Figure 2F. This new figure confirms that SAC9 does not localize at the PM but close to the PM.

5) Line 112-113: "mCIT-SAC9 and TdTOM-SAC9 were mainly soluble and excluded from the nucleus"; and Line 114-115: "We observed that mCIT-SAC9C459A was less soluble with a three-fold increase in the number of mCIT-SAC9C459A labeled dotty structures". To employ soluble to describe the localization of a protein does not seem to be the most appropriate term. SAC9 seems to be localized mainly diffused in the cytosol, but the mutated version accumulates in vesicular compartments.

We replaced soluble by “diffused in the cytosol” through out the manuscript.

6) Figure 3A. For the FM4-64 staining, there is a clear difference in the portion of the plasma membrane (PM) stained by FM4-64 in SAC9 WT compared to mutated SAC9. Please, discuss: are they performed using the same conditions? Could the absence of SAC9 (mutant) or the expression of the non-functional SAC9 alter the PM composition (as a complementary information of the Figure 4)?

We are now presenting several images of the FM4-64 in the WT and sac9-3 mutants in supplemental Figure 4. Overall, we did not observe a clear difference between the WT and sac9-3 regarding the portion of the plasma membrane (PM) stained by FM4-64. We confirm that the data have been produced in the same conditions for both genotypes in several replicates.

7) Figure 3C-H: The authors quantified the colocalization of SAC9 and its mutant version with different organelles markers, suggesting that SAC9 colocalized mainly with EE/TGN structures. However, in the presented pictures seems obvious that SAC9 also colocalizes with LE/MVB structures, although in the quantification it is scored with low colocalization numbers. It is not very clearly explained in the Material and Methods section how the colocalization quantification was performed, manually counting or mediating other methods, i.e., Pearson coefficient? How the z-stack was selected? Always to the same distance to the PM? Are the vesicles that are transported to the vacuole excluded from the PI(4,5)P2 to PI4P conversion (it is explained in the text that they are richer in PI3P)? Please include RabF2a in the quantification in Figure 3G.8) Figure 3C-D: The CLC2-RFP labeled structures look significantly different in the SAC9 C459A compared to the SAC9 WT version. Please, discuss.

We apologize for the quality of the figure during the submission process. We believe that with high-resolution images now presented for this resubmission, the details of the localization both for SAC9 and the TGN markers is clearer. In the analyzed pictures we did not observe an obvious colocalization between SAC9 and LE/MVB structures.

We revised Figure 4 and included the plane (Zi or Zii) in each image representative of the plane used for the analysis. We added in the legend: “The plane (Zi or Zii) in each image is mentioned, and the image display is representative for the plane used for the analysis.”

We also included the details for the quantification in the material and method, and the details for the statistical analysis are presented in Supplementary file 1

9) Figure 4: In the text it is described an increase of the number of intracellular compartments label by the PI4P biosensor in the sac9-3 mutant (Figure 4B-C), suggesting that it happens due to a depletion of the PI4P pool at the PM and a relocation to the vesicles. How can the authors explain that this can happen? How could you demonstrate that this increase of the PI4P in the intracellular compartments is from the PM? And why is not happening in the mCIT-PHOSB sensor? Also, in the figure is not clear that there is any increase in the intracellular compartments using the mCIT-P4MSidM sensor. In the cited paper was quantified the number of intracellular compartments labeled with the mCIT-P4MSidM and mCIT-PHOSB, which was practically 0, being very surprising the quantification of the intracellular particles in both cases in the present paper. In these sensors, there is not an obvious decrease (is not quantified) of the PM signal that could explain the increase of the intracellular compartments signal.

In the present study we used an automatic spot detection as we published in Bayle et al., 2017 (bio-protocol), while we used double blind manual counting in Simon et al., 2016. This difference in quantification method explains the slight discrepancies between the two studies (i.e. few spots inside the cells are found in P4M by automatic counting, while no spots could be found when done by eye). Indeed, automatic spot detection has the advantage of being fully unbiased, but it sometimes detects few spots that would not be counted as “endosomes” by the user. This also explains why we do find some spots for the PI(4,5)P2 sensors, while in the wild type, these sensors never localize in intracellular compartments.

We now explained this better in the method section of the revised manuscript.

We do observe the same tendency for PH(OSBP) than for the other two PI4P sensors, however it is true that the effect is not statistically different in that case. We don't know what cause the effect on PI4P, but that it is likely due to the fact that the absence of SAC9 impact PI4P homeostasis, which is the putative product of the enzyme. The quantity of PI4P present at the PM (using the biosensors as a rideout) is greater that at the TGN (Simon et al., Nature plant 2016). Therefore we are not expecting to see a difference in the intensity of the signal at the PM for the PI4P biosensors in sac9-3. Indeed, when depleting the PI4P from the PM using a genetic tool described in Simon et al., Nature Plant 2016, we did not observed a complete disparition of the PI4P biosensor at the PM as we did when depleting the PI(4,5)P2 from the PM using similar approach (Doumane et al., Nature Plant 2021)

10) Line 174-177: "When co-imaging 2xmCH-2xPHFAPP1 together mCIT-TUBBYc in sac9-3-/- (Figure 4D), the dotty structures decorated by the PI(4,5)P2 biosensor -but not with the PI4P biosensor- were observed at the cortex of the cell, at the close vicinity with the plasma membrane (Figure 4D upper panel and 4E), whereas those structures were rarely observed in the internal part of the cell". In that case, what are the structures decorated with PI4P? Co-expression with the different organelle markers or/and SAC9C459A could be explanatory.

As previously shown using biosensors, PI4P localizes at the PM and to one or possibly several post-Golgi/endosomal compartments (early endosomes/TGN and recycling endosomes), and, to a lesser extent, the Golgi apparatus (Simon et al., Plant Journal 2014). Moreover, it was recently shown, by colocalization of the mCIT-2×PH^FAPP1^ PI4P biosensor with the SVs/TGN markers VHA-a1 fused to mRFP and ECHIDNA, that strong colocalization between mCIT-2× or mCIT-3×PH^FAPP1^ and either VHA-a1-mRFP or ECHIDNA was observed upon Metazachlore treatment, confirming that the intracellular accumulation of PI4P sensors occurs at Secretory vesicle part of the TGN when the acyl-chain length of SLs is reduced (Ito et al., Nature comm. 2021). We assume that PI4P accumulates in those compartments in the sac9 mutant like in the wild type. This hypothesis is supported by the BFA sensitivity of the PI4P marker in sac9 mutant, showing that the intracellular compartments labeled by the PI4P sensor aggregates in the BFA bodies (and strongly suggesting a TGN localization similar to the WT situation).

11) Are the intracellular compartments labeled by the PI(4,5)P2 sensors the same where SAC9C459A is accumulating (or where WT SAC9 can be also found)?

We now added data showing that in sac9-3-/- coexpressing the PI(4,5)P2 biosensor mCIT-TUBBYc together with the non-functional SAC9pro:tdTOM-SAC9C459A, mCIT-TUBBYc intracellular structure are not the same but are observed at the same focal plane than tdTOM-SAC9C459A. We added the sentence:

“Besides, in sac9-3^-/-^coexpressing the PI(4,5)P_2_ biosensor mCIT-TUBBYc together with the non-functional SAC9pro:tdTOM-SAC9^C459A^, mCIT-TUBBYc-labelled intracellular structures did not strictly colocalized but were observed at the same Z plan in close association with tdTOM-SAC9^C459A^.”

12) Line 186-188: in vivo time-lapse imaging of PI(4,5)P2 biosensor mCITTUBBYc and mCIT-2xPHPLC in sac9-3-/- mutant revealed that those intracellular structures were mobile in the cortex of root epidermal cells, hence, behaving like intracellular compartments (Supplemental Figure 3D, Supplemental video 2). Are they more or less mobile than in WT? This parameter could already point to the alteration of the endocytic dynamics.

There is no mobile dot in the wild type to be compared with since in the control condition both PI(4,5)P2 biosensors are detected at the PM but not in intracellular compartments (Figure 4-supplemental Figure 1).

13) Figure 6: The FM4-64 staining of PM is not homogenous even in the same plant. Same for the amount of a certain protein, like PIN2. Therefore, the endocytosis should be quantified using the ratio of internal signal/PM signal.14) Figure 6A. There is a decrease in the number of FM4-64 labeled endosomes in the sac9-3 mutant (Figure 6A). Could that be explained due to a possible alteration of the PM density in the sac9-3 mutant? Is that a direct effect or a consequence of an increase of non-labeled FM4-64 vesicles (observed for the PI4,5P2 and PI4P biosensors)? In the case that PM is affected in the sac9-3 mutant, in order to calculate the density of intracellular compartments would be convenient to normalize it with the signal of "available lipids" at the PM to avoid indirect effects.

FM4-64 itself is a lipid that intercalates in membrane. There is no need for "available" lipids. In addition, PI(4,5)P2 and PI4P represent less than a percent of the total phospholipids, which themselves are only about a third of total lipids. Moreover, PI4P and PI(4,5)P2 are embedded in the cytosolic leaflet of the plasma membrane while FM4-64 insert in the outer leaflet of the membrane. Altogether, we thus have no reason to believe FM4-64 labeling itself would be affected in sac9, and this is in line with our confocal observations.

Overall, we did not observe a clear difference in between the WT and sac9-3 regarding the portion of the plasma membrane (PM) stained by FM4-64.

We added in Figure 5-supplemental Figure 1 representative images used for the quantification with FM4-64.

We clarified in the material and method how we quantified the analysis:

“To compare the effects of BFA on FM4-64 we tried to automatically count the number and size of the BFA bodies in Col-0 and sac9-3^-/-^ seedlings, but the analysis was not optimal to treat the images acquired for sac9-3. We therefore manually counted the number of BFA body per cell in multiple samples, using the same region of the root (see Figure 5-supplemental Figure 1). We then compared the results of the BFA treated Col-0 and sac9-3^-/-^ seedlings using a generalized linear mixed model (Poisson family) with image ID (id est root) as a random factor (Type II Wald χ^2^ test : χ^2^ = 33.8, p < 0.001). To compare the effects of BFA on Col-0 and sac9-3^-/-^ seedings expressing PIN2-GFP, we counted manually and compared the treatments BFA-Col-0 with BFA-Sac9-3 using a generalized linear mixed model (Poisson family) with image ID (id est root) as a random factor. (Type II Wald χ^2^ test : χ^2^ = 42.1, p < 0.001). For dissociation index we performed all our statistical analyses in R v. 3.6.1, (R Core Team, 2019), using R studio interface and the packages ggplot2 (Wickham 2016), lme4 (Bates et al., 2014), car (Fox and Weisberg 2011), multcomp (Hothorn et al., 2008) and lsmeans (Lenth and Lenth 2018). ”.

15) Figure 6G: It is already published (Vollmer et al., 2011) that sac9-3 mutant has wall protuberances that are randomly distributed and can be extended as a consequence of the PM accumulation. In that case, is the appearance of these protuberances an indirect effect of the lower endocytosis rate in the sac9-3 mutant? It seems to appear even in absence of inhibitors of the endocytosis. In that case, would these PM-protuberances be enriched with PI(4,5)P2 lipids?

We agree with reviewer #3 that the appearance of these protuberances are possibly an indirect effect of the lower endocytosis rate in the sac9-3 mutant. As suggested, we now include a new set of data showing that in sac9-3, the PI(4,5)P2 biosensor mCIT-TUBBYc localized to the protuberances 3h after treatment with ES9-17 (new Figure 5I).

We changed the text accordingly:

“Strikingly, we observed a much higher number of dome-shaped plasma membrane invagination decorated by the PI(4,5)P_2_ biosensors mCIT-TUBBYc in cells from ES9-17 treated sac9-3^-/-^ (Figure 5I), showing that SAC9 depletion causes over-sensitivity to inhibition of endocytosis. Hypersensitivity to endocytosis inhibition, decreased internalization of the bulk endocytic tracer FM4-64 and defects in PIN2 protein trafficking together indicate that endocytic trafficking is impacted in the absence of SAC9.”

16) Figure 7: SH3P2-sGFP and SAC9C459A are partially co-localizing (although colocalization quantification is missing and required). Are they also colocalizing when WT SAC9 version is used? Additionally, colocalization does not always imply interaction, and Line 285-286 "In planta, SAC9 interacts and colocalizes with the endocytosis component SH3P2" is an overstatement since no interaction assay has been done in planta (Only yeast two-hybrid). To further confirm the interaction of both proteins, other methods like Co-immunoprecipitation (with or without crosslinker) or biotin proximity labeling (PL) (Mair et al., 2019) are recommendable.

We are now including in figure 6 the colocalization between SAC9pro-mCIT-SAC9 x

Ub10pro:TdTOM-SH3P2 and the corresponding quantification.

We encountered difficulties working with the SAC9 protein in vitro or with western blots extracted from Arabidopsis. To confirm the interaction, we thus collaborated with Suayb Üstün (ZMBP Tübingen) and his lab. They expressed SH3P2-GFP in Nicotiana benthamiana leaves and performed IP-MS experiments in the absence (3 replicates) and presence (3 replicates) of a proteasome inhibitor (to increase SAC9 stability). SAC9 was found as a top 10 interactor in all 6 experimental replicates (never found in the GFP only controls) and as the top SH3P2 interactor after proteasome inhibition. We also now include colocalization analysis (representative pictures and quantification) between SH3P2 and wild-type SAC9. In addition, we performed new experiments showing that SH3P2 plasma membrane association is compromised in the sac9 mutant. Together, with our yeast-two hybrid data, we believe that our new IP-MS and in planta colocalization analyses strengthen our conclusion on the SH3P2/SAC9 complex. – The data are presented in Figure 6.

17) Lines 312-314: "Together, we propose that SAC9 represent the long-sought-after enzyme which performs the PI(4,5)P2-to-PI4P conversion during the plant endocytic process (Figure 9)." How is SAC9 recruited to the PM-vesicles for performing its function? It is clear that this question could be difficult to address for this publication, but could this recruitment be through the interaction with other proteins, i.e., SH3P2? Are they mutually needing each other to localize properly? In that case, what is the localization of SAC9 in the sh3p2 mutant background?

We believe that SAC9 is recruited to the PM-vesicles for performing its function through its putative Coiled Coiled domain, since the cytosolic SAC9-deltaCC version does not seems to interact with membranes and do not complement the sac9-3 drawf phenotype. As coied-coiled domain are putative protein-protein interaction platform, we beleive that SAC9 is likely targeted to PM-vesicles via such interaction, possibly SH3P2.

Based on our observation in sac9-3 mutant, it looks like that it is SH3P2 that is regulated by SAC9 more than the contrary. We could not test the localization of SAC9 in the sh3p2 mutant background since literature reports variably on the effects of SH3P2 deficiency. T-DNA alleles that are likely not full knock-outs, did not exhibit obvious phenotypic defects (Ahn et al., 2017; Nagel et al., 2017), while the RNAi line silencing SH3P2 exhibited an arrest of seedling development (Zhuang et al., 2013): We tested this published line, but the construct was silenced, so we were unable to reproduce the results. Moreover, this phenotype was not reproduced with a similar artificial amiRNA line (Nagel et al., 2017).

18) Figure 8A: the representative image shown for sca19-3 TPLATE-GFP is not homogeneously in focus.

We changed the pictures in Figure 7A (now Figure 6A) accordingly.

19) Figure 2D: The reduction of TPLATE-GFP dwell time at the PM in sca9-3 is not obvious based on this figure. Include average, SD, and statistics. Also indicate in methods how these data were obtained.

We agree with the reviewer that the graph previously displayed was not easy to read. We reanalyzed the results and we are now presenting in Figure 6 Histograms of median normalized fluorescence for TPLATE-GFP in WT and SAC9 mutant representing the density of tracks per track lenght. We included in this new analysis the average, SD, and statistics (Figure 6 and Supplementary files 10).

We also rewrote the material and method to explain in details how this analysis, without a priori was performed: “Because manual verification of TPLATE-GFP lifetimes is greatly limited by the number of CCVs that can be detected, particle identification and tracking were performed using ImageJ plugin Trackmate. Trajectories were reconstructed following a three-stage workflow: (i) detection of peaks potentially associated with fluorescent emitters, (ii) quality test and estimation of the subpixel position and (iii) track reconnection. To discriminate between signal and background, particular attention has been paid to the size and shape of the observable objects. Particle of minimum size 0.5 with a threshold of 50 and a contrast >0.04 were filtered, to capture as many spots as possible without background. For many reasons, such as variation in fluorescent intensity, loss of focus or photobleaching, the emitter can be missing for several time points causing the premature stop of tracks. Therefore, the maximum number of frames separating two detections was set to three frames (Bayle et al., 2021). As a final verification, a visual inspection of the tracks can be performed on a reconstituted image, where all the tracks from a video are represented.

2000 tracks were selected per acquisition starting from frame 5 to avoid segmenting truncated tracks. Acquisition were made on 7 hypocotyl cells from three different plants per genotype and per replicate.”

If a significant lower TPLATE-GFP dwell time in sac19-3 is confirmed, please, discuss this result in context with the published data showing an inverse correlation between TPLATE dwell time and endocytosis (longer time>less endocytosis; Wang et al., 2020, https://pubmed.ncbi.nlm.nih.gov/32321842/)

In Wang et al., membrane trafficking was slowed down using low temperature (12°C vs 25°C). Indeed, at 12°C, the dwelltime of TPLATE-GFP at the plasma membrane is longer than at 25°C and in this condition endocytosis is reduced. Although such an inverse correlation between TPLATE dwell time at the plasma membrane and a reduction of the endocytic rate was observed for temperature, to our knowledge, it cannot be generalized. For example, it is possible that interaction between the TPLATE complex and the plasma membrane is less stable in the sac9 mutant, leading to shorter TPLATE dwell time and aborded endocytic events. Such scenario is also compatible with a decrease density of TPLATE-GFP foci (which itself is consistent with a decrease endocytic rate in the mutant). However, we believe the scenario highlighted above is too speculative and we prefer to err on the side of caution and not to discuss how an alterned TPLATE-GFP dynamics translates on endocytosis. We think the important message is that when PI(4,5)P2 patterning is disturbed, protein involved in clathrin-mediated endocytosis at the plasma membrane have an altered localization (i.e., density) and dynamics.

20) Please, re-check the methodology to describe the experimental set-up and data analysis in the detail required to be repeated by other colleagues. Among others, indicate what is "N" and "n" in the graphs, cite the published lines used in the study, and indicate how the particles were chosen for their analysis.

Additional information can now be found in the Supplementary files 2-12, including details about the sampling and the statistics. Description of the N and n were added in the legend of the figures.

21) Avoid over conclusions not supported by data like– Lines 11-12: «it interacts (only shown by Y2H) and colocalizes (not quantified) with the endocytic component Src 11 Homology 3 Domain Protein 2 (SH3P2)»

We are now including in figure 7 the colocalization between SAC9pro-mCIT-SAC9 x

Ub10pro:TdTOM-SH3P2 and the corresponding quantification

We are now presenting a set of independant evidences showing the interaction of SAC9 with SH3P2. Proteomic analysis of the SH3P2-GFP interacting protein confirmed its interaction with SAC9 when transiently expressed in N. Benthamiana. The data are presented in Figure 6.

– Lines 132-133: "catalytically dead SAC9 fusion proteins localize to endosomes and are likely part of the early steps of endocytic trafficking pathway» No data at this point indicate that SAC9 can be part of the early steps of the endocytosis.

Here, we are considering the endocytosis as a whole including the internalization step sensus stricto (i.e., “clathrin-mediated endocytosis”) followed by the subsequent transport of lipid and proteins through the endosomal system. As such, the endocytosis is the process that allows (1) cells to transport particles and molecules across the plasma membrane and (2) the termination of signaling through transport toward the vacuole for degradation. We now include this definition early on in the introduction, so that the term “endocytosis” is not mistaken for “clathrin-mediated endocytosis”.

Having this vision in mind, we consider that SAC9 is likely part of the early steps of endocytic trafficking pathway, since (i) it localizes at the close vicinity to the plasma membrane, (ii) it regulates the PI(4,5)P2 homeostasis, confining it to the plasma membrane, (iii) SAC9 interacts and colocalize with SH3P2 in the close vicinity of the PM (iv) SH3P2 localization at the PM is altered in SAC9, (v) the dynamic of TPLATE is perturbed at the PM, (vi) SAC9 does not localize to late endosome or even most TGN that are more deeply located in the cytoplasm.

– Line 311: "SAC9 localizes to clathrin-coated vesicles close to the plasma membrane» This is not shown in the results.

We show that SAC9 colocalizes with Clathrin light chain marker, CLC2-RFP. Furthermore, we now show that SAC9 is enriched in the cell cortex (Figure 2F, Figure 2-supplemental Figure 1). With these new results and related quantification, we reinforce our conclusion that “SAC9 localizes to clathrin-coated vesicles close to the plasma membrane”. Moreover, it was recently shown by another group that SAC9 is found in the proteome of the isolated Clathrin-coated vesicle, which supports our observations (Dahhan et al., 2022).

[Editors’ note: further revisions were suggested prior to acceptance, as described below.]

The manuscript has been improved but there are some remaining issues that need to be addressed, as outlined below:1) Explain discrepancies in the quantification of cortical vs endoplasmic SAC9-labelled particles (Reviewer 1 points 1 and 2 and Reviewer 2 point 2).Reviewer 1 points 1 and 21. Quantification of cortical vs endoplasmic SAC9-labelled particles has been added to the manuscript in the figure on P. 45 G (the figures are not numbered in the document I was sent for review, so I refer to them by page number in the pdf). However, there is a flaw with the approach: density per cell is a misleading measurement since, in the zii plane, there are of course fewer puncta per area since there is less cytoplasm because the nucleus and vacuole take up about half of the area. Please present data as punta per area of cytoplasm. If the differences in SAC9 puncta density in cortical/endoplasmic cytoplasm do not hold when properly quantified, please revise the text and title accordingly.

We agree with reviewer 1 that the presence of the vacuole at zii could have made it difficult to compare zi and zii. Taking into account this pitfall, we quantified the number of labeled endosomes with FM4-64 in the Col-0 plant at zi and zii. The data is presented in Figure 2. We saw that even if the vacuole is present in the image at zii, the difference in the density of FM4-64 labeled endosomes at zi and zii is not significant (p = 0.098; N=7 roots, n=35 cells) while the density of labeled mCIT-SAC9 is different between zi and zii (p<2.2e-16; N=7 roots, n=35 cells).

We, therefore, consider, based on these results, that the enrichment of SAC9 at zi is not due to an imaging problem.

2. Why is p. 45 G "density of SAC9 puncta per cell" in a range of 0.001-0.003, but presented as "number of SCA9 puncta per cell" in p 53 C in a range of 5-20? Why present two different measures? Why are there 10-fold more SAC9 puncta per cell than FM4-64 puncta in the figure on p 35 C vs D, when the authors described these markers as colocalized (line 160)?

As pointed out by reviewer 1, there are fewer mCIT-SAC9-labeled structures that FM4-64-containing compartment. This is expected since the endosomal domain labeled by FM4-64 is greater than mCIT-SAC9. However, as described in Figure 3 —figure supplement 1, all the mCIT-SAC9-containing compartments are FM4-64 positive. We agree that using two different measures and presenting them across a main and a supplemental figure was confusing. To simplify this, we now present all the quantifications in Figure 2 as the number of endosome observed per cell at a given focal plane.*Reviewer 2 point 2*

I remain somewhat concerned that the analysis would have been more convincing had the authors compared the distribution of wild-type SAC9 relative to intracellular FM4-64. Supplemental Figure 2 imaging comparison was conducted between mCIT-Sac9delta 999-1027 and FM4-64. This is confusing as in figure 1 the authors show that the SAC9 mutant variant lacking the coiled-coil region is cytosolic. More informative would have been the comparison of wild-type SAC9 and internalized FM4-64 rather than PH domain or Lti6b reporters in Zi and Zii focal planes.

We did compare the distribution of wild-type SAC9 relative to intracellular FM4-64. Indeed, we quantified the density of both wild-type mCit-SAC9 and FM4-64 in Zi and Zii. This analysis shows that SAC9-mCIT-labbeled compartment are much denser in Zi as compared to Zii, while this is not the case for FM4-64-labelled endosomes (see also our answer to referee #1 above). We apologize if this was not clearer and we have now amended the text and figure to make this point clearer. In particular, we have now:

1) included the quantification of the number of SAC9 and FM4-64-labelled compartments observed per cell at a given focal plane on the same figure (Figure 2G and H).

2) added a representative image of the double labeling of mCIT-SAC9 with FM4-64 at zi and zii in Figure 2 supplementary figure 1, next to the images showing mCIT-Sac9delta 999-1027 and FM4-64.

The data presented in Figure2-supplemental Figure 1 showing mCIT-Sac9delta 999-1027 and FM4-64 answer a different question raised during the previous round of revision on the fact that the punctuated structures observed for mCit-SAC9 could be an artifact (i.e. cytosolic densities). By this supplemental figure, we confirmed here that both in zi and zii the mutated form of SAC9, which is soluble, does not colocalize with FM4-64 and is not present in cytosolic densities that could be mistaken for endosomes. We believe that this analysis, together with our results showing that SAC9 colocalizes with TGN markers and is sensitive to BFA firmly established that SAC9 localizes in TGN compartments and not “cytosolic densities” as suggested during the review process.

The colocalization of mCIT-SAC9 with a plasma membrane marker, here PH domain of PLC, was also requested in the previous review, to confirm that SAC9 is not present at the PM but only in a subcortical population of endosomes. Moreover, this experiment confirms that mCIT-SAC9 does not colocalize with the PI(4,5)P_2_ biosensor, suggesting a potential role of this phosphatase in removing the pool of PI(4,5)P_2_. Furthermore, the labeling of the plasma membrane by the PI(4,5)P_2_ marker in the wild type provides the cell contour. We believe that it is useful to visualize the cell boundaries when comparing the localization of SAC9 in two different focal plane.

2) Clarify details about the number of transgenic lines (Reviewer 1, point 3 and Reviewer 2 point 6).Reviewer 1, point 33. The number of independent transgenic lines analyzed is still not indicated. The text says "multiple independent lines" (line 103) and evidence from only one or two lines is presented in the supplemental figures. Please present data from at least three independent transgenic lines for each new construct.

We respectfully disagree with this comment from referee #1. We have never heard of a “rule” stating that at least three independent transgenic lines for each new construct need to be analyzed in depth and quantified in a paper. This is certainly not the standard in the field. The number of lines to be analyzed should be depend on the type of analyses that is carried out and not on some arbitrary numbers. Indeed, it might be required to analyzed independent when the result outcome is sensitive to the insertion site (i.e., sensitive to expression level and/or expression pattern) or when it is impossible to control for protein expression (for example absence of tag and antibodies).

As you will see from the discussion below, we have solid arguments showing that the localization results presented in our paper are fully independent from the insertion site of the transgenic line analyzed. Indeed, in this story, we focus on the localization of SAC9 (and some of its mutant variants). We show that the functionality and localization of SAC9 is not dependent upon its expression level (we obtained similar results with SAC9prom and UBQ10prom) or fluorescent tags (we obtain similar results with mCitrine and tdTomato). These results were obtained with independent constructs, which is even more powerful than independent transgenic lines. Indeed, by definition, using independent constructs implies that the resulting lines are independent and that the T-DNA are inserted in different portion of the genome. So, it appears that we get similar localization and complementation independent of (i) the insertion sites, (ii) the fluorescent tag used and (iii) the strength of the promoter (to some extent at least, we off course tested only two promoters).

Please note also that the sac9 mutant has been characterized before (Williams et al., 2005) and that in this paper, we still phenotyped two independent alleles. Furthermore, for complementation analyses (or lack of complementation), we verified that proteins are expressed (using two independent methods: western blots and confocal microscopy). This effectively shows that the absence of complementation for SAC9^C459A^ and SAC9^ΔCC^ is not due to the insertion site (i.e., absence of expression of the transgene due to the insertion). Moreover, for complementation analyses, we present two independent lines for each construct, which again rules out that the outcome of the experiment could be due in any way to the insertion site. Furthermore, and this has been clarified in the text and in the method section, for each complementation analyses, we analyzed 24 independent transgenic lines in T1, with none of them showing a rescue of the sac9 phenotype. We assessed the phenotype of these line qualitatively in T1, but they were not quantified at the time. However, the sac9 phenotype is very strong and fully penetrant, there is no way we could have missed it if any mCIT-SAC9^C459A^ or mCIT-SAC9^ΔCC^ expressing sac9 mutant would have been rescued.

We also want to point out that when we generate new transgenic lines (not only for complementation analyses but for any transgenic line that we generate in the lab), we systematically select and analyze between 20 and 24 independent lines and we spent a lot of time in T2 (and then in T3) in selecting representative transgenic lines. Indeed, after analyzing between 20 and 24 independent lines in T2, we select between 3 and 5 lines that are representative and are single insertion (i.e., segregate with a 3:1 ratio on antibiotics). In T3, we then rescreen plants from each independent lines to confirm the observations obtained in T2. At this stage, we select homozygous plants and we choose one line to carry out further quantification and crosses. This means that while we present quantified data obtained with one line, we in fact have analyzed 20-24 independent lines in T2 and 3-5 independent lines in T3. This procedure is now described in details in the method section:

“For each construct generated in this paper (UBQ10prom:tdTOM-SAC9g/pH, SAC9prom:mCIT-SAC9g/pB, pAtSAC9:mCIT-SAC9g^DEAD^/pB, SAC9prom:TdTOM-SAC9g^DEAD^/pH, pAtSAC9:mCIT-SAC9g^∆CC^/pB, SH3P2shortprom:SH3P2gDNA-tdTOM, pUb10:SH3P2gDNA-tdTOM), between 20 and 24 independent T1 were selected on antibiotics (Basta or hygromycin) and propagated. In T2, all lines were screened using confocal microscopy for fluorescence signal and localization. Between 3 to 5 independent lines with a mono-insertion and showing a consistent, representative expression level and localization were selected and grown to the next generation. Each selected line was reanalyzed in T3 by confocal microscopy to confirm the results obtained in T2 and to select homozygous plants. At this stage, we selected one representative line for in depth analysis of the localization and crosses and two representative lines for in depth analysis of mutant complementation.”

Reviewer 2 point 66. Analyze multiple independent transgenic rescues.This is supported. Supplemental Figures 1A and 1B demonstrate multiple sac9 alleles, a full rescue of sac9 by two different fluorescent tag fusions of SAC9, and the inability of multiple transformants of δ CC SAC9 to rescue sac9 (but only one transformant of C459A inability to rescue sac9 is shown). The additional language regarding independent transformants is helpful.

We are grateful to reviewer 2 for acknowledging the fact that the added information regarding independent transformants and that the additional language is helpful.

Regarding SAC9 C459A, we apologize if this was not clearer but we in fact shown the inability to complement the mutant on multiple independent lines. One line (#1354-12-14) is presented in Figure 1 and another (#1354-15-11F) is presented in Figure 1-supplemental Figure 1A. Moreover, in Figure 1-supplemental figure 1B, we showed by western blot that SAC9 C459A is expressed in both of these 2-independent homozygous T3 lines. We changed the text to make it clearer that independent lines were used to show the absence of complementation with both SAC9 C459A and δ CC SAC9. Finally, we are also confirmed the localization with red fluorescent-tagged SAC9 C459A presented in Figure 6D.

The text now reads: “By contrast to wild-type mCIT-SAC9, we could not find any transgenic lines expressing SAC9pro:mCIT-SAC9^C459A^ that were able to rescue the sac9-3 phenotype, out of 24 independent lines analyzed in T1 (Figure 1B, 1C, Figure 1-Supplemental Figure 1A, 1B). Further analyses on two independent T3 homozygous lines confirmed these initial results and showed that mCIT-SAC9^C459A^ fusions were stable and accumulated to similar extent as wild-type mCIT-SAC9 (Figure 1B, 1C, Figure 1-Supplemental Figure 1A, 1B).”

To avoid any confusion, we also included the following sentence in the legend of figure 1: “Note that a second independent transgenic line is presented for each construct in Figure 1—figure supplement 1”.

3) Clarify details of statistical analysis (Reviewer 1, point 4).4. It is not clear in the main figure legends or text what N and n are in the graphs. If this means seedlings and cells, please clarify how statistical analyses are being conducted (i.e. which is being used as sample size).Inappropriately identifying N drastically affects p-values, and therefore conclusions from statistical analysis. N is the number of independent biological replicates (e.g. plants), not the number of measurements taken (Lord et al., 2020 J Cell Biol). For example, it's unlikely that >1000 independent plants were analyzed in the figure on p. 48 B. Please revise accordingly.

We are now clarifying what is N and n in the legend of each figure. We confirmed that what we call “N” are the number of independent roots/plants and n = the number of cells analyzed. For the figure 7D and E: 21 cells from 10 plants were used, so N = 10 and n=33983 tracks; For sac9, N = 13 n=27142 tracks. We now specified it in the legend of the figure.

4) Clarify details of the colocalization experiments (Reviewer 2 points 4 and 8 and Reviewer 3 points 1 and 2).Reviewer 2 points 44. Address disconnect between images and quantitation of images and/or image mCIT (or other tags) relative to markers used.This is somewhat supported. Harmonization of the majority of images in the manuscript is appreciated but is not totally consistent (e.g. mCIT-TUBBYc imaging in Figure 4A and 4C; imaging of SAC9 between mock and BFA treatments occurs in zi and zii, respectively). Figure 3 remains the same as in the previous manuscript draft, where the authors had included images of RabF2 colocalization with C459A SAC9 but not the corresponding quantitation and the authors had included the quantitation of VTI12 with WT SAC9 but did not show the image. In this revision, the authors have supplemented the quantitation in Figure 3F with 'representative' colocalization image of WT SAC9 and VTI12 in Figure 3-supplemental Figure 1, but it is not apparent if the image shown in the supplement is actually quantitated in the main manuscript figures. (Note: the panel in Figure 3-supplemental Figure 1 is not labeled as VTI12 but instead as W13R – is this the same? Authors need to make it clear, as the figure legend for Figure 3-supplemental Figure 1 refers to VTI12.)The authors do not address the fact that C459A SAC9 colocalization with CLC2 is quantitated but not shown by images or that RabF2 colocalization with C459A SAC9 is not quantitated (Figures 3G and 3H). I would ask that the authors confirm that the quantitation of the colocalization between VTI12 and WT SAC9 directly corresponds to the image shown or otherwise replace the quantitation in Figure 3 with that directly corresponding to Supplemental Figure 3.In response to the suggestion from reviewer 2, there is no imaging of mCIT alone relative to the other markers used.The additional data corresponding to the loss of the coiled-coil domain (SAC9-deltaCC) resulting in the loss of endosomal localization pattern is interesting to note. And, while the inability of this variant to rescue the sac9 mutant indeed supports that this feature is important for the function of the protein, it does not necessarily indicate that the coiled-coil region mediates membrane association. But, this is asserted only in the response to reviewers and not in the manuscript itself.

We revised figure 3E, which is now showing all the colocalization between the functional mCIT-SAC9 and CLC2, VTI12, Rab A1g, Rab D1, Rab F2a, Got1p, and the corresponding quantification in D. We decided to move the colocalization (and related quantification) of mCit-SACC459A to Figure 3- Supplemental Figure 2, as we agree that it is more important to focus on the wild-type protein. Because the quantification between mCIT-SAC9C459A and Rab F2a is missing, we removed the corresponding image from the Supplementary file and revised the manuscript accordingly. Overall, we believe that the new figure 3 provides enough information about SAC9 localization in the EE/TGN.

Regarding VTI12 (which is also known as Waveline #13, or W13, we apologize for the double labeling, which has now been corrected), we confirm here that the image shown was one of the images used for quantification.

We revised the figure, the text, and the legend of the figure to clarify this point.

Reviewer 2 points 88. Validate choice of RabD1 as a post-Golgi endosomal markers. Pinheiro et al., support the role of RabD1 as a post-Golgi marker as it colocalizes with FM4-64 and VHAa1.The authors argue that the Pinheiro paper does not quantify these interactions, and so they have used as support the Geldner et al., Plant J 2009 paper. However, in the Geldner paper, the assignment of wave25 (RabD1) as a post-Golgi/endosomal marker protein appears arbitrary. Indeed in Table 2 Remarks that Geldner and colleagues state that RabD1 (wave25) is similar to wave lines 29 and 33 (i.e. RabD2a and D2b) which are assigned as Intermediate Golgi/endosomal. This is more similar to what was reported by Pinheiro and thus I feel that the authors are not justified in relying on RabD1 as a post-Golgi/endosomal marker.

We removed the label from the figures and we are now comparing the colocalization between TGN markers and MVB or Golgi markers, without specifying the subdomains of the TGN. We revised the text accordingly. We agree with Reviewer #2 that the fact that SAC9 partially colocalizes with RabD1 is fully coherent with the idea that RabD1 has a broader localization than SAC9.

Reviewer 3 points 1 and 21) It is not clarified yet in the material and methods how the co-localization was quantified. Please detail this point.

We added a section in the material and method and rephrased the paragraph to clarify this analysis.

This new methods paragraph reads:

“Quantification of mCIT-SAC9 compartment densities and colocalization with compartments markers

Because the signal of mCIT-SAC9 is mainly diffused in the cytosol, no automatic spot detection could be used for quantification of densities and colocalization analyses in Figure 2, 3, and 6. Therefore, for comparing the number of intracellular compartments containing mCIT-SAC9 or mCIT-SAC9^C459A^ protein-fusions per cell across conditions, we manually counted them and used either a generalized linear mixed-effect model (Poisson family) for counting comparisons or a linear mixed effect model (and associated ANOVAs) for density comparisons, accounting for image ID (id est root) as a random factor.

Since the localization of the marker for the membrane compartment was larger in z compared to the restricted localization of SAC9 (only present close to the surface of the cell), we counted the number of mCIT-SAC9 labeled structures which were also labelled by the compartment markers in the cell cortex (Zi plane). The percentage of endosomes labelled by mCIT-SAC9 or mCIT-SAC9C459A colocalizing with a given marker, counted manually are presented in the graphs. Positive colocalization was called when the compartment marker was present as a dotted structured overlaying the mCit-SAC9 signal.

The same approach was used to deduce the localization of the mutated version of SAC9. After running our mixed models, we subsequently computed two-sided Tukey post hoc tests (function glht in R package “multcomp”, Horthorn et al., 2008) to specifically compare each pair of conditions.”

2) Co-localization of SAC9 and organelle markers. It was not discussed why mCit-SAC9-C459A seems to co-localize more importantly with LE-MVB markers, compared with the SAC9 wt version. Quantification is not included in Figure 3G.

We revised figure 3E, which is now showing all the colocalization between the functional mCIT-SAC9 and CLC2, VTI12, Rab A1g, Rab D1, Rab F2a, Got1p, and the corresponding quantification in D.

We are confident that the new figure 3 provides enough information about SAC9 localization with the EE/TGN. We, therefore, moved the colocalization analysis of mCIT-SAC9C459A in Figure 3- Supplemental Figure 2, showing that the mutated form of SAC9 also colocalizes with EE/TGN. Because the quantification between mCIT-SAC9C459A and Rab F2a is missing, we remove the corresponding image from the Supplementary file.

Please also consider the detailed comments from reviewers, but a point-by-point response to all of their comments will not be strictly necessary for a revised version.Reviewer #1 (Recommendations for the authors):The authors have substantially revised the manuscript to address many of my previous comments, including adding several new experiments. They have analyzed π biosensors in the SAC9(C459A) to provide the support that this mutation affects SCA9 enzymatic activity. The evidence documenting SAC9 interactions with SH3P2 is now much stronger with the addition of SH3P2-GFP IP-MS experiments. The localization data are now much better aligned with the authors' claims and much more clearly communicated. The authors have also provided a very detailed response to the previous reviewer comments. However, several of my previous comments have not been adequately addressed:1. Quantification of cortical vs endoplasmic SAC9-labelled particles has been added to the manuscript in the figure on P. 45 G (the figures are not numbered in the document I was sent for review, so I refer to them by page number in the pdf). However, there is a flaw with the approach: density per cell is a misleading measurement since, in the zii plane, there are of course fewer puncta per area since there is less cytoplasm because the nucleus and vacuole take up about half of the area. Please present data as punta per area of cytoplasm. If the differences in SAC9 puncta density in cortical/endoplasmic cytoplasm do not hold when properly quantified, please revise the text and title accordingly.

See the answer to the main points above.

2. Why is p. 45 G "density of SAC9 puncta per cell" in a range of 0.001-0.003, but presented as "number of SCA9 puncta per cell" in p 53 C in a range of 5-20? Why present two different measures? Why are there 10-fold more SAC9 puncta per cell than FM4-64 puncta in the figure on p 35 C vs D, when the authors described these markers as colocalized (line 160)?

See the answer to the main points above*.*

3. The number of independent transgenic lines analyzed is still not indicated. The text says "multiple independent lines" (line 103) and evidence from only one or two lines is presented in the supplemental figures. Please present data from at least three independent transgenic lines for each new construct.

See the answer to the main points above.

4. It is not clear in the main figure legends or text what N and n are in the graphs. If this means seedlings and cells, please clarify how statistical analyses are being conducted (i.e. which is being used as sample size). Inappropriately identifying N drastically affects p-values, and therefore conclusions from statistical analysis. N is the number of independent biological replicates (e.g. plants), not the number of measurements taken (Lord et al., 2020 J Cell Biol). For example, it's unlikely that >1000 independent plants were analyzed in the figure on p. 48 B. Please revise accordingly.

See the answer to the main points above*.*

5. Introduction line 27: why redefine endocytosis, rather than just calling this "endocytic trafficking" throughout (i.e. as you do on line 41)? This will be less confusing to the broad readership of eLife.

We replaced “endocytosis”, by “endocytic trafficking”.

6. The article requires careful proofreading, particularly for tense use/agreement, article use, and number agreement. As just a few examples: intro line 18 should read "abundance" not "abundant", in intro line 31: remove "the" from "the endocytosis", intro line 54: "FM4-64 experiment" should be plural, intro line 78: "the sac9 mutants is dwarf" should be "sac9 mutants are dwarf"

Corrected.

7. Timestamp and scale bars are missing in supplemental videos.

The details about the timeframe are described in the legend of the videos.

Reviewer #2 (Recommendations for the authors):In my opinion, the authors have overall satisfactorily addressed the editor's and my major comments/concerns. As detailed below I have only a few remaining issues (that do not require further experimentation) that I feel the authors should address.Response to Editor's comments/concerns:1. Imaging PIP2 marker in a sac9 C459A mutant background to see changes in PI distribution as in loss-of-function sac9 backgroundThis is supported. The marker mCIT-TUBBYc is imaged in WT, sac9, and sac9C459A lines. The images of both mutant backgrounds look identical, but this is not quantitated. It should be noted that imaging of mCIT-TUBBYc in the WT was done at the zii level (4A) while in sac9 is at zii and zi (4A and 4C) but C459A is in zi (4C).

Image of zii in WT and sac9-3 can be found in figure 4A; while we can find the image for zi for the WT (figure 4-supplemental figure 1G), sac9-3 and sac9-3-/- expressing mCIT-TUBBY in figure 4C. As suggested, we added an image of zi for the sac9-3 het in figure 4C.

2. Confirm cortical nature of SAC9: image this relative to TGN/EE markers in multiple layers of the cell AND clarify when each image is taken in the cortex or interior of the cellOverall the analysis is improved. Most images indicate whether the plane of focus is at the cell cortex or interior using the appreciated zi vs zii notation. The images and quantitation in Figures 2F and 2G indicate that there is a statistically significant difference in the number of endosomes labeled by SAC9 between the cortex and interior.I remain somewhat concerned that the analysis would have been more convincing had the authors compared the distribution of wild-type SAC9 relative to intracellular FM4-64. Supplemental Figure 2 imaging comparison was conducted between mCIT-Sac9delta 999-1027 and FM4-64. This is confusing as in figure 1 the authors show that the SAC9 mutant variant lacking the coiled coil region is cytosolic. More informative would have been the comparison of wild-type SAC9 and internalized FM4-64 rather than PH domain or Lti6b reporters in Zi and Zii focal planes.

See the answer to the main points above.

3. Verification of SH3P2 and SAC9 interaction by additional method (e.g. coIP or FRET)I feel that the additional reciprocal co-IP data presented in Figure 6 showing that SH3P2-GFP interacts with tobacco SAC9 addresses this concern. Additionally, confocal microscopy shows that localization of SH3P2 to the plasma membrane is strongly affected by the loss of SAC9.4. Address disconnect between images and quantitation of images and/or image mCIT (or other tags) relative to markers used.This is somewhat supported. Harmonization of the majority of images in the manuscript is appreciated but is not totally consistent (e.g. mCIT-TUBBYc imaging in Figure 4A and 4C; imaging of SAC9 between mock and BFA treatments occurs in zi and zii, respectively). Figure 3 remains the same as in the previous manuscript draft, where the authors had included images of RabF2 colocalization with C459A SAC9 but not the corresponding quantitation and the authors had included the quantitation of VTI12 with WT SAC9 but did not show the image. In this revision, the authors have supplemented the quantitation in Figure 3F with 'representative' colocalization image of WT SAC9 and VTI12 in Figure 3-supplemental Figure 1, but it is not apparent if the image shown in the supplement is actually quantitated in the main manuscript figures. (Note: the panel in Figure 3-supplemental Figure 1 is not labeled as VTI12 but instead as W13R - is this the same? Authors need to make it clear, as the figure legend for Figure 3-supplemental Figure 1 refers to VTI12.) The authors do not address the fact that C459A SAC9 colocalization with CLC2 is quantitated but not shown by images or that RabF2 colocalization with C459A SAC9 is not quantitated (Figures 3G and 3H). I would ask that the authors confirm that the quantitation of the colocalization between VTI12 and WT SAC9 directly corresponds to the image shown or otherwise replace the quantitation in Figure 3 with that directly corresponding to Supplemental Figure 3.In response to the suggestion from reviewer 2, there is no imaging of mCIT alone relative to the other markers used.The additional data corresponding to the loss of the coiled-coil domain (SAC9-deltaCC) resulting in the loss of endosomal localization pattern is interesting to note. And, while the inability of this variant to rescue the sac9 mutant indeed supports that this feature is important for the function of the protein, it does not necessarily indicate that the coiled-coil region mediates membrane association. But, this is asserted only in the response to reviewers and not in the manuscript itself.

See the answer to the main points above.

5. Confirm that loss of endocytic proteins at PM in sac9 is not due to their downregulation. Possibly also address whether sac9 affects other TGN/EE-related processes, e.g. post-Golgi trafficking.Overall the authors have addressed this concern however the quality of the immunoblot in Supplementary data figure 6 is low and the data is not quantitated. Immunoblot analyses show that with equal loading (as assessed by Ponceau and anti-tubulin) of total protein extracts from WT and sac9 plants, the GFP signal of TPLATE-GFP or SH3P2-GFP does not change in the sac9 mutant. Curious - what are the roughly 67 kDa bands present between the columns where WT and sac9 total protein extract were loaded? The data in Figures 6F and 6G show better than the ratio of SH3P2 in the cytosol relative to the PM is increased in sac9 relative to WT and is actually more convincing in showing that downregulation contributes less to decreased abundance of SH3P2 at the PM than does the change in SH3P2 re-distribution to the cytosol.The authors do have the tools to assess disruption of post-Golgi trafficking in sac9 backgrounds, as they already have a PIN2::PIN2-GFP in sac9 line which was used for a PIN2-GFP localization to BFA body assay in Figure 5. This assay has been used to show that endocytosis is disrupted in sac9 background because the distribution of cells with BFA bodies labeled by PIN2-GFP is decreased/shifted to the left compared to WT. Problematically, cycloheximide has not been used in this BFA assay. The internalization of FM4-64 in Figures 5A-5D is more appropriate to show that endocytosis is impaired, and perhaps the BFA/PIN2-GFP internalization assay could be moved to the supplement of the manuscript. But, ultimately, I am satisfied by the authors' statement in the Discussion that likely multiple explanations exist for why sac9 displays impaired endocytosis independently of/concomitant with PIP-related factors.6. Analyze multiple independent transgenic rescues.This is supported. Supplemental Figures 1A and 1B demonstrate multiple sac9 alleles, a full rescue of sac9 by two different fluorescent tag fusions of SAC9, and the inability of multiple transformants of delta CC SAC9 to rescue sac9 (but only one transformant of C459A inability to rescue sac9 is shown). The additional language regarding independent transformants is helpful.7. Quantitate FM4-64 internalization defects of sac9 lines using images where FM4-64 staining at the PM is comparable to WT.This is supported. Supplemental Figure 5 shows that FM4-64 staining at the PM in sac9 is comparable to WT.Response to Reviewer #2 comments/concerns:1. Validate the use of FM4-64 labeling of BFA bodies in sac9 by showing that formation of BFA bodies in this mutant is similar to WT (e.g. by showing that BFA body formation labeled by TGN/EE markers is unaltered in sac9).Although the author did not directly address this concern the authors effectively argue that endocytosis is impaired in sac9 mutants due to impaired internalization of FM4-64 and altered dynamics of endocytic protein players, and they also agree with the reviewer that, as is, their experiment is insufficient to show whether a combination of impairment of endocytosis and/or post-Golgi trafficking occurs in sac9. They have included a discussion about the interpretation of the BFA results (lines 474-476). Note, the authors should consider an additional alternative that BFA body formation is affected in the sac9 mutant due to defects in endocytic/exocytic which alter TGN/EE function as shown in the study by Yan et al Plant Cell 2021

We rephrased it accordingly “We showed that the number of BFA bodies labeled with FM4-64 in sac9-3 vs wild-type is reduced. Such reduction may be due to (i) a lower amount of BFA bodies formed per cell (i.e., the number per cell area or volume), (ii) reduced FM4-64 internalization from the cell surface, (iii) defects in the balance between endocytic and exocytic trafficking which alter TGN/EE function, or (iv) a combination of those. In either case, such decrease suggests that membrane trafficking flux through the endosomal system is impacted in sac9.”

2. Explain why colocalization between SH3P2 and SAC9 was not performed using WT SAC9.This is mostly resolved by Figures 6D and 6E which show colocalization between SH3P2 and WT SAC9 as well as between SH3P2 and SAC9 C459A variant. However, a minor concern is that the imaging experiment with WT SAC9 is performed in the zi plane while the experiment with the C459A variant has been performed in the zii plane, and both are quantitated in panel 6E where the y-axis shows SAC9/SH3P2 colocalization in the zi plane. Authors should consider addressing this.

We confirm here that all the colocalization analyses were done in Zi, as shown in Figure 6D. We inverted the panels in Figure 6D for clarity, and added in the legend of the graph “colocalization between SAC9 and SH3P2-tdTOM in Zi (%)”

3. Address localization of PIP2 probe, mCIT-TUBBYc, to the cytoplasm/nucleus as well as PM in comparison to 2xPH probe which localizes only to PM.Resolved by the new language.4. Address apparent increase in intracellular PI4P levels in sac9 and how changes in PI4P levels fit into the model (Figure 8).The authors state that we will not be able to resolve the interplay between the effects of PI4P and PIP2 in mediating sac9 and acknowledge that the effect on ARF1 GTPase is unknown. But, they do effectively argue that the observed increase in intracellular PI4P levels as measured by probes that do not localize concomitantly to the TGN/EE due to interaction with ARF1, e.g. mCIT-P4M, provides evidence that the ARF1 effect is not critical here. But, the authors do not satisfyingly address the role of PI4P in their model in this response.

We added a note on the legend of the Figure 8: “Note that in this model, the slight increase observed with PI4P sensors in intracellular compartments is not included in this model.”

5. Validate the use of ARF protein binding defective mutant, FAPP-E50A, as opposed to WT variant used elsewhere.Satisfactorily, addressed by moving figure to supplemental materials.6. Indicate whether the image shown at higher magnification in Figure 4F is from the Zi or Zii plane of focus.Resolved by removing the image.7. The authors need to provide more information in the manuscript text or methods section to explain how they calculated/quantitated the 'density' of intracellular puncta in the various backgrounds. Does density refer to the number of endosomes labeled by FM4-64, e.g. per cell? Or, does it refer to the number of intracellular puncta relative to the area of the cell imaged? Similarly, how was the number of BFA bodies quantitated (Figure 6)?Satisfactorily addressed by the addition of quantitation methodology to Methods8. Validate choice of RabD1 as a post-Golgi endosomal markers. Pinheiro et al. support the role of RabD1 as a post-Golgi marker as it colocalizes with FM4-64 and VHAa1.The authors argue that the Pinheiro paper does not quantify these interactions, and so they have used as support the Geldner et al. Plant J 2009 paper. However, in the Geldner paper, the assignment of wave25 (RabD1) as a post-Golgi/endosomal marker protein appears arbitrary. Indeed in Table 2 Remarks that Geldner and colleagues state that RabD1 (wave25) is similar to wave lines 29 and 33 (i.e. RabD2a and D2b) which are assigned as Intermediate Golgi/endosomal. This is more similar to what was reported by Pinheiro and thus I feel that the authors are not justified in relying on RabD1 as a post-Golgi/endosomal marker.

See the answer to the main points above.

9. Support decrease in PM associated SH3P2 with data showing that total levels of SH3P2 are not changed.Overall satisfactorySee the response to the editor's comments/concerns point #5.Reviewer #3 (Recommendations for the authors):Doumane and colleagues have addressed most of the reviewer comments. Overall we are satisfied with the revised version of the manuscript. Most of the questions/comments have been answered and appear to support the authors' findings as written in the manuscript.Nevertheless, for better clarification, and to fully support the publication of the manuscript, it will be beneficial to address some points not answered during the first revision:1) It is not clarified yet in the material and methods how the co-localization was quantified. Please detail this point.

See the answer to the main points above.

2) Co-localization of SAC9 and organelle markers. It was not discussed why mCit-SAC9-C459A seems to co-localize more importantly with LE-MVB markers, compared with the SAC9 wt version. Quantification is not included in Figure 3G.

See the answer to the main points above.

3) For the PI4P biosensors (Fig 4), it would be recommendable to use pictures that are representative of the quantification of Fig4B. For instance, for mCIT-P4MSidM, 0 intracellular compartments are visible for sac9-3, but in the quantification, it is shown a clear increase of the intracellular compartments (two stars).With regard to the clarification of the focal plane (commented in the first revision), would be recommendable to use replace the picture of sac9-3 x mCIT2xPHPLC (that is Zi) for a picture of the Zii plane, to be consistent with the rest of the pictures present in the panel (all of them in the plane Zii).

We replaced the picture for mCIT2xPHPLC to harmonize the plane used to compare the data in Figure 4A.

4) The manuscript would benefit from adding all the explanations included in the "response to reviewers". Ex. from our comment #9, among others.

The response to the reviewer is public and will be therefore part of the online information. We, therefore, did not include all the responses to the reviewer in the manuscript.

Suggestion:The classification of C1 and C2 (Fig 2A), it is not used in the rest of the paper, so it would be recommendable to erase it.

We used the classification C1 and C2 showing that we are imaging the same cells at two different Z (Zi and Zii) in Figure 2A but also in Figure 2F and H, Figure 4D, Figure 3-supplemental Figure 1, so we feel that it is helpful to keep it.